# Reward Distance Comparisons Under Transition Sparsity

## Abstract

Reward comparisons are vital for evaluating differences in agent behaviors induced by a set of reward functions. Most conventional techniques employ optimized policies to derive these behaviors; however, learning these policies can be computationally expensive and susceptible to safety concerns. Direct reward comparison techniques obviate policy learning but suffer from transition sparsity, where only a small subset of transitions are sampled due to data collection challenges and feasibility constraints. Existing state-of-the-art direct reward comparison methods are ill-suited for these sparse conditions since they require high transition coverage, where the majority of transitions from a given coverage distribution are sampled. When this requirement is not satisfied, a distribution mismatch between sampled and expected transitions can occur, introducing significant errors. This paper introduces the Sparsity Agnostic Reward Distance (SARD) pseudometric, designed to eliminate the need for high transition coverage by accommodating diverse sample distributions, likely common under transition sparsity. We provide theoretical justifications for SARD's robustness and conduct empirical studies to demonstrate its practical efficacy across various domains, namely Gridworld, Bouncing Balls, Drone Combat, and StarCraft 2.

## 1 Introduction

In sequential decision problems, reward functions often serve as the most "succinct, robust, and transferable" representations of a task (Ng & Russell, 2000), encapsulating agent goals, social norms, and intelligence (Silver et al., 2021; Zahavy et al., 2021; Singh et al., 2009). For problems where a reward function is specified and the goal is to find an optimal policy that maximizes cumulative rewards, Reinforcement Learning (RL) is predominantly employed (Sutton & Barto, 2018). Conversely, when a reward function is complex or difficult to specify, and past expert demonstrations (or policies) are available, the reward function can be learned via Inverse Reinforcement Learning (IRL) (Ng & Russell, 2000).

In both RL and IRL contexts, reward functions govern agent decision-making, and reward comparisons can help assess the similarity of these functions in terms of the behaviors that they induce. These comparisons could be useful for: (1) *Evaluating Agent Behaviors* – By comparing how different reward functions align or differ through specified similarity measures, agent rewards can be grouped (clustering) or categorized (classification), to reason and interpret the agents' behaviors. This can be useful in IRL domains, where there is need to extract meaning from intrinsic rewards computed to represent agent preferences and motivations (Ng & Russell, 2000). For instance, in sport domains such as hockey, reward comparisons could be useful in inferring player rankings and their decision-making strategies (Luo et al., 2020). (2) *Initial Reward Screening* – In RL domains, direct reward comparisons (without computing policies) could serve as a preliminary step to quickly identify rewards that will achieve a spectrum of desired behaviors before actual training. This could be beneficial in scenarios where multiple possible reward configurations exist, but some might be more efficient. For example, it might be important to distinguish rewards that support defensive versus offensive strategies in military scenarios (Van Evera, 1998), or competitive versus cooperative behaviors in team settings (Santos & Nyanhongo, 2019). (3) *Addressing Reward Sparsity*[1] – Reward comparisons could also tackle issues such as reward sparsity, by identifying more informative, and easier-to-learn reward functions, that might be similar in terms of optimal policies but more desirable than sparse reward functions.

---

[1]Transition sparsity arises when a minority of transitions are sampled. This is different from the concept of reward sparsity, which occurs when rewards are infrequent or sparse, making RL tasks difficult.

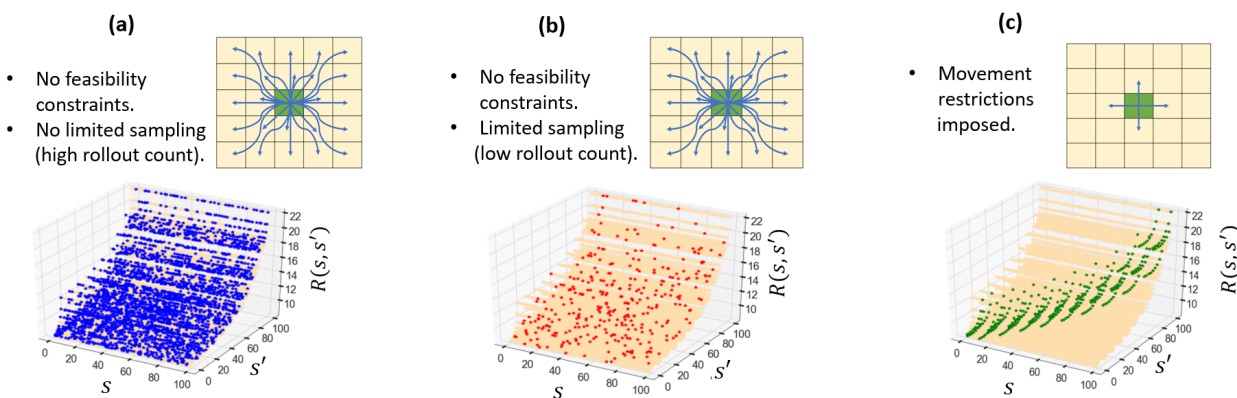

Figure 1: (Transition Sparsity in a $10 \times 10$ Gridworld Domain) In this illustration, each transition starts from a starting state $s$ and ends in a destination state $s'$. For clarity in visualization, we consider action-independent rewards such that actions can be omitted, $R(s, s')$. In (**a**), high transition coverage results from a high rollout count (number of policy rollouts) in the absence of feasibility constraints, leading to the majority of transitions being sampled (blue points). In (**b**), low coverage results from a low rollout count in the absence of feasibility constraints, leading to fewer sampled transitions (red points). In (**c**), low coverage results from feasibility constraints, such as movement restrictions that only allow actions to adjacent cells; which can significantly reduce the space of sampled transitions (green points) irrespective of rollout count.

The task of reward comparisons aims to identify the similarity among a collection of reward functions. This can be done through pairwise comparisons where the similarity distance $D(R_A, R_B)$ between two reward functions, $R_A$ and $R_B$ (vectors, not scalars), is computed. The similarity distance should reflect variations not only in magnitude but also in the preferences and behaviors induced by the reward functions. This is characterized by the property of policy invariance, which ensures that reward functions yielding the same optimal policies are considered similar even if their numerical reward values differ (Ng et al., 1999). This makes direct reward comparisons via distance measures such as Euclidean or Kullback-Leibler (KL) divergence unfavorable since these distances do not maintain policy invariance. To satisfy policy invariance, traditional reward comparison techniques have adopted indirect approaches, which compare behaviors derived from optimized policies generated from the reward functions under comparison (Arora & Doshi, 2021). However, these indirect approaches pose the following challenges: (1) they can be slow and resource-intensive due to policy learning via RL, and (2) policy learning might not be favorable in critical environments such as healthcare or autonomous vehicles, where safety considerations are important (Amodei et al., 2016; Thomas et al., 2021). Therefore, the development of direct reward comparison methods that bypass policy learning while maintaining policy invariance is highly important.

To achieve policy invariance in direct reward comparisons, Gleave et al. (2020) introduced the Equivalent Policy Invariant Comparison (EPIC) pseudometric. Given the task to compare two reward functions, EPIC first performs reward canonicalization to express the rewards in a standardized form by removing shaping; and then computes the Pearson distance to calculate the difference between the canonical reward functions. Although theoretically rigorous, EPIC falls short in practice since it is designed to compare reward functions under high transition coverage, when the majority of transitions within the space of explored states and actions, are sampled. In practical scenarios, this might be impractical due to transition sparsity—a condition where only a minority of transitions are sampled. This sparsity makes EPIC vulnerable to unsampled transitions, which can distort the computation of reward expectations in canonicalization (see Section 3.4). Transition sparsity can be attributed to: (1) **limited sampling** - when challenges in data collection result in few transitions being sampled; and (2) **feasibility constraints** - where environmental or agent-specific limitations restrict certain transitions. Consider, for instance, a standard $10 \times 10$ Gridworld domain where each $(x, y)$ coordinate represents a state. The total number of possible transitions is at least 10000 if one or more actions exist between any two states (see Figure 1). However, feasibility constraints such as movement restrictions might significantly limit the transitions explored. For example, an agent that can only take four

single-step cardinal actions per given state will explore fewer than 400 transitions (based on 100 states $\times$ 4 directions; the exact number is 360 due to boundary limitations, such as the inability to transition from $(0,0)$ to $(-1,0)$), as shown in Figure 1c. To illustrate the impact of limited sampling, consider a scenario where transitions are sampled via policy rollouts (trajectory simulations from a given policy). Assuming that factors such as feasibility constraints, the transition model, and the policy rollout method are kept constant, the extent of sampled transitions is directly influenced by the number of policy rollouts. When the number of rollouts is low, fewer transitions will likely be sampled, and conversely, when the number of rollouts is high, a greater proportion of transitions is likely to be sampled (see Figure 1a and 1b).

**Contributions**   In this paper, we introduce the Sparsity Agnostic Reward Distance (SARD) pseudometric, designed to improve reward comparisons in environments characterized by high transition sparsity. SARD demonstrates greater robustness compared to existing pseudometrics (such as EPIC), which assume reward samples with high transition coverage. SARD's strength lies in its ability to integrate reward samples with diverse transition distributions, which are common in scenarios with low coverage. We provide a theoretical justification for SARD's robustness and demonstrate its superiority through experiments in four domains of varying complexity: Gridworld, Bouncing Balls, Drone Combat, and a StarCraft 2 environment. For the simpler domains, Gridworld and Bouncing Balls, we evaluate SARD against manually defined factors such as nonlinear reward functions and feasibility constraints, to fully understand its strengths and limitations under controlled conditions. In the more complex domains, StarCraft 2 and Drone Combat, we assess SARD in battlefield scenarios characterized by large state and action spaces, to gauge how it will likely to perform in realistic settings. Our final experiment explores a novel and practical application of these pseudometrics as distance measures within a k-nearest neighbors algorithm, tailored to classify agent behaviors based on reward functions computed via IRL. Empirical results highlight SARD's superior performance, evidenced by its ability to find higher similarity between rewards generated from the same agents and higher variation between rewards from different agents. These results emphasize the crucial need to accommodate transition sparsity in reward comparisons.

## 2   Related Works

The EPIC pseudometric is the first direct reward comparison technique that circumvents policy learning while maintaining policy invariance (Gleave et al., 2020). In practical settings, EPIC's major limitation is that it is designed to compare rewards under high transition coverage. In scenarios characterized by transition sparsity, EPIC underperforms due to its high sensitivity to unsampled transitions, which can cause a distribution mismatch between sampled and expected transitions, distorting the computation of reward expectations (refer to Section 3.4). This limitation has been observed by Wulfe et al. (2022), who suggested the Dynamics-Aware Reward Distance (DARD) pseudometric. DARD improves on EPIC by considering transitions that are closer to being physically realizable; however, it remains sensitive to unsampled transitions, which limits its efficacy. Additionally, DARD requires access to transition models, which might be inaccessible or difficult to reliably estimate under transition sparsity.

Skalse et al. (2023) also introduced a family of reward comparison pseudometrics, known as Standardized Reward Comparisons (STARC). These pseudometrics are shown to induce lower and upper bounds on worst-case regret, implying that a small STARC difference between two reward functions corresponds to similar behaviors. Among the different STARC canonical forms explored, the Value-Adjusted Levelling (VAL) function is shown to have a higher correlation to regret compared to both EPIC and DARD. While an improvement, the major flaw with VAL is that it involves the estimation of value functions, which carry a significantly higher computational cost compared to both EPIC and DARD. To compute VAL in small environments, value iteration can be performed, which has polynomial complexity in terms of the state and action spaces (Skalse et al., 2023). For larger environments, value functions can be approximated via neural networks updated with on-policy Bellman updates (Skalse et al., 2023). Since the primary motivation for direct reward comparisons is to eliminate policy learning to lower computational costs, incorporating value functions in these pseudometrics is somewhat contradictory. Therefore, we do not consider VAL as a viable direct reward comparisons pseudometric.

The task of reward comparisons lies within the broader theme of reward evaluations, which aim to explain or interpret the relationship between rewards and agent behavior. Some notable works tackling this theme, include, Lambert et al. (2024), who developed benchmarks to evaluate reward models in Large Language Models (LLMs), which are often trained using RL via human feedback (RLHF), to align with human values. These benchmarks assess criteria such as communication, safety and reasoning capabilities across a variety of reward models. In another line of work, Mahmud et al. (2023) presented a framework leveraging human explanations to evaluate and realign rewards for agents trained via IRL on limited data. Lastly, Russell & Santos (2019) proposed a method that examines the consistency between global and local explanations, to determine the extent to which a reward model captured complex agent behavior. Similar to reward comparisons, reward evaluations can be influenced by shaping functions, thus necessitating techniques such as canonicalization as preprocessing steps to eliminate shaping (Jenner & Gleave, 2022).

Reward shaping is a technique that transforms a base reward function into alternate forms (Ng et al., 1999). This technique is mainly employed in RL for reward design where heuristics and domain knowledge are integrated to accelerate learning (Mataric, 1994; Hu et al., 2020; Cheng et al., 2021; Gupta et al., 2022; Suay et al., 2016). Several applications of reward shaping have been explored, and some notable examples include: training autonomous robots for navigation (Tenorio-Gonzalez et al., 2010); training agents to ride bicycles (Randløv & Alstrøm, 1998); improving agent behavior in multiagent contexts such as the Prisoner's Dilemma (Babes et al., 2008); and scaling RL algorithms in complex games (Lample & Chaplot, 2017; Christiano et al., 2017). Among several reward shaping techniques, potential-based shaping is the most popular due to its preservation of policy invariance, ensuring that the set of optimal policies remains unchanged between different versions of reward functions (Ng et al., 1999; Wiewiora et al., 2003; Gao & Toni, 2015).

## 3 Preliminaries

This section establishes the necessary foundation for understanding the task of direct reward comparisons. We review existing pseudometrics and discuss their limitations, which SARD aims to address.

### 3.1 Markov Decision Processes

A Markov Decision Process (MDP) is defined as a tuple $(\mathcal{S}, \mathcal{A}, \gamma, T, R)$, where $\mathcal{S}$ and $\mathcal{A}$ are the state and action spaces, respectively. The transition model $T : \mathcal{S} \times \mathcal{A} \times \mathcal{S} \to [0, 1]$, dictates the probability distribution of moving from one state, $s \in \mathcal{S}$, to another state, $s' \in \mathcal{S}$, under an action $a \in \mathcal{A}$, and each given transition is specified by the tuple $(s, a, s')$. The discount factor $\gamma \in [0, 1]$ reflects preference for immediate over future rewards. The reward function is denoted by $R : \mathcal{S} \times \mathcal{A} \times \mathcal{S} \to \mathbb{R}$, and for each transition, the reward is denoted by $R(s, a, s')$. Given a batch of sampled transitions from a reward function $R$, we define the the coverage distribution, $\mathcal{D}(s, a, s')$, as the probability distribution over transitions from which the sample is generated. The distributions over $\mathcal{A}$ and $\mathcal{S}$ are denoted by $\mathcal{D}_\mathcal{A}$ and $\mathcal{D}_\mathcal{S}$ respectively; and the corresponding sets of distributions are denoted by $\Delta\mathcal{A}$ and $\Delta\mathcal{S}$ respectively. A trajectory $\tau = \{(s_0, a_0), (s_1, a_1), \cdots, (s_n)\}$, $n \in \mathbb{Z}^+$, is a sequence of states and actions, with a total return: $g(\tau) = \sum_{t=0}^{\infty} \gamma^t R(s_t, a_t, s_{t+1})$. The goal in an MDP is to find a policy $\pi : \mathcal{S} \times \mathcal{A} \to [0, 1]$ (often via RL) that maximizes the expected return $\mathbb{E}[g(\tau)]$. In some situations, the rewards of an MDP are unknown, and IRL can be used to compute the rewards given agent demonstrations (Abbeel & Ng, 2004; Ng & Russell, 2000; Wulfmeier et al., 2015).

### 3.2 Policy Invariance

Policy invariance is a condition where an optimal policy remains unchanged when a reward function is modified typically through shaping (Ng et al., 1999; Jenner et al., 2022). In reward comparisons, policy invariance is a key property to satisfy, since it ensures that equivalent rewards will yield the same optimal policies (Gleave et al., 2020). Formally, any shaped reward can be represented by the additive relationship: $R'(s, a, s') = R(s, a, s') + F(s, a, s')$, where $F(s, a, s')$ is a shaping function. Potential shaping guarantees policy invariance, and it takes the form: $F(s, a, s') = \gamma\phi(s') - \phi(s)$, where $\phi$ is a state potential function. A potentially-shaped reward function, $R'$, is thus represented as:

$$R'(s, a, s') = R(s, a, s') + \gamma\phi(s') - \phi(s). \tag{1}$$

where $R$ is the original reward function. Reward functions $R$ and $R'$ can be deemed equivalent since they yield the same optimal policies.

To effectively compare reward functions that may differ in numerical values but are equivalent since they yield the same optimal policies, the use of *pseudometrics* is highly important. Let $X$ be a set, with $(x, y, z)$ elements of $X$, and let $d : X \times X \to [0, \infty)$ define a pseudometric. This pseudometric adheres to the following axioms: (premetric) $d(x, x) = 0$ for all $x \in X$; (symmetry) $d(x, y) = d(y, x)$ for all $x, y \in X$; and (triangular inequality) $d(x, y) \leq d(x, z) + d(z, y)$ for all $x, y, z \in X$. Unlike a true metric, a pseudometric does not require that: $d(x, y) = 0 \implies x = y$, making it ideal for identifying equivalent reward functions that might have different numerical values.

### 3.3 Equivalent Policy Invariant Comparison (EPIC)

The EPIC pseudometric directly compares reward functions without computing policies, while maintaining policy invariance (Gleave et al., 2020). EPIC's *reward comparison process* involves two steps: first, reward functions are canonicalized into a standard form without shaping; and second, a Pearson distance is computed to differentiate the canonical rewards. The following definitions, by Gleave et al. (2020), describe EPIC.

**Definition 1.** *(Canonically Shaped Reward) Let $R : \mathcal{S} \times \mathcal{A} \times \mathcal{S} \to \mathbb{R}$ be a reward function. Given distributions $\mathcal{D}_{\mathcal{S}} \in \Delta \mathcal{S}$ and $\mathcal{D}_{\mathcal{A}} \in \Delta \mathcal{A}$ over states and actions respectively, let $S$ and $S'$ be random variables distributed as $\mathcal{D}_{\mathcal{S}}$ and $A$ be distributed as $\mathcal{D}_{\mathcal{A}}$. The canonically shaped reward is:*

$$C_{EPIC}(R)(s, a, s') = R(s, a, s') + \mathbb{E}[\gamma R(s', A, S') - R(s, A, S') - \gamma R(S, A, S')]. \tag{2}$$

Canonicalization expresses rewards in form free of shaping. Given a potentially shaped reward, $R'(s, a, s') = R(s, a, s') + \gamma \phi(s') - \phi(s)$, canonicalization yields: $C_{EPIC}(R')(s, a, s') = C_{EPIC}(R)(s, a, s') + \phi_{res}$, where $\phi_{res} = \gamma \mathbb{E}[\phi(S)] - \gamma \mathbb{E}[\phi(S')]$ is the remaining residual potential. EPIC assumes that $S$ and $S'$ are identically distributed such that $\mathbb{E}[\phi(S)] = \mathbb{E}[\phi(S')]$, which results in $\phi_{res} = 0$. This assumption leads to Proposition 1:

**Proposition 1.** *(The Canonically Shaped Reward is Invariant to Shaping) Let $R : \mathcal{S} \times \mathcal{A} \times \mathcal{S} \to \mathbb{R}$ be a reward function, $\mathcal{D}_{\mathcal{S}} \in \Delta \mathcal{S}$ and $\mathcal{D}_{\mathcal{A}} \in \Delta \mathcal{A}$ be distributions over states and actions, $\phi : \mathcal{S} \to \mathbb{R}$ a state potential function, and $R'(s, a, s') = R(s, a, s') + \gamma \phi(s') - \phi(s)$ be the shaped reward. Then: $C_{EPIC}(R) = C_{EPIC}(R')$.*

Proposition 1 means that reward functions $R$ and $R'$, can be compared in their basis form, without the effect of shaping. Finally, the EPIC pseudometric between two reward functions, $R_A$ and $R_B$, is computed as:

$$D_{EPIC}(R_A, R_B) = D_\rho(C_{EPIC}(R_A), C_{EPIC}(R_B)), \tag{3}$$

where:

$$D_\rho(R_A, R_B) = \sqrt{1 - \rho(R_A, R_B)}/\sqrt{2}, \tag{4}$$

is the Pearson distance. $\rho(R_A, R_B)$ is the Pearson correlation between the reward functions, defined as:

$$\rho(R_A, R_B) = \frac{\mathbb{E}[(R_A - \mu_{R_A})(R_B - \mu_{R_B})]}{\sigma_{R_A} \sigma_{R_B}}, \tag{5}$$

where $\mu$ denotes the mean, $\sigma$ denotes the standard deviation, and $\mathbb{E}[(R_A - \mu_{R_A})(R_B - \mu_{R_B})$ is the covariance between $R_A$ and $R_B$. The Pearson distance ensures that EPIC is scale and shift invariant, whilst canonicalization makes EPIC invariant to shaping (Gleave et al., 2020). Computing the exact expectation terms in EPIC is almost impractical because reward values are needed for all transitions. When a reward function has a large (or infinite) state or action space, the sample-based EPIC can be computed as follows:

**Definition 2.** *(Sample-based EPIC) Given transition samples from a coverage distribution $\mathcal{D}$ and a batch $B_M$ of $N_M$ samples from the joint state and action distributions. The canonically shaped reward is:*

$$
\begin{aligned}
C_{EPIC}(R)(s, a, s') \approx R(s, a, s') &+ \frac{\gamma}{N_M} \sum_{(x,u) \in B_M} R(s', u, x) - \frac{1}{N_M} \sum_{(x,u) \in B_M} R(s, u, x) \\
&- \frac{\gamma}{N_M^2} \sum_{(x,\cdot) \in B_M} \sum_{(x',u) \in B_M} R(x, u, x')
\end{aligned}
\tag{6}
$$

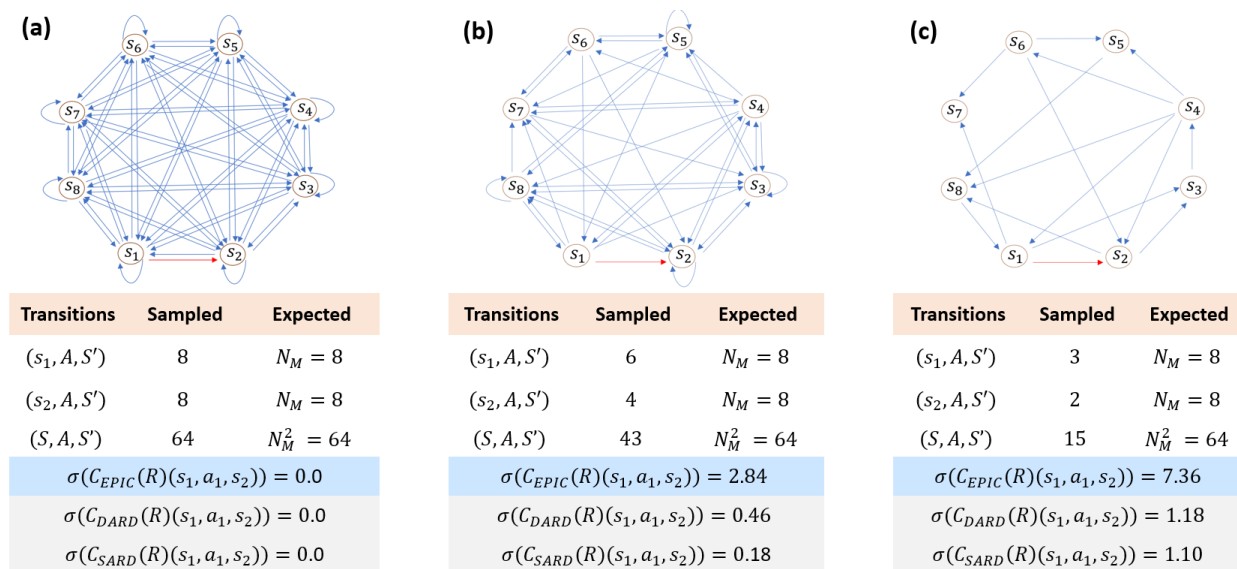

Figure 2: (Impact of unsampled transitions on $C_{EPIC}(R)(s_1, a_1, s_2)$) Sampled transitions are those explored in the reward sample, while expected transitions are those anticipated by EPIC, assuming full coverage. As coverage decreases from (**a**) to (**c**), due to a reduction in the number of sampled transitions, the standard deviation of $C_{EPIC}(R)(s_1, a, s_2)$ increases, indicating EPIC's increased instability to unsampled transitions. For comparison, DARD and SARD have lower standard deviations, signifying higher stability.

## 3.4 Unsampled Transitions

Consider a reward function $R : \mathcal{S} \times \mathcal{A} \times \mathcal{S}$, where $\mathcal{S}$ is the state space and $\mathcal{A}$ is the action space. Rewards are sampled from a coverage distribution, $\mathcal{D}$, and they span a state space $S^{\mathcal{D}} \subseteq \mathcal{S}$ and an action space $A^{\mathcal{D}} \subseteq \mathcal{A}$. We define the following sets of transitions:

**Full Coverage Transitions ($\mathcal{T}^{\mathcal{D}}$)** - The set of all transitions (could be realizable or not) that span the explored state and action space from the reward sample, $\mathcal{T}^{\mathcal{D}} = S^{\mathcal{D}} \times A^{\mathcal{D}} \times S^{\mathcal{D}} \subseteq \mathcal{S} \times \mathcal{A} \times \mathcal{S}$.

**Sampled Transitions ($\mathcal{T}^S$)** - The set of transitions that are actually present in the reward sample. Due to feasibility constraints and limited sampling, this set is a subset of full coverage transitions: $\mathcal{T}^S \subseteq \mathcal{T}^{\mathcal{D}}$.

**Unsampled Transitions ($\mathcal{T}^U$)** - The set of full coverage transitions that are not explored in the reward sample. These transitions can be both realizable and unrealizable, and $\mathcal{T}^U \subseteq \mathcal{T}^{\mathcal{D}} \setminus \mathcal{T}^S$, where $\mathcal{T}^U \subseteq \mathcal{T}^D$.

A major limitation of pseudometrics such as EPIC (and to some extend, DARD), is that they are designed to compare reward functions under high transition coverage where $|\mathcal{T}^S| \approx |\mathcal{T}^{\mathcal{D}}|$. As $|\mathcal{T}^U| \to |T^D|$, the performance of these pseudometrics significantly degrades due to unsampled transitions. To illustrate, consider Equation 6, used to approximate EPIC (Equation 2). To perform the computation, we need to estimate: $\mathbb{E}[R(s', A, S')]$ by dividing the sum of rewards from $s'$ to $S'$ by $N_M$ transitions; $\mathbb{E}[R(s, A, S')]$ by dividing the sum of rewards from $s$ to $S'$ by $N_M$ transitions; and $\mathbb{E}[R(S, A, S')]$ by dividing the sum of all rewards from $S$ to $S'$ by $N_M{}^2$ transitions, where $N_M = |S^{\mathcal{D}} \times A^{\mathcal{D}}|$. Every state $s \in S$ is expected to have $N_M$ transitions to all other states $S'$, which is impractical under transition sparsity when a high number of transitions remain unsampled. Since reward summations are divided by large denominators $N_M$ and $N_M^2$ (see Equation 6), when coverage is low, the number of sampled transitions needed to estimate the reward expectation terms will be fewer than expected, introducing significant error. Moreover, to eliminate residual shaping, $\phi_{res} = \gamma \mathbb{E}[\phi(S)] - \gamma \mathbb{E}[\phi(S')]$, EPIC assumes that $S$ and $S'$ are identically distributed, which favors reward samples with high transition coverage where each state likely has a transition to every other state.

Figure 2 illustrates an example showing the effect of unsampled transitions on computing EPIC across three reward samples spanning a state space $S = S' = \{s_1, ..., s_8\}$, and an action space, $A = \{a_1\}$, such that

$N_M = 8$, under different levels of transition sparsity. Rewards are defined as $R(s_i, a_1, s_j) = 1 + \gamma\phi(s_j) - \phi(s_i)$, where $i, j \in \{1, ..., 8\}$, and state potentials are randomly generated such that: $|\phi(s)| \leq 20$, with $\gamma = 0.5$. The task is to compute $C_{EPIC}(R)(s_1, a_1, s_2)$ over 1000 simulations. For all reward samples, the mean $\mu(C_{EPIC}(R)(s_1, a_1, s_2)) \approx 0$, but the standard deviation, $\sigma(C_{EPIC}(R)(s_1, a_1, s_2))$ varies based on coverage. In Figure 2a, the reward sample has high coverage (100%), hence, the number of observed and expected transitions are equal. In this scenario, EPIC is highly effective and all shaped rewards are mapped to the same value ($\approx 0$), resulting in a standard deviation $\sigma(C_{EPIC}(R)(s_1, a_1, s_2)) = 0$, highlighting consistent reward canonicalization. In Figure 2b, the reward sample has moderate coverage and the fraction of unsampled transitions is approximately 33%. As a result, $\sigma(C_{EPIC}(R)(s_1, a_1, s_2)) = 2.84$ is relatively high, signifying EPIC's sensitivity to unsampled transitions. In Figure 2c, the reward sample exhibits low coverage and the fraction of unsampled transitions is approximately 77%, indicating a significant discrepancy between the number of observed and expected transitions. Consequently, $\sigma(C_{EPIC}(R)(s_1, a_1, s_2)) = 7.36$, highlighting EPIC's increased instability due to unsampled transitions. For comparison, we have added the DARD and SARD estimates, and these distances have lower standard deviations signifying greater stability.

EPIC's limitations have also been acknowledged by Wulfe et al. (2022), who propose DARD, to only consider transitions that are closer to being physically realizable. The DARD pseudometric is given by:

$$C_{DARD}(R)(s, a, s') = R(s, a, s') + \mathbb{E}[\gamma R(s', A, S'') - R(s, A, S') - \gamma R(S', A, S'')],$$

where $A \sim \mathcal{D}_\mathcal{A}$, $S' \sim T(s, A)$, $S'' \sim T(s', A)$, and $T$ is the transition model. DARD is invariant to shaping and generally improves upon EPIC by eliminating the identical distribution assumption and by separating the states connected from $s$ (denoted by $S'$) and $s'$ (denoted by $S''$). However, DARD can suffer from the presence of unsampled transitions since it requires transitions from $S'$ to $S''$, which are not guaranteed to exist. Furthermore, DARD can be highly sensitive to variations in transition distributions between reward functions under comparison (Wulfe et al., 2022), since it relies on transitions that are very close to the reward being canonicalized ($S'$ is close to $s$), and ($S''$ is close to $s'$); and might lack the context of transitions further from $s$ and $s'$. These factors make DARD a modest improvement over EPIC, but it still struggles to compare reward functions under high transition sparsity. This paper uses DARD as an experimental baseline.

## 4  Approach: Sparsity Agnostic Reward Distance (SARD)

Our motivation in SARD is to develop a direct reward comparison technique that minimizes assumptions on the structure and distribution of reward samples, ensuring robustness under high transition sparsity. To derive SARD, we first modify $C_{EPIC}$ (Equation 2) to eradicate the need for high transition coverage. This is achieved by eliminating the assumption that $S$ and $S'$ are identically distributed; and ensuring that reward expectations computed for transitions from $s$, $s'$ and $S$ are different. This approach reduces the impact of unsampled transitions, since reward expectations are computed based on the observed distribution of sampled transitions. With these considerations, the modified canonical equation becomes:

$$C_1(R)(s, a, s') = R(s, a, s') + \mathbb{E}[\gamma R(s', A, S_1) - R(s, A, S_2) - \gamma R(S_3, A, S_4)], \tag{7}$$

where the random variables: $S_1$ and $S_2$ are subsequent states to $s'$ and $s$, respectively; $S_3$ encompasses all initial states for all sampled transitions; and $S_4$ are subsequent states to $S_3$. Applying $C_1$ to a shaped reward $R'(s, a, s') = R(s, a, s') + \gamma\phi(s') - \phi(s)$, we get:

$$C_1(R')(s, a, s') = C_1(R)(s, a, s') + \phi_{res1}, \tag{8}$$

where,

$$\phi_{res1} = \mathbb{E}[\gamma^2\phi(S_1) - \gamma^2\phi(S_4) + \gamma\phi(S_3) - \gamma\phi(S_2)]. \tag{9}$$

$C_1$ is not theoretically robust since its prone to shaping due to residual potential $\phi_{res1}$. To cancel $\mathbb{E}[\phi(S_i)]$, $\forall_i \in \{1, ..., 4\}$, we can add rewards $R(S_i, A, k_i)$ to induce potentials $\gamma\phi(k_i) - \phi(S_i)$; where $k_i$ can be any arbitrary distribution of states. This results in the equation:

$$\begin{aligned} C_2(R)(s, a, s') = {} & R(s, a, s') + \mathbb{E}[\gamma R(s', A, S_1) - R(s, A, S_2) - \gamma R(S_3, A, S_4) \\ & + \gamma^2 R(S_1, A, k_1) - \gamma R(S_2, A, k_2) + \gamma R(S_3, A, k_3) - \gamma^2 R(S_4, A, k_4)]. \end{aligned} \tag{10}$$

Applying $C_2$ to shaped reward $R'(s, a, s') = R(s, a, s') + \gamma\phi(s') - \phi(s)$, we get:

$$C_2(R')(s, a, s') = C_2(R)(s, a, s') + \phi_{res2}, \tag{11}$$

where,

$$\phi_{res2} = \mathbb{E}[\gamma^3\phi(k_1) - \gamma^3\phi(k_4) + \gamma^2\phi(k_3) - \gamma^2\phi(k_2)]. \tag{12}$$

*See Appendix A.5 for derivations of $\phi_{res1}$ and $\phi_{res2}$.*

The canonical form $C_2$ is preferable to $C_1$, since it enables the selection of $k_i$ to eradicate $\phi_{res2}$. A convenient solution is to ensure that: $k_1 = k_4$ and $k_2 = k_3$ such that $\mathbb{E}[\phi(k_1)] = \mathbb{E}[\phi(k_4)]$ and $\mathbb{E}[\phi(k_2)] = \mathbb{E}[\phi(k_3)] \implies \phi_{res2} = 0$. We choose the solution: $k_1 = k_4 = S_5$, and $k_2 = k_3 = S_6$; where $S_5$ are states subsequent to $S_1$, and $S_6$ are states subsequent to $S_2$. This leads to the following SARD definition:

**Definition 3.** *(Sparsity Agnostic Canonically Shaped Reward) Let $R : \mathcal{S} \times \mathcal{A} \times \mathcal{S} \to \mathbb{R}$ be a reward function. Given distributions $\mathcal{D}_\mathcal{S} \in \Delta(\mathcal{S})$ and $\mathcal{D}_\mathcal{A} \in \Delta(\mathcal{A})$, let $S$ be a random variable distributed as $\mathcal{D}_\mathcal{S}$ and $A$ be a random variable distributed as $\mathcal{D}_A$. Given a transition model $T(S'|S, A)$ that governs the conditional distribution from current states to next states. For each $s \in S$ and $s' \in S$, let $\{S_1, ..., S_6\}$ be random variables distributed such that: $S_1$ and $S_2$ are subsequent states to $s'$ and $s$, respectively; $S_3$ encompasses all initial states from all sampled transitions; $S_4$, $S_5$, and $S_6$ are subsequent states to $S_3$, $S_1$ and $S_2$, respectively. The Sparsity Agnostic Canonically Shaped Reward is defined as:*

$$\begin{aligned}C_{SARD}(R)(s, a, s') = R(s, a, s') &+ \mathbb{E}[\gamma R(s', A, S_1) - R(s, A, S_2) - \gamma R(S_3, A, S_4) \\ &+ \gamma^2 R(S_1, A, S_5) - \gamma R(S_2, A, S_6) + \gamma R(S_3, A, S_6) - \gamma^2 R(S_4, A, S_5)].\end{aligned} \tag{13}$$

The SARD canonicalization is also invariant to shaping, as described by Proposition 2.

**Proposition 2.** *(The Sparsity Agnostic Canonically Shaped Reward is Invariant to Shaping) Let $R : \mathcal{S} \times \mathcal{A} \times \mathcal{S}$ be a reward function, $\phi : \mathcal{S} \to R$, a state potential function. Assuming that, for each reward expectation term in $C_{SARD}$, the cross-product for all transitions, $(S_i, A, S_j)$ can be computed, the shaped reward $R'(s, a, s') = R(s, a, s') + \gamma\phi(s') - \phi(s)$ satisfies:*

$$C_{SARD}(R) = C_{SARD}(R').$$

*Proof.* See Appendix A.2.

Given reward functions $R_A$ and $R_B$, the SARD pseudometric is computed as follows:

$$D_{SARD}(R_A, R_B) = D_\rho(C_{SARD}(R_A), C_{SARD}(R_B)), \tag{14}$$

where, $D_p$ is the Pearson distance (see Equation 4).

The term 'agnostic' in $C_{SARD}$ underscores its robustness and flexibility in accommodating variations in transition distributions when canonicalizing reward samples that may lack full coverage but have enough transitions to compute each reward expectation term in Equation 13. However, in many practical scenarios with high transition sparsity, some transitions required for at least one of the reward expectation terms in $C_{SARD}$ might still remain unsampled. In these cases (with missing transitions), although the term 'agnostic' might seem somewhat optimistic, it still reflects SARD's high robustness relative to other methods. This robustness is facilitated by the strategic choices of $k_i$ in $C_{SARD}$ (Equation 13), $S_5$ and $S_6$, ensuring that sample-based approximations for SARD are ideal for the following two reasons: First, for any reward sample, we are guaranteed to compute reliable expectation estimates for the first six terms, since for each set of

transitions ($S_i$ to $S_j$), $S_j$ is created based on $S_i$; hence, these transitions naturally align with any reward sample distribution. However, for transitions in the last two terms, ($S_3$ to $S_6$) and ($S_4$ to $S_5$), $S_6$ and $S_5$, are not directly created based on $S_3$ and $S_4$, as they were previously defined for $S_2$ and $S_1$; therefore, these terms are highly susceptible to unsampled transitions, since they might require transitions that may not be present in the reward sample. Second, while there might be unsampled transitions from ($S_4$ to $S_5$), and ($S_3$ to $S_6$), a minimal set of sampled transitions is likely to exist, because:

- **Transitions** $(S_1, A, S_5) \subseteq (S_4, A, S_5)$.

  Since $S_1$ are subsequent states to $s'$, and $S_4$ are subsequent states for all sampled transitions. It follows that, $S_1 \subseteq S_4$ (See Appendix A.10), hence, $(S_1, A, S_5) \subseteq (S_4, A, S_5)$.

- **Transitions** $(\hat{S}_2, A, S_6) \subseteq (S_3, A, S_6)$.

  $S_2$ is the set of subsequent states from $s$, and it can contain non-terminal and terminal states (usually fewer). Lets denote all non-terminal states in $S_2$ (excludes all terminal) by $\hat{S}_2$, such that the states in $\hat{S}_2$ are initial states for some sampled transitions. Since $S_3$ encompasses all initial states from all sampled transitions, it follows that: $\hat{S}_2 \subseteq S_3$., hence, $(\hat{S}_2, A, S_6) \subseteq (S_3, A, S_6)$.

Therefore, we are likely to get decent reward expectation estimates for the transitions: ($S_3$ to $S_6$) and ($S_4$ to $S_5$). These factors ensure that under significant transition sparsity, the upper bound relative shaping error (see Definition 4) for the approximation of $C_{SARD}$ is lower than that of $C_{EPIC}$ and $C_{DARD}$, as described by Theorem 1.

**Theorem 1.** *Let $R_1$ and $R_2$ be samples of similar reward functions under comparison, each potentially shaped by a different shaping function, but sharing the same set of explored transitions. Under high transition sparsity, where the fraction of sampled transitions is minimal, the upper bound of the Relative Shaping Error (RSE) for the approximation of $C_{SARD}$ is lower than that of $C_{EPIC}$ and $C_{DARD}$.*

*Proof.* See Appendix A.1.

The relative shaping error (RSE) aims to quantify the influence of the residual potentials when computing the Pearson distance during reward comparisons, and it is defined as follows:

**Definition 4.** *(Relative Shaping Error (RSE)) Given an arbitrary reward canonicalization method $C_S$, and a shaped reward sample $R'$ (assumed to be non-zero), canonicalizing $R'$ results in: $C_S(R') = C_S(R) + \phi_r$, where $\phi_r$ represents the residual shaping term. Let $U(C_S(R))$ represent the upper bound of the base canonical reward. We define the relative shaping error as:*

$$RSE(C_S) = \frac{|\phi_r|}{U(C_S(R))}. \tag{15}$$

The denominator, $U(C_S(R))$, represents an upper bound on the magnitude of the base (unshaped) canonical reward, and it serves to normalize the impact of shaping. It is defined as follows:

**Definition 5.** *(Upper Bound Canonical Reward) Let $R$ be an unshaped reward sample and $C_S(R)$ be a canonicalization function represented as the sum of $n$ terms: $C_S(R)(s, a, s') = R(s, a, s') + \mathbb{E}\sum_{i=1}^{n-1}[\alpha_i R(S_i, A, S_j)]$, where $|\alpha| \leq 1$. From all subsets of rewards from $R$, consider the subset that maximizes the absolute expectation $|\mathbb{E}[R(S_x, A, S_y)]|$, where $S_x$ or $S_y$ could also refer to single states. Let this maximum value be denoted as $Z \in \mathbb{Z}^+$. The upper bound of the canonical reward is given by:*

$$U(C_S(R)) = nZ \tag{16}$$

A low RSE value suggests that $U(C_S(R))$ is substantially larger than $|\phi_r|$, indicating that the impact of shaping is likely minimal. Conversely, a high RSE value implies that $U(C_S(R))$ is relatively small compared to $|\phi_r|$, highlighting a more significant influence of shaping. The RSE provides a measure of the worst-case

impact of shaping in extreme scenarios, and it helps to assess the robustness of reward canonicalization methods in mitigating shaping effects.

Computing the exact SARD canonicalization can be challenging for reward samples that might be too large or have unsampled transitions. To approximate SARD under transition sparsity, the sample-based SARD approximation, $\hat{C}_{SARD}$, defined below is used:

**Definition 6.** *(Sample-based SARD) Given transition samples from a coverage distribution $\mathcal{D}$ and a batch $B_M$ of $N_M$ samples from the joint state and action distributions. From $B_M$, we can derive sets $X_i \subseteq B_M$, for $i \in \{1, ..., 6\}$. Each $X_i$ is a set, $\{(x, u)\}$, where $x$ is a state and $u$ is an action. The magnitude, $|X_i|$, is denoted by $N_i$. We define $X_1 = \{(x_1, u)\}$, where $x_1$ denotes subsequent states for transitions starting from $s'$; $X_2 = \{(x_2, u)\}$, where $x_2$ denotes subsequent states for transitions that start from $s$; $X_3 = \{(x_3, u)\}$, where $x_3$ denotes all initial states for all transitions; $X_4 = \{(x_4, u)\}$, where $x_4$ denotes all subsequent states for all transitions; $X_5 = \{(x_5, u)\}$, where $x_5$ denotes subsequent states to $X_1$; $X_6 = \{(x_6, u)\}$, where $x_6$ denotes subsequent states to $X_2$. $C_{SARD}$ can be approximated as:*

$$
\begin{aligned}
\hat{C}_{SARD}(R)(s,a,s') \approx{}& R(s,a,s') + \frac{\gamma}{N_1} \sum_{(x_1,u) \in X_1} R(s',u,x_1) - \frac{1}{N_2} \sum_{(x_2,u) \in X_2} R(s,u,x_2) \\
& - \frac{\gamma}{N_3 N_4} \sum_{(x_3,u) \in X_3} \sum_{(x_4,\cdot) \in X_4} R(x_3,u,x_4) + \frac{\gamma^2}{N_1 N_5} \sum_{(x_1,u) \in X_1} \sum_{(x_5,\cdot) \in X_5} R(x_1,u,x_5) \\
& - \frac{\gamma}{N_2 N_6} \sum_{(x_2,u) \in X_2} \sum_{(x_6,\cdot) \in X_6} R(x_2,u,x_6) + \frac{\gamma}{N_3 N_6} \sum_{(x_3,u) \in X_3} \sum_{(x_6,\cdot) \in X_6} R(x_3,u,x_6) \\
& - \frac{\gamma^2}{N_4 N_5} \sum_{(x_4,u) \in X_4} \sum_{(x_5,\cdot) \in X_5} R(x_4,u,x_5).
\end{aligned}
\tag{17}
$$

**Proposition 3.** *(The sample-based SARD approximation can be invariant to shaping provided all transition definitions are fully covered) Given a potentially-shaped reward sample $R'$ that spans a state space $S^{\mathcal{D}}$ and an action space $A^{\mathcal{D}}$, where $\mathcal{D}$ is the coverage distribution. If the SARD state definitions, $S_i \subseteq S^{\mathcal{D}}$ for all $i \in \{1, ...6\}$, are derived from the sample, and all the necessary transition definitions are fully covered from the sample, then:*

$$\hat{C}_{SARD}(R') = \hat{C}_{SARD}(R)$$

*Proof.* See Appendix A.3 $\qquad\qquad\square$

Each term in the SARD approximation aims to estimate the corresponding term in the exact $C_{SARD}$, Equation 13. For example, $-\frac{\gamma}{N_3 N_4} \sum_{(x_3,u) \in X_3} \sum_{(x_4,\cdot) \in X_4} R(x_3, u, x_4)$ estimates $-\gamma \mathbb{E}[R(S_3, A, S_4)]$ and $\frac{\gamma}{N_1} \sum_{(x_1,u) \in X_1} R(s', u, x_1)$ estimates $\gamma \mathbb{E}[R(s', A, S_1)]$. The structure of the batch, $B_M$ (often sampled uniformly), ensures that we likely consider both underrepresented and overrepresented transitions, to reduce bias in computations due to the transition model. Undefined reward triples (not present in reward sample due to feasibility constraints or limited sampling), $R(x_i, u, x_j)$, are ignored in the reward summation terms. When all the necessary transition terms are present, the sample-based SARD approximation satisfies Proposition 3, as shown in Appendix A.3. $\hat{C}_{SARD}$ can fully canonicalize reward samples with significantly fewer transitions, as long as all the necessary transitions for the reward expectation terms are available. However, this is often not the case, as some transitions might be missing. In rare instances, when a reward sample has full coverage, the three psuedometrics are equivalent as described in Proposition 4:

**Proposition 4.** *The SARD, DARD, and EPIC canonical rewards are similar when a reward sample has full coverage.*

*Proof.* See Appendix A.6. $\qquad\qquad\square$

For the SARD psuedometric, we establish an upper bound for regret in terms of the SARD, showing that when $D_{SARD} \to 0$, the difference between the policies induced by the reward functions under comparison is close to 0, as described by Theorem 2:

**Theorem 1.** *Let $R_A, R_B : S \times A \times S \to \mathbb{R}$ be reward functions with respective optimal policies, $\pi_A^*, \pi_B^*$. Let $D_\pi(t, s_t, a_t, s_{t+1})$ be the distribution over transitions $S \times A \times S$ induced by policy $\pi$ at time $t$, and $D(s, a, s')$ be the coverage distribution. Suppose there exists $K > 0$ such that $KD(s_t, a_t, s_{t+1}) \geq D_\pi(t, s_t, a_t, s_{t+1})$ for all times $t \in \mathbb{N}$, triples $(s_t, a_t, s_{t+1}) \in S \times A \times S$ and policies $\pi \in \{\pi_A^*, \pi_B^*\}$. Then the regret under $R_A$ from executing $\pi_B^*$ instead of $\pi_A^*$ is at most:*

$$G_{R_A}(\pi_A^*) - G_{R_A}(\pi_B^*) \leq 32K\|R_A\|_2(1-\gamma)^{-1}D_{SARD}(R_A, R_B),$$

*where $G_R(\pi)$ is the return of policy $\pi$ under reward $R$.*

*Proof.* See Appendix A.8. $\qquad\square$

In summary, SARD is designed to improve reward comparisons under transition sparsity. To achieve this, SARD eliminates the assumption that $S$ and $S'$ are identically distributed, and also reduces the impact of missing transitions, by canonicalizing rewards solely based on sampled transitions. In terms of computational complexity, the EPIC, DARD and SARD psuedometrics all have a complexity of $O(N_M^2)$ (see Appendix A.9), where $N_M$ is the size of the batch of state-action pairs, $B_M$. Empirically though, computing EPIC and DARD is generally faster than SARD by a constant factor, since SARD has more expectation terms. Lastly, in Appendix A.4, we present a generalized formula for possible SARD extensions.

## 5   Experiments

To evaluate SARD, we examine the following hypotheses:

**H1:** SARD is a reliable reward comparison pseudometric under high transition sparsity.

**H2:** SARD can enhance the task of classifying agent behaviors based on their reward functions.

In these hypotheses, we compare the performance of SARD to both EPIC and DARD using sample-based approximations. In **H1**, we analyze SARD's robustness under transition sparsity resulting from limited sampling and feasibility constraints. In **H2**, we investigate a practical use case to classify agent behaviors using their reward functions. Experiment 1 tests **H1** and Experiment 2 tests **H2**.

**Domain Specifications**   To conduct Experiment 1, we need the capability to vary the number of sampled transitions, since the goal is to test SARD's performance under different levels of transition sparsity. Therefore, Experiment 1 is performed in the Gridworld and Bouncing Balls domains, as they provide the flexibility for parameter variation to control the size of the state and action spaces[2]. These two domains have also been studied in the EPIC and DARD papers, respectively. The Gridworld domain simulates agent movement from a given initial state to a specified terminal state under a static policy. States are defined by $(x, y)$ coordinates where $0 \leq x < N$ and $0 \leq y < M$ implying $|\mathcal{S}| = NM$. The action space consists of four cardinal directions (single steps), and the environment is stochastic, with a probability $\epsilon$ of transitioning to any random state irrespective of the selected action. When $\epsilon = 0$, a feasibility constraint is imposed, preventing the agent from making random transitions. The Bouncing Balls domain, adapted from (Wulfe et al., 2022), simulates a ball's motion from a starting state to a target state while avoiding randomly mobile obstacles. These obstacles add complexity to the environment since the ball might need to change its strategy to avoid obstacles (at a distance, $d = 3$). Each state is defined by the tuple $(x, y, d)$, where $(x, y)$ indicates the ball's current location, and $d$ indicates the ball's Euclidean distance to the nearest obstacle, such that: $0 \leq x < N$, $0 \leq y < M$, and $d \leq max(M, N)$. The action space includes eight directions (cardinals and ordinals), we also define the stochasticity-level parameter $\epsilon$ for choosing random transitions.

For Experiment 2, the objective is to test SARD's performance in near-realistic domain settings, where we have no control over factors such as the nature of rewards and the level of transition sparsity. Therefore, for domain settings, in addition to the Gridworld and the Bouncing Balls domains with the setup similar

---

[2]Experiment 1 excludes the Drone Combat and StarCraft 2 environments because these domains have very large state and action spaces that hinder effective coverage computation.

to Experiment 1 but fixed parameters, we also examine the StarCraft 2 and Drone Combat domains which both simulate battlefield environments where a controlled multiagent team aims to defeat a default AI enemy team (Anurag, 2019; Vinyals et al., 2019). These domains resemble complex scenarios with large state and action spaces (almost infinite for Starcraft2), enabling us to test SARD's (as well as the other pseudometrics) generalization to near-realistic scenarios. Additional details about these domains, including information about the state and action features are described in Appendix C.1.

**Reward Functions**  Extrinsic reward functions are manually defined using a combination of state and action features. For the Starcraft2 and Drone Combat domains, we use the default game engine scores (also based on state and action features), as the reward function (see Appendix B.1). For the Gridworld and Bouncing Balls domains, in each reward function, the reward value $R(s, a, s')$ is derived from the decomposition of state and action features, where, $(s_{f1}, ..., s_{fn})$ is from the starting state $s$; $(a_{f1}, ..., a_{fm})$ is from the action $a$; and $(s'_{f1}, ..., s'_{fn})$ is from the subsequent state $s'$. For the Gridworld domain, these features are the $(x, y)$ coordinates, and for the Bouncing Balls domain, these include $(x, y, d)$, where $d$ is the distance of the obstacle nearest to the ball. For each unique transition, using randomly generated constants: $\{u_1, ..., u_n\}$ for incoming state features; $\{w_1, ..., w_m\}$ for action features; $\{v_1, ...v_n\}$ for subsequent state features, we create polynomial and random rewards as follows:

$$\text{Polynomial:} \quad R(s, a, s') = u_1 s_{f1}^\alpha + \ldots + u_n s_{fn}^\alpha + w_1 a_{f1}^\alpha + \ldots + w_m a_{fm}^\alpha + v_1 s_{f1}'^\alpha + \ldots + v_n s_{fn}'^\alpha,$$

where $\alpha$ is randomly generated from 1-10, denoting the degree of the polynomial.

$$\text{Random:} \quad R(s, a, s') = \beta,$$

where $\beta$ is a randomly generated reward for each unique transition.

For the polynomial rewards, $\alpha$ is the same across the entire sample, but other constants vary between different transitions. The same reward relationships are used to model potential shaping functions. In addition, we also explore linear and sinusoidal reward models as described in Appendix B.1.

For complex environments such as Starcraft2 and the Drone Combat domain, specifying reward functions can be challenging, hence we also incorporate IRL to infer rewards from demonstrated behavior. For our experiments, we consider the following IRL rewards: Maximum Entropy IRL (Maxent) (Ziebart et al., 2008); Adversarial IRL (AIRL) (Fu et al., 2018); and Preferential-Trajectory IRL (PTIRL) (Santos et al., 2021). The full descriptions for these algorithms is described in Appendix C.3.

## 5.1   Experiment 1: Transition Sparsity

**Objective:**   The goal of this experiment is to test SARD's ability to identify similar reward samples under transition sparsity as a result of limited sampling and feasibility constraints.

**Relevance:**   The EPIC pseudometric struggles under high transition sparsity since it is designed to compare reward functions (as well as samples), under high coverage. SARD is developed to overcome EPIC's limitations, and this experiment tests SARD's performance relative to EPIC and DARD, on varying levels of transition coverage due to feasibility constraints and limited sampling.

**Approach:**   This experiment is conducted on a $20 \times 20$ Gridworld domain and a $20 \times 20$ Bouncing Balls domain. For all simulations, manual rewards are used since they enable the flexibility to vary the nature of the relationship between reward values and features, enabling us to test the performance of the pseudometrics on diverse reward values, which include polynomial and random reward relationships. We also vary the shaping potentials such that $|R(s, a, s')| \leq |\gamma\phi(s') - \phi(s)| \leq 5|R(s, a, s')|$.

For each domain, a ground truth reward function ($GT$) and an equivalent potentially shaped reward function ($SH$) are generated, both with full coverage (100%). Using rollouts from a uniform policy, rewards $R$ and $R'$ are sampled from $GT$ and $SH$ respectively, and they might differ in transition composition. After sample generation, $R$ and $R'$ are canonicalized and reward distances are computed using common transitions between them, under varying levels of coverage (to test limited sampling) and feasibility constraints. The SARD, DARD, and EPIC reward distances are computed, as well as DIRECT, which is the Pearson distance of

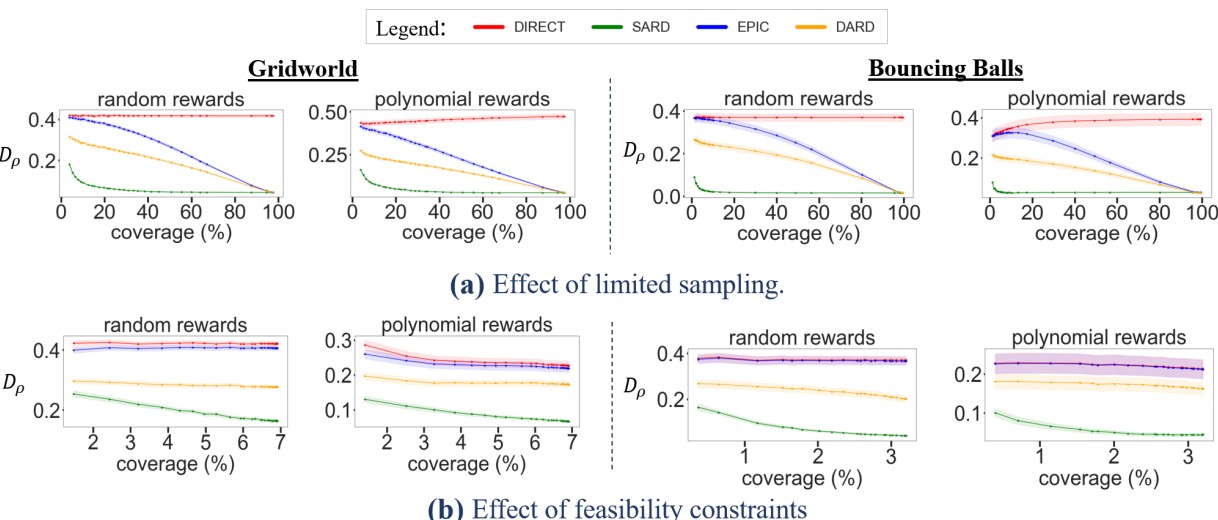

**(a)** Effect of limited sampling.

**(b)** Effect of feasibility constraints

Figure 3: (Transition Sparsity) The performance of reward comparison pseudometrics in identifying the similarity between potentially shaped reward functions under limited sampling and feasibility constraints. For this task, a more accurate pseudometric yields a Pearson distance $D_\rho$ close to 0, indicating policy equivalence between the shaped reward functions, and a less accurate pseudometric results in $D_\rho$ close to 1. In both experiments, transition coverage is calculated as the ratio of sampled transitions to the set of all theoretically possible transitions $|\mathcal{S} \times \mathcal{A} \times \mathcal{S}|$, including both feasible and infeasible transitions. Each coverage data point is averaged over 200 simulations at a constant policy rollout count, with coverage data points generated by varying the number of policy rollouts from 1 to 2000 (see Appendix B.3). In (**a**), EPIC and DARD lag behind SARD when transition coverage is low due to limited sampling from lower rollout counts but gradually improve as transition coverage increases with higher rollout counts. In (**b**), movement restrictions significantly reduce transition coverage, regardless of rollout sampling frequency, which negatively impacts EPIC's performance (almost similar to DIRECT).

rewards without canonicalization. Since $R$ and $R'$ are drawn from equivalent reward functions, an accurate pseudometric should yield distances close to the minimum Pearson distance, $D_\rho = 0$; and the least accurate should yield a distance close to the maximum, $D_\rho = 1$. DIRECT serves as a worst-case performance baseline, since it computes reward distances without canonicalization (needed to remove shaping). We perform 200 simulation trials for each comparison task, and record the mean reward distances.

**Simulations and Results: Limited Sampling:** Using rollouts from a uniform policy, we sample $R$ and $R'$ from $GT$ and $SH$, respectively, under a stochasticity-level parameter, $\epsilon = 0.1$. The number of transitions sampled is controlled by varying the number of policy rollouts (rollout counts) from 1 up to 2000. The corresponding coverage is computed as the number of sampled transitions over the number of all theoretically possible transitions ($= |\mathcal{S} \times \mathcal{A} \times \mathcal{S}|$). Figure 3a summarizes the variation of reward distances to transition coverage due to different levels of transition sampling in the Gridworld, and Bouncing Balls domains. As shown, SARD generally outperforms other baselines since it converges towards $D_\rho = 0$ faster, even when coverage is low. DARD generally outperforms EPIC; however, it's highly prone to shaping compared to SARD since it's more sensitive to unsampled transitions. All pseudometrics generally outperform DIRECT, illustrating the value of removing shaping via canonicalization. No significant difference in the general trends of results are observed between the two domains, and additional simulations are presented in Appendix B. In conclusion, the proposed SARD consistently outperforms both EPIC and DARD, especially under limited sampling when coverage is low.

**Simulations and Results: Feasibility Constraints:** Using rollouts (similar range from 1 to 2000) from a uniform policy, we sample $R$ and $R'$ from $GT$ and $SH$, respectively. To impose feasibility constraints, we set the stochasticity-level parameter, $\epsilon = 0$, to restrict random transitions between states such that

only movement to adjacent states is permitted. Figure 3b summarizes the results for the variation of reward distances to transition coverage under the movement restrictions. As shown, SARD significantly outperforms all the baselines. The movement restriction ensures that coverage is generally low ($< 10\%$), even though the number of rollouts is similar to those in the first experiment. DARD still outperforms EPIC, which performs relatively similar to DIRECT, indicating EPIC's degraded performance under feasibility constraints.

## 5.2 Experiment 2: Classifying Agent Behaviors

**Objective:** The goal of this experiment is to assess SARD's effectiveness as a distance measure in classifying agent behaviors based on their reward functions. If SARD is robust, it should identify similarities among reward functions from the same agents while differentiating reward functions from distinct agents.

**Relevance:** This experiment[3] demonstrates a practical use-case for incorporating reward comparison pseudometrics to interpret reward functions by relating them to agent behavior. In many real-world situations, samples of agent behaviors are available, and there is a need to interpret the characteristics of the agents that produced these behaviors. For example, several works have attempted to predict player rankings and strategies using past game histories (Luo et al., 2020; Liu & Schulte, 2018; Yanai et al., 2022). This experiment takes a similar direction by attempting to classify the identities of agents from their unlabeled past trajectories using reward functions. The reliance on rewards rather than the original trajectories is based on the premise that reward functions are "succinct" and "robust", hence a preferable means to interpret agent behavior (Abbeel & Ng, 2004; Michaud et al., 2020).

**Approach:** In this experiment, we train a $k$-nearest neighbors ($k$-NN) classifier to classify unlabeled agent trajectories by indirectly using computed rewards, to identify the agents that produced these trajectories. We examine the $k$-NN algorithm since it is one of the most popular distance-based classification techniques. The experiment is conducted across all domains, and since we want to maximize classification accuracy, we consider different IRL rewards, including: Maxent, AIRL, PTIRL as well as manual (extrinsic) rewards. For manual rewards, we utilize the default game score for the Starcraft2 and Drone Combat domains, and polynomial rewards for the Gridworld and Bouncing Balls domains, where we induce random potential shaping. For each domain, we examine SARD, DIRECT, EPIC, and DARD as distance measures for a $k$-NN reward classification task. The steps for the approach are as follows:

1. Create agents $X = \{x_1, ..., x_m\}$ with distinct behaviors.

2. For each agent, $x_i \in X$, generate trajectories $\{\tau_1^{x_i}, ..., \tau_p^{x_i}\}$; and compute reward functions $\{R_1^{x_i}, ..., R_p^{x_i}\}$ using IRL or manual specification.

3. Randomly shuffle all the computed reward functions $\mathcal{R}$ (from all agents), and split into the training $\mathcal{R}_{train}$ and testing $\mathcal{R}_{test}$ sets.

4. Use each reward pseudometric as a distance measure to train a $k$-NN classifier with $\mathcal{R}_{train}$ and test it with $\mathcal{R}_{test}$.

In step 1, different agent behaviors are controlled by varying the agents' policies (see Appendix C.2). In step 4, to train the classifier, grid-search is used to identify candidate values for $k$ and $\gamma$, and twofold cross-validation (using $R_{train}$) is used to optimize hyper-parameters based on accuracy. Since we assume potential shaping, the value of $\gamma$ is unknown, hence its a hyper-parameter. To classify an arbitrary reward function $R_i \in R_{test}$, we traverse reward functions $R_j \in R_{train}$, and compute the distance, $D_\rho(R_i, R_j)$ using the reward pseudometrics. We then identify the top $k$-closest rewards to $R_i$, and choose the label of the most frequent class. We select a training to test set ratio is $70 : 30$, and repeat this experiment 200 times.

---

[3]Experiment 2 is related to prior works, Gleave et al. (2020) and Wulfe et al. (2022), where reward distances are computed for the ground truth, regressed, and IRL-generated reward types. Their results show that distances from each reward type are relatively similar, reflecting some form of reward grouping or clustering. However, these works do not explore real-world use-cases of this 'reward grouping' phenomenon, and our work presents a k-NN classification algorithm that utilizes reward distance similarity (between reward sample vectors) to classify agent behaviors, without the need for additional reward preprocessing.

Table 1: The accuracy (%) of different reward comparison distances in $k-$NN reward classification.

| DOMAIN | REWARDS | DIRECT | EPIC | DARD | SARD |
|--------|---------|--------|------|------|------|
| Gridworld | Manual | 69.8 | 69.3 | 70.0 | **75.8** |
| | Maxent | 57.4 | 57.5 | 68.9 | **70.0** |
| | AIRL | 82.3 | 84.9 | 85.0 | **86.2** |
| | PTIRL | 82.2 | 84.2 | 83.4 | **86.0** |
| Bouncing Balls | Manual | 46.5 | 47.3 | 52.0 | **55.2** |
| | Maxent | 39.7 | 46.0 | **50.8** | 49.9 |
| | AIRL | 41.2 | 46.1 | 49.8 | **56.3** |
| | PTIRL | 70.3 | 71.1 | 69.5 | **72.4** |
| Drone Combat | Manual | 67.1 | 67.2 | 66.2 | **73.9** |
| | Maxent | 70.3 | **77.7** | 73.2 | 76.7 |
| | AIRL | 90.1 | 90.7 | 92.3 | **93.8** |
| | PTIRL | 52.5 | 63.7 | 65.1 | **78.3** |
| StarCraft 2 | Manual | 65.5 | 67.4 | 69.5 | **76.5** |
| | Maxent | 72.3 | 74.1 | 73.9 | **74.8** |
| | AIRL | 75.1 | 75.3 | **78.1** | 77.0 |
| | PTIRL | 77.2 | 78.1 | 77.6 | **79.8** |

**Simulations and Results:** Table 1 summarizes experimental results. As shown, SARD generally achieves higher accuracy compared to DIRECT, EPIC and DARD across all domains, indicating SARD effectiveness at discerning similarities between rewards produced by the same agents, and differences between those generated by different agents. This trend is more pronounced with manual rewards where SARD significantly outperforms other baselines. This can be attributed to potential shaping, which is intentionally induced in manual rewards that SARD is specialized to tackle. Therefore, SARD proves to be a more effective distance measure at classifying rewards subjected to potential shaping. For IRL-based rewards such as Maxent, AIRL, and PTIRL, while we assume potential shaping, non-potential shaping could be present. This explains the reduction in SARD's performance gap over EPIC and DARD, as well as the few instances where EPIC and DARD outperform SARD, though SARD is still generally dominant. We also observe that all the psuedometrics tend to perform better on AIRL rewards compared to other IRL-based rewards. This result is likely due to the formulation of the AIRL algorithm, which is designed to effectively mitigate the effects of unwanted shaping in reward approximation (Fu et al., 2018), thus providing more consistent rewards. Overall, SARD, EPIC, and DARD outperform DIRECT, emphasizing the importance of canonicalization at reducing the impact of shaping.

To verify the validity of results, Welch's t-tests for unequal variances are conducted across all domain and reward type combinations, to test the null hypotheses: (1) $\mu_{SARD} \leq \mu_{DIRECT}$, (2) $\mu_{SARD} \leq \mu_{EPIC}$, and (3) $\mu_{SARD} \leq \mu_{DARD}$; against the alternative: (1) $\mu_{SARD} > \mu_{DIRECT}$, (2) $\mu_{SARD} > \mu_{EPIC}$, and (3) $\mu_{SARD} > \mu_{DARD}$, where $\mu$ represents the sample mean. We reject the null when the p-value $< 0.05$ (level of significance), and conclude that: (1) $\mu_{SARD} > \mu_{DIRECT}$ for all instances; (2) $\mu_{SARD} > \mu_{EPIC}$ for 11 out of 12 instances, and (3) $\mu_{SARD} > \mu_{DARD}$ for 10 out of 12 instances. These tests are performed assuming normality as per central limit theorem, since the number of trials is 200. For additional details about the tests and accuracy metrics such as F1-scores, refer to Appendix C. In summary, we conclude that SARD is a more effective distance measure for classifying reward samples compared to its baselines.

# 6 Conclusion and Future Work

This paper introduces SARD, a reward comparison pseudometric designed to address transition sparsity, a significant challenge encountered when comparing reward functions without high transition coverage. Conducted experiments demonstrate SARD's superiority over state-of-the-art pseudometrics, such as EPIC and DARD, under limited sampling and feasibility constraints. Additionally, SARD proves effective as a distance measure for $k$-NN classification using reward functions to represent agent behavior. This implies that SARD

can find higher similarities between reward functions generated by the same agent and higher differences between reward functions that are generated from different agents.

Most existing studies, including ours, have primarily focused on potential shaping, as it is the only additive shaping technique that guarantees policy invariance (Ng et al., 1999; Jenner et al., 2022). Future research should consider the effects of non-potential shaping on SARD (see Appendix B.5) or random perturbations, as these might distort reward functions that would otherwise be similar. This could help to standardize and preprocess a wider range of rewards that might not necessarily be potentially shaped. In computing reward distances, the Pearson distance is employed for its shift and scale invariance properties. However, this distance measure is highly sensitive to outliers, especially in the case of small reward samples. Future work should explore modifications such as Winsorization to mitigate the impact of outliers. Future studies should also explore applications of reward distance comparisons in scaling reward evaluations in IRL algorithms. For example, iterative IRL approaches such as MaxentIRL, often perform policy evaluations to assess the quality of the updated reward in each training trial. Integrating direct reward comparison pseudometrics to determine if rewards are converging, could help to skip the policy evaluation steps, thereby speeding up IRL. Finally, the development of reward comparison metrics has primarily aimed to satisfy policy invariance. A promising area to examine in the future is multicriteria policy invariance, where invariance might be conditioned to different criteria. For example, in the context of reward functions in Large Language Models (LLMs), it might be important to compute reward distance pseudometrics that consider different criteria such as bias, safety, or reasoning, to advance interpretability, which could be beneficial for applications such as reward fine-tuning and evaluation (Lambert et al., 2024).

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

# A Derivations, Theorems and Proofs

## A.1 Relative Shaping Error Comparisons

The process to compare two reward samples, $R_1$ and $R_2$, using a canonicalization method, $C_S$ (e.g., $C_{SARD}$ or $C_{DARD}$), computes the Pearson distance between the canonicalized reward samples,

$$D_\rho(C_S(R_1), C_S(R_2)) = \sqrt{1 - \rho(C_S(R_1), C_S(R_2))}/\sqrt{2} \tag{18}$$

where, $\rho$ is the Pearson correlation. Assuming that the reward samples under comparison are similar in terms of optimal policies induced, but have different values due to shaping potentials $\phi_1$ and $\phi_2$, respectively, and share also share a similar base canonical reward function[4], $C_S(R)$, the Pearson correlation can be expressed as:

$$\rho(C_S(R_1), C_S(R_2)) = \rho(C_S(R) + \phi_{r1}, C_S(R) + \phi_{r2}), \tag{19}$$

where the residual shaping potentials, $\phi_{r1}$ and $\phi_{r2}$, account for the differences between the reward samples.

- In general, when the magnitude of the shaping residuals is significantly less than that of the base canonical reward, $|\phi_{ri}| << |C_S(R)|$ for all $i \in \{1, 2\}$ then $\rho(C_S(R_1), C_S(R_2)) \approx \rho(C_S(R), C_S(R)) = 1$, since the shaping terms have negligible impact. Conversely, when $|\phi_{ri}| >> |C_S(R)|$, then $\rho(C_S(R_1), C_S(R_2)) \approx \rho(\phi_{r1}, \phi_{r2})$, which can reach towards 0, when $\phi_{r1}$ and $\phi_{r2}$ significantly differ.

- Since the base canonical reward $C_S(R)$ is the same across the compared rewards, the Pearson correlation variation can be primarily attributed to the impact of the residual shaping potentials. Therefore, our analysis focuses on quantifying the influence of these residual shaping terms on the overall correlation.

- To estimate the impact of residual shaping potentials, it is essential to consider the magnitude of the base canonical reward, $C_S(R)$. When $|C_S(R)|$ is significantly larger relative to the residual shaping terms $|\phi_{r_i}|$, where $i \in \{1, 2\}$, the influence of the residual potentials diminishes, resulting in a higher Pearson correlation, closer to 1. Conversely, if $|C_S(R)|$ is significantly smaller compared to $|\phi_{r_i}|$, the shaping terms exert a greater impact, leading to a lower Pearson correlation, closer to 0. To quantify this effect, we define the Relative Shaping Error (RSE) as outlined in Definition 4.

**Definition 4.** *(Relative Shaping Error (RSE)) Given an arbitrary reward canonicalization method $C_S$, and a shaped reward sample $R'$ (assumed to be non-zero), canonicalizing $R'$ results in: $C_S(R') = C_S(R) + \phi_r$, where $\phi_r$ represents the residual shaping term. Let $U(C_S(R))$ represent the upper bound of the base canonical reward. We define the relative shaping error as:*

$$RSE(C_S) = \frac{|\phi_r|}{U(C_S(R))}. \tag{20}$$

The denominator, $U(C_S(R))$, represents an upper bound on the magnitude of the base (unshaped) canonical reward, and it serves to normalize the impact of shaping. It is defined as follows:

**Definition 5**. *(Upper Bound Canonical Reward) Let $R$ be an unshaped reward sample and $C_S$, be an arbitrary canonicalization function represented as the sum of $n$ terms such that:*

$$C_S(R) = R(s, a, s') + \mathbb{E} \sum_{i=1}^{n-1} [\alpha_i R(S_i, A, S_j)],$$

---

[4]The base canonical reward $C_S(R)$ of the similar but shaped rewards $R_1$ and $R_2$ could differ in practice when the transitions for the reward samples under comparison are different. However, since the goal of canonicalization is to mitigate the impact of shaping during reward comparisons, for our analysis, we assume that the base canonical rewards are similar, which is guaranteed when the transitions between the two rewards are the same. The only difference between the rewards under comparison becomes the shaping terms, $\phi_1$ and $\phi_2$.

where $|\alpha| \le 1$. *From all subsets of rewards from R, consider the subset that maximizes the absolute expectation* $|\mathbb{E}[R(S_x, A, S_y)]|$, *where $S_x$ or $S_y$ could also refer to single states. Let this maximum value be denoted as* $Z \in \mathbb{Z}^+$. *Then, the upper bound of the canonical reward is given by:*

$$U(C_S(R)) = nZ \tag{21}$$

*Explanation*:

$$
\begin{aligned}
C_S(R) &= R(s, a, s') + \mathbb{E}\sum_{i=1}^{n-1}[\alpha_i R(S_i, A, S_j)] \\
&\le Z + (n-1)Z \\
&\le nZ.
\end{aligned}
$$

This RSE aims to quantify the relative influence of the residual shaping potentials on the Pearson distance normalized by the upper bound of the base (unshaped) canonical rewards (Definition 5). A low RSE value suggests that $U(C_S(R))$ is substantially larger than $|\phi_r|$, indicating that the impact of shaping is likely minimal. Conversely, a high RSE value implies that $U(C_S(R))$ is relatively small compared to $|\phi_r|$, highlighting a more significant influence of shaping. By normalizing with the upper bound, the RSE provides a measure of the worst-case impact of shaping in extreme scenarios. This formulation helps assess the robustness of reward canonicalization methods in mitigating shaping effects during reward comparisons.

**Remark 1:** *(RSE Adjustment due to Constant Reward Expectation Terms):* After canonicalizing a potentially shaped reward sample, $R'$ with a method $C_S$, we get $C_S(R')$. From $C_S(R')$, suppose at least one of the expectation terms is a constant in the form, $\alpha\mathbb{E}[R'(S_i, A, S_j)]$, where, $\alpha$ is a constant, and $\mathbb{E}[R'(S_i, A, S_j)]$ does not depend on the variation of $s$, $a$ or $s'$. Then:

$$C_S(R') = C_S(R) + \phi_r = C_S(\tilde{R}) + k_R + \phi_{\tilde{r}} + k_\phi, \tag{22}$$

where, $k_R = \alpha\mathbb{E}[R(S_i, A, S_j)]$, $k_\phi = \alpha(\gamma\phi(S_j) - \phi(S_i))$ are both constants. The base canonical reward and shaping terms are: $C_S(R) = C_S(\tilde{R}) + k_R$, and $\phi_r = \phi_{\tilde{r}} + k_\phi$. When comparing shaped reward samples, $R_1$ and $R_2$. In computing the Pearson correlation, we have:

$$
\begin{aligned}
\rho(C_S(R_1), C_S(R_2)) &= \rho(C_S(\tilde{R}) + k_{R_1} + \phi_{\tilde{r}1} + k_{\phi_1}, C_S(\tilde{R}) + k_{R_2} + \phi_{\tilde{r}2} + k_{\phi_2}) \\
&= \rho(C_S(\tilde{R}) + \phi_{\tilde{r}1}, C_S(\tilde{R}) + \phi_{\tilde{r}2}) \tag{23}
\end{aligned}
$$

In computing the Pearson correlation (in the process of computing the Pearson distance), we can disregard the presence of the $k$-terms since they are constants and the Pearson correlation is shift invariant.

Relating this to the RSE (Equation 20), note that $|\phi_r|$ primarily serves as a measure of reward variation due to shaping, and $U(C_S(R))$ acts as a normalizing term. In computing $|\phi_r|$, we can disregard $k_\phi$ for the numerator since it won't affect the variation of the Pearson correlation. However, for the denominator $U(C_S(R))$, we cannot disregard $k_R$ because the denominator serves to normalize the residual shaping terms. When $k_R$ is large enough, even though it doesn't affect the pearson correlation, it will lower the impact of $|\phi_r|$ and vice-versa. Therefore, in computing the RSE under $k_R$ and $k_\phi$, the refined RSE becomes:

$$
\begin{aligned}
RSE(C_S) &= \frac{|\phi_r|}{U(C_S(R))} \\
&= \frac{|\phi_{\tilde{r}} + k_\phi|}{U(C_S(\tilde{R}) + k_R)} \\
&= \frac{|\phi_{\tilde{r}}|}{U(C_S(R))} \tag{24}
\end{aligned}
$$

Now for the proof, we will compare the ratios of the upper bound RSE (Equation 24) values for $\hat{C}_{EPIC}$, $\hat{C}_{DARD}$, and $\hat{C}_{SARD}$.

**Theorem 1.** *Let $R_1$ and $R_2$ be samples of similar reward functions under comparison, each potentially shaped by a different shaping function, but sharing the same set of explored transitions. Under high transition sparsity, where the fraction of sampled transitions is minimal, the upper bound of the Relative Shaping Error (RSE) for the approximation of $C_{SARD}$ is lower than that of $C_{EPIC}$ and $C_{DARD}$.*

*Proof.*

Assuming a finite reward sample that spans a state space, $S^{\mathcal{D}}$ and an action space, $A^{\mathcal{D}}$. The following subsets are defined: for $\hat{C}_{EPIC}$, $S \subseteq S^{\mathcal{D}}$ and $S' \subseteq S^{\mathcal{D}}$; for $\hat{C}_{DARD}$, $S'' \subseteq S^{\mathcal{D}}$ and $S' \subseteq S^{\mathcal{D}}$; and for $\hat{C}_{SARD}$, each $S_i \subseteq S^{\mathcal{D}}$, where $i \in \{1, \dots, 6\}$. The shaping components are bounded by $|\mathbb{E}[\phi(S)]| \leq M$, $|\mathbb{E}[\phi(S')]| \leq M$, $|\mathbb{E}[\phi(S'')]| \leq M$, and $|\mathbb{E}[\phi(S_i)]| \leq M$ for all $i \in \{1, \dots, 6\}$, where:

$$M = \max\{\mathbb{E}[|\phi(S_x)|] \mid \text{for all } S_x \subseteq S^{\mathcal{D}}\},$$

and $M \in \mathbb{R}^+$.

**Analysis of EPIC:**

The equation for $C_{EPIC}(R)$ is given by:

$$C_{EPIC}(R)(s, a, s') = R(s, a, s') + \mathbb{E}[\gamma R(s', A, S) - R(s, A, S') - \gamma R(S, A, S')].$$

The term $\mathbb{E}[R(S, A, S')]$ is an expectation over all transitions, where, $S$ contains initial states from all transitions, and $S'$ contains subsequent states from all transitions. Missing transitions can exist in ($s$ to $S'$) and ($s'$ to $S'$) since $S'$ is subsequent to $S$ not $s$ or $s'$. Let the fraction of sampled transitions in $R(s', A, S')$ and $R(s, A, S')$ be given by $u$ and $v$, respectively, where, $0 \leq u, v \leq 1$. Incorporating $u$ and $v$, we get the approximation, $\hat{C}_{EPIC}(R)$ as:

$$\hat{C}_{EPIC}(R)(s, a, s') = R(s, a, s') + \mathbb{E}[u\gamma R(s', A, S') - vR(s, A, S') - \gamma R(S, A, S')], \tag{25}$$

Applying $\hat{C}_{EPIC}$ to a shaped reward $R'(s, a, s')$, we get the residual potential:

$$\phi_{epic} = (\gamma - v\gamma)\phi(s') + (v - 1)\phi(s) + \mathbb{E}[(u\gamma^2 - \gamma^2)\phi(S')] + \mathbb{E}[(\gamma - v\gamma)\phi(S)]$$

For the *best case scenario* when all necessary transitions are available (full coverage), $u, v \approx 1$, and $|\phi_{epic}| \approx 0$, hence, the $RSE(\hat{C}_{EPIC}(R)) \approx 0$, highlighting effective canonicalization. For the *worst case scenario*: $u, v \approx 0$, hence we can eliminate terms with $u, v$ in Equation 25 such that:

$$\hat{C}_{EPIC}(R)(s, a, s') \approx R(s, a, s') - \mathbb{E}[\gamma R(S, A, S')] \tag{26}$$

Applying $\hat{C}_{EPIC}$ (Equation 26) to $R'(s, a, s')$, we obtain the shaping term:

$$\phi_{epic} \approx \gamma\phi(s') - \phi(s) - \mathbb{E}[\gamma^2\phi(S')] + \mathbb{E}[(\gamma\phi(S)] \tag{27}$$

In $\hat{C}_{EPIC}(R')$, the term $-\mathbb{E}[\gamma R'(S, A, S')]$ is a constant since it does not vary with changes in $s$, $a$ or $s'$. Using Remark 1, the shaping component from this constant is $k_\phi = \mathbb{E}[\gamma\phi(S) - \gamma^2\phi(S')]$, and it can be diregrarded when computing the Pearson distance in reward comparisons. Therefore: $\phi_{epic} \approx \gamma\phi(s') - \phi(s) + k_\phi$, and:

$$
\begin{aligned}
|\phi_{\tilde{epic}}| &\approx |\gamma\phi(s') - \phi(s)| && \text{(disregarded } k_\phi\text{)} \\
&\leq |\gamma M + M| && \text{(Since } |\phi(s')| \leq M \text{ and } |\phi(s)| \leq M\text{)} \\
&\leq 2M && \text{(Since } 0 \leq \gamma \leq 1\text{).}
\end{aligned}
$$

Following Definition 5, $U(\hat{C}_{EPIC}(R))$ is given by:

$$U(\hat{C}_{EPIC}(R)) = U(R(s, a, s') - \mathbb{E}[\gamma R(S, A, S')]) = 2Z.$$

Therefore, the upper bound RSE for $\hat{C}_{EPIC}$ is given by:

$$RSE(\hat{C}_{EPIC}(R)) \leq \frac{M}{Z} \tag{28}$$

**Analysis of DARD:**

The equation for $C_{DARD}$ is given by:

$$C_{DARD}(R)(s, a, s') = R(s, a, s') + \mathbb{E}[\gamma R(s', A, S'') - R(s, A, S') - \gamma R(S', A, S'')],$$

where, $S''$ are subsequent states to $s'$ and $S'$ are subsequent states to $s$. For approximations, we can have missing transitions in computing $\mathbb{E}[R(S', A, S'')]$, since $S''$ is naturally subsequent to $s'$, not $S'$. Let the fraction of sampled transitions in $R(S', A, S'')$ be $w$, where, $0 \leq w \leq 1$. Incorporating $w$, we get the approximation, $\hat{C}_{DARD}$, as follows:

$$\hat{C}_{DARD}(R)(s, a, s') = R(s, a, s') + \mathbb{E}[\gamma R(s', A, S'') - R(s, A, S') - w\gamma R(S', A, S'')], \tag{29}$$

Applying $\hat{C}_{DARD}$ (Equation 29) to a shaped reward $R'(s, a, s')$, we get the residual potential:

$$\phi_{dard} = \mathbb{E}[(\gamma^2 - w\gamma^2)\phi(S'') + (w\gamma - \gamma)\phi(S')]$$

For the *best case scenario* without missing transitions, $w \approx 1$, and $|\phi_{dard}| \approx 0$, hence, the $RSE(\hat{C}_{DARD}) \approx 0$, highlighting effective canonicalization. For the *worst case scenario*, $w \approx 0$, hence we can eliminate terms with $w$ in Equation 29. Also, in $\hat{C}_{DARD}$, every reward expectation term varies based on the value of $s$ and $s'$, hence, we do not have shaping constants $k_\phi$ to disregard in computations, such that:

$$\begin{aligned}
|\phi_{d\tilde{a}rd}| &= \left|\mathbb{E}[\gamma^2\phi(S'') - \gamma\phi(S')]\right| \\
&\leq \left|\gamma^2 M + \gamma M\right| \quad \text{(Since } |\phi(S')| \leq M \text{ and } |\phi(S'')| \leq M) \\
&\leq 2M \quad \text{(Since } 0 \leq \gamma \leq 1).
\end{aligned}$$

Following Definition 5, $U(\hat{C}_{DARD}(R))$ is given by:

$$U(\hat{C}_{DARD}(R)) = U(R(s, a, s') + \mathbb{E}[\gamma R(s', A, S'') - R(s, A, S')]) = 3Z.$$

Therefore,

$$RSE(\hat{C}_{DARD}(R)) \leq \frac{2M}{3Z} \tag{30}$$

**Analysis of SARD**

The equation for $C_{SARD}$ is given by:

$$\begin{aligned}
C_{SARD}(R)(s, a, s') = R(s, a, s') &+ \mathbb{E}[\gamma R(s', A, S_1) - R(s, A, S_2) - \gamma R(S_3, A, S_4) + \gamma^2 R(S_1, A, S_5) \\
&- \gamma R(S_2, A, S_6) + \gamma R(S_3, A, S_6) - \gamma^2 R(S_4, A, S_5)],
\end{aligned}$$

where: $S_1$ and $S_2$ are subsequent states to $s'$ and $s$, respectively; $S_3$ encompasses all initial states from all transitions; $S_4$, $S_5$, and $S_6$ are subsequent states to $S_3$, $S_1$ and $S_2$, respectively. As described in Section 4, while there might be unsampled transitions from ($S_4$ to $S_5$), and ($S_3$ to $S_6$), approximations for SARD are robust since a minimal set of sampled transitions is likely to exist, because: transitions $(S_1, A, S_5) \subseteq$

$(S_4, A, S_5)$, and transitions $(\hat{S}_2, A, S_6) \subseteq (S_3, A, S_6)$., where, $\hat{S}_2$ denotes all non-terminal states in $S_2$. For the approximation, $\hat{C}_{SARD}$, let the fraction for unsampled transitions for $(S_4, A, S_5)$ and $(S_3, A, S_6)$ in the sample be $p$ and $q$, respectively, where $0 \leq p, q \leq 1$, such that:

$$
\begin{aligned}
\hat{C}_{SARD}(R)(s,a,s') = R(s,a,s') &+ \mathbb{E}[\gamma R(s', A, S_1) - R(s, A, S_2) - \gamma R(S_3, A, S_4) + \gamma^2 R(S_1, A, S_5) \\
&- \gamma R(S_2, A, S_6) + q\gamma R(S_3, A, S_6) - p\gamma^2 R(S_4, A, S_5)]
\end{aligned}
\tag{31}
$$

Applying $\hat{C}_{SARD}$ (Equation 31) to a shaped reward $R'(s, a, s')$, we get the residual potential:

$$
\phi_{sard} = \mathbb{E}[(\gamma - q\gamma)\phi(S_3) + (p\gamma^2 - \gamma^2)\phi(S_4) + (\gamma^3 - p\gamma^3)\phi(S_5) + (q\gamma^2 - \gamma^2)\phi(S_6)]
$$

In the *best case scenario* without missing transitions, $p, q \approx 1$, and $|\phi_{sard}| \approx 0$, hence, the $RSE(\hat{C}_{SARD}) \approx 0$, highlighting effective canonicalization. For the *worst case scenario*: $p, q \approx 0$, hence we can eliminate terms with the coefficient $p, q$ in Equation 31. Also since the transitions $(S_1, A, S_5) \subseteq (S_4, A, S_5)$, and $(\hat{S}_2, A, S_6) \subseteq (S_3, A, S_6)$, where, $\hat{S}_2$ denotes all non-terminal states in $S_2$. If $p, q \approx 0$, then $\mathbb{E}[R(S_1, A, S_5)] = 0$, and $\mathbb{E}[R(\hat{S}_2, A, S_6)] = 0 \implies \mathbb{E}[R(S_2, A, S_6)] = 0$ (since terminal states in $S_2$ cannot transition to $S_6$). Therefore:

$$
\hat{C}_{SARD}(R)(s,a,s') \approx R(s,a,s') + \mathbb{E}[\gamma R(s', A, S_1) - R(s, A, S_2) - \gamma R(S_3, A, S_4)]
$$

Applying $\hat{C}_{SARD}$ to a shaped reward $R'(s, a, s')$ we get the residual potential:

$$
\phi_{sard} \approx \mathbb{E}[\gamma^2 \phi(S_1) - \gamma^2 \phi(S_4) + \gamma\phi(S_3) - \gamma\phi(S_2)].
$$

In $C_{SARD}(R')$, the term $-\mathbb{E}[\gamma R'(S_3, A, S_4)]$ is a constant since it does not vary due to changes in $s$, $a$ or $s'$. From Remark 1, the resultant shaping component from the constant, $k_\phi = \mathbb{E}[\gamma\phi(S_3) - \gamma^2 \phi_{S_4}]$, can be disregarded when computing the Pearson distance in reward comparisons. Therefore: $\phi_{sard} = \mathbb{E}[\gamma^2 \phi(S_1) - \gamma\phi(S_2)] + k_\phi$, and:

$$
\begin{aligned}
|\phi_{\tilde{sard}}| &= \left|\mathbb{E}[\gamma^2 \phi(S_1) - \gamma\phi(S_2)]\right| && \text{disregarded } k_\phi \\
&\leq \left|\gamma^2 M + \gamma M\right| && (\text{Since } |\phi(S_1)| \leq M \text{ and } |\phi(S_2)| \leq M) \\
&\leq 2M && (\text{Since } 0 \leq \gamma \leq 1).
\end{aligned}
$$

Following Definition 5, the upper bound $U(\hat{C}_{SARD}(R))$ is given by:

$$
U(\hat{C}_{SARD}(R)) = U(R(s,a,s') + \mathbb{E}[\gamma R(s', A, S_1) - R(s, A, S_2) - \gamma R(S_3, A, S_4)]) = 4Z.
$$

Therefore,

$$
RSE(\hat{C}_{SARD}(R)) \leq \frac{M}{2Z}
\tag{32}
$$

**Conclusion**

For the worst case scenario under transition sparsity, we see that:

$$
RSE(\hat{C}_{SARD}(R)) \leq \frac{M}{2Z}
$$

$$
RSE(\hat{C}_{DARD}(R)) \leq \frac{2M}{3Z}
$$

$$
RSE(\hat{C}_{EPIC}(R)) \leq \frac{M}{Z}
$$

Based on the upper bounds on the RSE values, we can conclude that:

$$
RSE(\hat{C}_{SARD}(R)) < RSE(\hat{C}_{DARD}(R)) < RSE(\hat{C}_{EPIC}(R)),
$$

showing that in the worst case scenarios of transition sparsity, $\hat{C}_{SARD}$ is likely more robust than $\hat{C}_{DARD}$ and $\hat{C}_{EPIC}$.

$\square$

## A.2 The Sparsity Agnostic Canonically Shaped Reward is Invariant to Shaping

**Proposition 2:** *(The Sparsity Agnostic Canonically Shaped Reward is Invariant to Shaping) Let $R :$ $\mathcal{S} \times \mathcal{A} \times \mathcal{S}$ be a reward function, $\phi : \mathcal{S} \to R$, a state potential function. Assuming that, for each reward expectation term in $C_{SARD}$, the cross-product for all transitions, $(S_i, A, S_j)$ can be computed, the shaped reward $R'(s, a, s') = R(s, a, s') + \gamma\phi(s') - \phi(s)$ satisfies:*

$$C_{SARD}(R) = C_{SARD}(R').$$

*Proof.*

Let's apply $C_{SARD}$, Definition 3, to a shaped reward $R'(s, a, s')$:

$$
\begin{aligned}
C_{SARD}(R')(s, a, s') = {} & R(s, a, s') + \gamma\phi(s') - \phi(s) + \mathbb{E}[\gamma[R(s', A, S_1) + \gamma\phi(S_1) - \phi(s')] \\
& - [R(s, A, S_2) + \gamma\phi(S_2) - \phi(s)] - \gamma[R(S_3, A, S_4) + \gamma\phi(S_4) - \phi(S_3)] \\
& + \gamma^2[R(S_1, A, S_5) + \gamma\phi(S_5) - \phi(S_1)] - \gamma[R(S_2, A, S_6) + \gamma\phi(S_6) - \phi(S_2)] \\
& + \gamma[R(S_3, A, S_6) + \gamma\phi(S_6) - \phi(S_3)] - \gamma^2[R(S_4, A, S_5) + \gamma\phi(S_5) - \phi(S_4)]],
\end{aligned}
$$

Regrouping the reward terms and the potentials, this reduces to:

$$
\begin{aligned}
C_{SARD}(R')(s, a, s') = {} & C_{SARD}(R)(s, a, s') + (\gamma\phi(s') - \gamma\mathbb{E}[\phi(s')]) + (-\phi(s) + \mathbb{E}[\phi(s)]) \\
& + \mathbb{E}[\gamma^2(\phi(S_1) - \phi(S_1))] + \mathbb{E}[\gamma(-\phi(S_2) + \phi(S_2))] + \mathbb{E}[\gamma(\phi(S_3) - \phi(S_3))] \\
& + \mathbb{E}[\gamma^2(-\phi(S_4) + \phi(S_4))] + \mathbb{E}[\gamma^3(\phi(S_5) - \phi(S_5))] + \mathbb{E}[\gamma^2(-\phi(S_6) + \phi(S_6))]
\end{aligned}
$$

Since $\mathbb{E}[\gamma\phi(s')] = \gamma\phi(s')$ and $\mathbb{E}[\phi(s)] = \phi(s)$, this leads to:

$$C_{SARD}(R')(s, a, s') = C_{SARD}(R)(s, a, s').$$

$\square$

## A.3 Sample-Based Approximation for SARD

**Definition 6** *(Sample-based SARD) Given transition samples from a coverage distribution $\mathcal{D}$ and a batch $B_M$ of $N_M$ samples from the joint state and action distributions. From $B_M$, we can derive sets $X_i \subseteq B_M$, for $i \in \{1, ..., 6\}$. Each $X_i$ is a set, $\{(x, u)\}$, where $x$ is a state and $u$ is an action. The magnitude, $|X_i|$, is denoted by $N_i$. We define $X_1 = \{(x_1, u)\}$, where $x_1$ denotes subsequent states for transitions starting from $s'$; $X_2 = \{(x_2, u)\}$, where $x_2$ denotes subsequent states for transitions that start from $s$; $X_3 = \{(x_3, u)\}$, where $x_3$ denotes all initial states for all transitions; $X_4 = \{(x_4, u)\}$, where $x_4$ denotes all subsequent states for all transitions; $X_5 = \{(x_5, u)\}$, where $x_5$ denotes subsequent states to $X_1$; $X_6 = \{(x_6, u)\}$, where $x_6$ denotes subsequent states to $X_2$. The sample-based approximation, $\hat{C}_{SARD}$ can be computed as:*

$$
\begin{aligned}
\hat{C}_{SARD}(R)(s, a, s') = {} & R(s, a, s') + \frac{\gamma}{N_1} \sum_{(x_1, u) \in X_1} R(s', u, x_1) - \frac{1}{N_2} \sum_{(x_2, u) \in X_2} R(s, u, x_2) \\
& - \frac{\gamma}{N_3 N_4} \sum_{(x_3, u) \in X_3} \sum_{(x_4, \cdot) \in X_4} R(x_3, u, x_4) + \frac{\gamma^2}{N_1 N_5} \sum_{(x_1, u) \in X_1} \sum_{(x_5, .) \in X_5} R(x_1, u, x_5) \\
& - \frac{\gamma}{N_2 N_6} \sum_{(x_2, u) \in X_2} \sum_{(x_6, \cdot) \in X_6} R(x_2, u, x_6) + \frac{\gamma}{N_3 N_6} \sum_{(x_3, u) \in X_3} \sum_{(x_6, \cdot) \in X_6} R(x_3, u, x_6) \\
& - \frac{\gamma^2}{N_4 N_5} \sum_{(x_4, u) \in X_4} \sum_{(x_5, \cdot) \in X_5} R(x_4, u, x_5).
\end{aligned}
\tag{17}
$$

**Lemma 1.** *Let $\phi : S \to \mathbb{R}$ be a potential function defined on a state space $S$ state-based potential function. Given states $x_i \in S$ and $x_j \in S$, then:*

$$\frac{1}{n_1 n_2} \sum_{i=1}^{n_1} \sum_{j=1}^{n_2} (\gamma \phi(x_i) - \phi(x_j)) = \frac{\gamma}{n_1} \sum_{i=1}^{n_1} \phi(x_i) - \frac{1}{n_2} \sum_{j=1}^{n_2} \phi(x_j).$$

*Proof.*

$$\frac{1}{n_1 n_2} \sum_{i=1}^{n_1} \sum_{j=1}^{n_2} (\gamma \phi(x_i) - \phi(x_j)) = \frac{\gamma}{n_1 n_2} \sum_{i=1}^{n_1} \sum_{j=1}^{n_2} \phi(x_i) - \frac{1}{n_1 n_2} \sum_{i=1}^{n_1} \sum_{j=1}^{n_2} \phi(x_j)$$

$$= \frac{\gamma}{n_1 n_2} \sum_{i=1}^{n_1} \left( \sum_{j=1}^{n_2} \phi(x_i) \right) - \frac{1}{n_1 n_2} \sum_{j=1}^{n_2} \left( \sum_{i=1}^{n_1} \phi(x_j) \right)$$

Notice that $\phi(x_i)$ is independent of $j$ and $\phi(x_j)$ is independent of $i$, thus,

$$= \frac{\gamma}{n_1 n_2} \sum_{i=1}^{n_1} n_2 \phi(x_i) - \frac{1}{n_1 n_2} \sum_{j=1}^{n_2} n_1 \phi(x_j)$$

Simplifying terms,

$$= \frac{\gamma}{n_1} \sum_{i=1}^{n_1} \phi(x_i) - \frac{1}{n_2} \sum_{j=1}^{n_2} \phi(x_j).$$

$\square$

**Proposition 3:** *(The sample-based SARD approximation can be invariant to shaping if all transition definitions are fully covered.) Given a potentially-shaped reward sample $R'$ that spans a state space $S^{\mathcal{D}}$ and an action space $A^{\mathcal{D}}$, where $\mathcal{D}$ is the coverage distribution. If the SARD state definitions, $S_i \subseteq S^{\mathcal{D}}$ for all $i \in \{1, ...6\}$, are derived from the sample, and all the necessary transition definitions are fully covered from the sample, then: $\hat{C}_{SARD}(R') = \hat{C}_{SARD}(R)$*

*Proof.*

$$\hat{C}_{SARD}(R')(s,a,s') \approx R(s, a, s') + \gamma \phi(s') - \phi(s)$$
$$+ \frac{\gamma}{N_1} \sum_{(x_1,u) \in X_1} [R(s', u, x_1) + \gamma \phi(x_1) - \phi(s')]$$
$$- \frac{1}{N_2} \sum_{(x_2,u) \in X_2} [R(s, u, x_2) + \gamma \phi(x_2) - \phi(s)]$$
$$- \frac{\gamma}{N_3 N_4} \sum_{(x_3,u) \in X_3} \sum_{(x_4,\cdot) \in X_4} [R(x_3, u, x_4) + \gamma \phi(x_4) - \phi(x_3)]$$
$$+ \frac{\gamma^2}{N_1 N_5} \sum_{(x_1,u) \in X_1} \sum_{(x_5,\cdot) \in X_5} [R(x_1, u, x_5) + \gamma \phi(x_5) - \phi(x_1)]$$
$$- \frac{\gamma}{N_2 N_6} \sum_{(x_2,u) \in X_2} \sum_{(x_6,\cdot) \in X_6} [R(x_2, u, x_6) + \gamma \phi(x_6) - \phi(x_2)]$$
$$+ \frac{\gamma}{N_3 N_6} \sum_{(x_3,u) \in X_3} \sum_{(x_6,\cdot) \in X_6} [R(x_3, u, x_6) + \gamma \phi(x_6) - \phi(x_3)]$$
$$- \frac{\gamma^2}{N_4 N_5} \sum_{(x_4,u) \in X_4} \sum_{(x_5,\cdot) \in X_5} [R(x_4, u, x_5) + \gamma \phi(x_5) - \phi(x_4)].$$

Rearranging the terms, the above equation can be written as:

$$\hat{C}_{SARD}(R')(s,a,s') = \hat{C}_{SARD}(R)(s,a,s') + \phi_{residuals}, \tag{33}$$

where:

$$
\begin{aligned}
\phi_{residuals} = {}& \gamma\phi(s') - \phi(s) + \frac{\gamma}{N_1}\sum_{(x_1,u)\in X_1}[\gamma\phi(x_1)-\phi(s')] - \frac{1}{N_2}\sum_{(x_2,u)\in X_2}[\gamma\phi(x_2)-\phi(s)] \\
& - \frac{\gamma}{N_3N_4}\sum_{(x_3,u)\in X_3}\sum_{(x_4,\cdot)\in X_4}[\gamma\phi(x_4)-\phi(x_3)] + \frac{\gamma^2}{N_1N_5}\sum_{(x_1,u)\in X_1}\sum_{(x_5,\cdot)\in X_5}[\gamma\phi(x_5)-\phi(x_1)] \\
& - \frac{\gamma}{N_2N_6}\sum_{(x_2,u)\in X_2}\sum_{(x_6,\cdot)\in X_6}[\gamma\phi(x_6)-\phi(x_2)] + \frac{\gamma}{N_3N_6}\sum_{(x_3,u)\in X_3}\sum_{(x_6,\cdot)\in X_6}[\gamma\phi(x_6)-\phi(x_3)] \\
& - \frac{\gamma^2}{N_4N_5}\sum_{(x_4,u)\in X_4}\sum_{(x_5,\cdot)\in X_5}[\gamma\phi(x_5)-\phi(x_4)].
\end{aligned} \tag{34}
$$

Applying Lemma 1 to Equation 34, and simplifying terms, we get:

$$
\begin{aligned}
\phi_{residuals} = {}& \gamma\phi(s') - \phi(s) + \frac{\gamma^2}{N_1}\sum_{(x_1,u)\in X_1}[\phi(x_1)] - \gamma\phi(s') - \frac{\gamma}{N_2}\sum_{(x_2,u)\in X_2}[\phi(x_2)] + \phi(s) \\
& - \frac{\gamma^2}{N_4}\sum_{(x_4,u)\in X_4}[\phi(x_4)] + \frac{\gamma}{N_3}\sum_{(x_3,u)\in X_3}[\phi(x_3)] + \frac{\gamma^3}{N_5}\sum_{(x_5,u)\in X_5}[\phi(x_5)] - \frac{\gamma^2}{N_1}\sum_{(x_1,u)\in X_1}[\phi(x_1)] \\
& - \frac{\gamma^2}{N_6}\sum_{(x_6,u)\in X_6}[\phi(x_6)] + \frac{\gamma}{N_2}\sum_{(x_2,u)\in X_2}[\phi(x_2)] + \frac{\gamma^2}{N_6}\sum_{(x_6,u)\in X_6}[\phi(x_6)] - \frac{\gamma}{N_3}\sum_{(x_3,u)\in X_3}[\phi(x_3)] \\
& - \frac{\gamma^3}{N_5}\sum_{(x_5,u)\in X_5}[\phi(x_5)] + \frac{\gamma^2}{N_4}\sum_{(x_4,u)\in X_4}[\phi(x_4)] = 0
\end{aligned}
$$

Therefore,

$$\hat{C}_{SARD}(R') = \hat{C}_{SARD}(R)$$

.

$\square$

## A.4 Generalized SARD Extensions

The following steps result in the generalized formula for potential SARD extensions.

1. To eliminate EPIC's need for full coverage (Section 4), we first create $C_1$ to ensure that rewards are canonicalized based on actual transition sample distributions:

$$C_1(R)(s,a,s') = R(s,a,s') + \mathbb{E}[\gamma R(s',A,S_1) - R(s,A,S_2) - \gamma R(S_3,A,S_4)].$$

$C_1$ yields a residual potential: $\phi_{res1} = \mathbb{E}[\gamma^2\phi(S_1) - \gamma^2\phi(S_4) + \gamma\phi(S_3) - \gamma\phi(S_2)]$.

2. To cancel $\mathbb{E}[\phi(S_i)]$, $\forall_i \in \{1,...,4\}$, we add rewards $R(S_i,A,k_i^1)$ to induce potentials $\gamma\phi(k_i^1) - \phi(S_i)$, which results in $C_2$:

$$
\begin{aligned}
C_2(R)(s,a,s') = {}& R(s,a,s') + \mathbb{E}[\gamma R(s',A,S_1) - R(s,A,S_2) - \gamma R(S_3,A,S_4) \\
& + \gamma^2 R(S_1,A,k_1^1) - \gamma^2 R(S_4,A,k_4^1) + \gamma R(S_3,A,k_3^1) - \gamma R(S_2,A,k_2^1)].
\end{aligned}
$$

$C_2$ yields a residual potential: $\phi_{res2} = \mathbb{E}[\gamma^3\phi(k_1^1) - \gamma^3\phi(k_4^1) + \gamma^2\phi(k_3^1) - \gamma^2\phi(k_2^1)]$.

3. To cancel $\mathbb{E}[\phi(k_i^1)]$, we add rewards $R(k_i^1, A, k_i^2)$ to induce potentials $\gamma\phi(k_i^2) - \phi(k_i^1)$, yielding $C_3$:

$$C_3(R)(s,a,s') = R(s,a,s') + \mathbb{E}[\gamma R(s', A, S_1) - R(s, A, S_2) - \gamma R(S_3, A, S_4)$$
$$+ \gamma^2 R(S_1, A, k_1^1) - \gamma^2 R(S_4, A, k_4^1) + \gamma R(S_3, A, k_3^1) - \gamma R(S_2, A, k_2^1)$$
$$+ \gamma^3 R(k_1^1, A, k_1^2) - \gamma^3 R(k_4^1, A, k_4^2) + \gamma^2 R(k_3^1, A, k_3^2) - \gamma^2 R(k_2^1, A, k_2^2)]$$

$C_3$ yields a residual potential: $\phi_{res3} = \mathbb{E}[\gamma^4\phi(k_1^2) - \gamma^4\phi(k_4^2) + \gamma^3\phi(k_3^2) - \gamma^3\phi(k_2^2)]$.

4. As we can see, this process results in the generalized formula:

$$C_n(R)(s,a,s') = R(s,a,s') + \mathbb{E}[\gamma R(s', A, S_1) - R(s, A, S_2) - \gamma R(S_3, A, S_4)$$
$$+ \gamma^2 R(S_1, A, k_1^1) - \gamma^2 R(S_4, A, k_4^1) + \gamma R(S_3, A, k_3^1) - \gamma R(S_2, A, k_2^1)$$
$$+ \gamma^3 R(k_1^1, A, k_1^2) - \gamma^3 R(k_4^1, A, k_4^2) + \gamma^2 R(k_3^1, A, k_3^2) - \gamma^2 R(k_2^1, A, k_2^2)$$
$$\cdots$$
$$+ \gamma^n R(k_1^{n-2}, A, k_1^{n-1}) - \gamma^n R(k_4^{n-2}, A, k_4^{n-1}) + \gamma^{n-1} R(k_3^{n-2}, A, k_3^{n-1}) - \gamma^{n-1} R(k_2^{n-2}, A, k_2^{n-1})],$$

where, $n \geq 3$. $C_n$ yields a residual potential:

$$\phi_n = \mathbb{E}[\gamma^{n+1}\phi(k_1^{n-1}) - \gamma^{n+1}\phi(k_4^{n-1}) + \gamma^n\phi(k_3^{n-1}) - \gamma^n\phi(k_2^{n-1})].$$

- Looking at $\phi_n$, as $n$ increases, we generally multiply the state distributions, $k_i$ by ($\approx \gamma^n$). Therefore, the upper bound magnitude of $\phi_n$ significantly decreases since $0 \leq \gamma < 1$, and each $|\phi(k_i)| \leq M$, where $M$ is the upper bound potential for all distributions $k_i \subseteq S^{\mathcal{D}}$ (see Appendix A.1). Therefore, as $n$ approaches infinity, $\phi_n$ approaches 0.

- The advantage of the generalized SARD form is that $\phi_n$ approaches 0 as $n$ increases, without any assumptions on the distribution of a reward sample. The challenge, however, is that many $k_i$ terms need to be computed making the process very expensive and difficult to implement. Therefore, a smaller $n$ is preferable. In SARD, we choose $n = 2$, then use our intuition to select sets, $k_i$, which further reduces the residual potential.

## A.5 Residual Potentials

Residual potentials can be defined as the remaining sum of potentials after reward canonicalization.

**Derivation of $\phi_{res1}$:** As shown in Section 4, the equation for $C_1(R)(s,a,s')$ is:

$$C_1(R)(s,a,s') = R(s,a,s') + \mathbb{E}[\gamma R(s', A, S_1) - R(s, A, S_2) - \gamma R(S_3, A, S_4)],$$

Applying $C_1$ to a shaped reward $R'(s,a,s') = R(s,a,s') + \gamma\phi(s') - \phi(s)$:

$$C_1(R')(s,a,s') = R'(s,a,s') + \mathbb{E}[\gamma R'(s', A, S_1) - R'(s, A, S_2) - \gamma R'(S_3, A, S_4)]$$
$$= R(s,a,s') + \gamma\phi(s') - \phi(s) + \mathbb{E}[\gamma(R(s', A, S_1) + \gamma\phi(S_1) - \phi(s'))$$
$$- (R(s, A, S_2) + \gamma\phi(S_2) - \phi(s)) - \gamma(R(S_3, A, S_4) + \gamma\phi(S_4) - \phi(S_3))]$$
$$= C_1(R)(s,a,s') + \mathbb{E}[\gamma^2\phi(S_1) - \gamma^2\phi(S_4) + \gamma\phi(S_3) - \gamma\phi(S_2)]$$

Hence, $C_1(R)(s,a,s')$ yields the residual potential:

$$\phi_{res1} = \mathbb{E}[\gamma^2\phi(S_1) - \gamma^2\phi(S_4) + \gamma\phi(S_3) - \gamma\phi(S_2)].$$

**Derivation of $\phi_{res2}$:**   As shown in Section 4, the equation for $C_2(R)(s, a, s')$ is:

$$C_2(R)(s,a,s') = R(s,a,s') + \mathbb{E}[\gamma R(s', A, S_1) - R(s, A, S_2) - \gamma R(S_3, A, S_4) + \gamma^2 R(S_1, A, k_1)$$
$$- \gamma R(S_2, A, k_2) + \gamma R(S_3, A, k_3) - \gamma^2 R(S_4, A, k_4)].$$

Applying $C_2$ to shaped reward $R'(s, a, s') = R(s, a, s') + \gamma\phi(s') - \phi(s)$:

$$C_2(R')(s,a,s') = R(s,a,s') + \gamma\phi(s') - \phi(s) + \mathbb{E}[\gamma(R(s', A, S_1) + \gamma\phi(S_1) - \phi(s'))$$
$$- (R(s, A, S_2) + \gamma\phi(S_2) - \phi(s)) - \gamma(R(S_3, A, S_4) + \gamma\phi(S_4) - \phi(S_3))$$
$$+ \gamma^2(R(S_1, A, k_1) + \gamma\phi(k_1) - \phi(S_1)) - \gamma(R(S_2, A, k_2) + \gamma\phi(k_2) - \phi(S_2))$$
$$+ \gamma(R(S_3, A, k_3) + \gamma\phi(k_3) - \phi(S_3)) - \gamma^2(R(S_4, A, k_4) + \gamma\phi(k_4) - \phi(S_4))].$$
$$= C_2(R)(s,a,s') + \mathbb{E}[\gamma^3\phi(k_1) - \gamma^3\phi(k_4) + \gamma^2\phi(k_3) - \gamma^2\phi(k_2)]$$

Hence, $C_2(R)(s, a, s')$ yields the residual potential:

$$\phi_{res2} = \mathbb{E}[\gamma^3\phi(k_1) - \gamma^3\phi(k_4) + \gamma^2\phi(k_3) - \gamma^2\phi(k_2)].$$

## A.6  Pseudometric Equivalence Under Full Coverage

**Proposition 4.**   *The SARD, DARD, and EPIC canonical rewards are similar when a reward sample has full coverage.*

*Proof.*

$$C_{EPIC}(R)(s,a,s') = R(s,a,s') + \mathbb{E}[\gamma R(s', A, S') - R(s, A, S') - \gamma R(S, A, S')]$$

$$C_{DARD}(R)(s,a,s') = R(s,a,s') + \mathbb{E}[\gamma R(s', A, S') - R(s, A, S') - \gamma R(S', A, S')]$$

$$C_{SARD}(R)(s,a,s') = R(s,a,s') + \mathbb{E}[\gamma R(s', A, S_1) - R(s, A, S_2) - \gamma R(S_3, A, S_4)$$
$$+ \gamma^2 R(S_1, A, S_5) - \gamma R(S_2, A, S_6) + \gamma R(S_3, A, S_6) - \gamma^2 R(S_4, A, S_5)]$$

Under full coverage, every state $s \in S$, is connected by $A$ actions to every other state $s' \in S$. Thus, $S = S' = S'' = S_1 = S_2 = S_3 = S_4 = S_5 = S_6$, such that:

$$C_{EPIC} = C_{DARD} = C_{SARD} = R(s,a,s') + \mathbb{E}[\gamma R(s', A, S) - R(s, A, S) - \gamma R(S, A, S)]. \qquad (35)$$

$\square$

## A.7  Repeated Canonicalization Under Full Coverage

**Proposition 5.** *The SARD, DARD, or EPIC canonical reward cannot be further canonicalized if the reward sample has full coverage.*

From Proposition 4, under full coverage, Equation 35 states that:

$$C_S = C_{EPIC} = C_{DARD} = C_{SARD} = R(s,a,s') + \mathbb{E}[\gamma R(s', A, S) - R(s, A, S) - \gamma R(S, A, S)].$$

Applying this equation to canonicalize a previously canonical reward we get:

$$C_S[C_S(R)(s, a, s')] = C_S[R(s, a, s') + \mathbb{E}[\gamma R(s', A, S) - R(s, A, S) - \gamma R(S, A, S)]]$$

$$= C_S(R(s, a, s')) + \gamma \mathbb{E}[C_S(R(s', A, S))] - \mathbb{E}[C_S(R(s, A, S))] - \gamma \mathbb{E}[C_S(R(S, A, S))]$$

$$= C_S(R)(s, a, s') + \gamma \mathbb{E}[C_S[R(s', a, S) + \mathbb{E}[\gamma R(S, A, S) - R(s', A, S) - \gamma R(S, A, S)]]]$$
$$- \mathbb{E}[C_S[R(s, a, S) + \mathbb{E}[\gamma R(S, A, S) - R(s, A, S) - \gamma R(S, A, S)]]]$$
$$- \gamma \mathbb{E}[C_S[R(S, a, S) + \mathbb{E}[\gamma R(S, A, S) - R(S, A, S) - \gamma R(S, A, S)]]]$$

$$= C_S(R)(s, a, s') + \gamma \mathbb{E}[C_S[(R(s', A, S) - \mathbb{E}[R(s', A, S)]) + \mathbb{E}[\gamma R(S, A, S) - \gamma R(S, A, S)]]]$$
$$- \mathbb{E}[C_S[(R(s, A, S) - \mathbb{E}[R(s, A, S)]) + \mathbb{E}[\gamma R(S, A, S) - \gamma R(S, A, S)]]]$$
$$- \gamma \mathbb{E}[C_S[R(S, a, S) + \mathbb{E}[\gamma R(S, A, S) - R(S, A, S) - \gamma R(S, A, S)]]]$$

After explicit cancellations, the above equation reduces to:

$$C_S[C_S(R)(s, a, s')] = C_S(R)(s, a, s').$$

## A.8 Regret Bound

In this section, we establish a regret bound in terms of the SARD distance. The procedure for the analysis is adapted from related work on EPIC by Gleave et al. (2020).

Given reward functions $R_A$ and $R_B$ and their optimal policies $\pi_A^*$ and $\pi_B^*$, we show that the regret of using policy $\pi_B^*$ instead of a policy $\pi_A^*$ is bounded by a function of $D_{SARD}(R_A, R_B)$. We also show that as the regret tends to be 0 suggesting that $\pi_A^* \approx \pi_B^*$, the distance, $D_{SARD}(R_A, R_B) \to 0$. The concept of regret bounds is important as it shows that differences in $D_{SARD}$ reflect differences between optimal policies induced by the input rewards.

For our analysis, we will use the following Lemmas:

**Lemma 2.** *Let $f$ be a one-dimensional vector of real numbers and $f_i \subseteq f$. Then:*

$$||f_i||_2 \leq ||f||_2 \tag{36}$$

*Proof.* Suppose $f$ has $n$ elements and $f_i$ has $k$ elements. Since $f_i \subseteq f$, every element in $f_i$ is also in $f$, and $k \leq n$. Therefore, $\sum f^2 \geq \sum f_i^2$ (Euclidean distance always positive) such that: $||f_i||_2 \leq ||f||_2$. $\square$

**Lemma 3.** *Let $R_A, R_B : S \times A \times S \to \mathbb{R}$ be reward functions with corresponding optimal policies $\pi_A^*$ and $\pi_B^*$. Let $D_\pi(t, s_t, a_t, s_{t+1})$ denote the distribution over trajectories that policy $\pi$ induces at time step $t$. Let $D(s, a, s')$ be the coverage distribution over transitions $S \times A \times S$. Suppose that there exists some $K > 0$ such that $KD(s_t, a_t, s_{t+1}) \geq D(t, s_t, a_t, s_{t+1})$ for all time steps $t \in \mathbb{N}$, triples $s_t, a_t, s_{t+1} \in S \times A \times S$ and policies $\pi \in \{\pi_A^*, \pi_B^*\}$. Then the regret under $R_A$ from executing $\pi_B^*$ optimal for $R_B$ instead of $\pi_A^*$ is at most:*

$$G_{R_A}(\pi_A^*) - G_{R_A}(\pi_B^*) \leq \frac{2K}{1 - \gamma} D_{L_1, D}(R_A, R_B).$$

*where: $D_{L_1, D}$ is either a metric or pseudometric in $L_1$ space, and $G_R(\pi)$ resembles the return of $R$ under a policy $\pi$.*

*Proof.* See Gleave et al. (2020) $\square$

**Lemma 4.** *Let $R_A, R_B : S \times A \times S \to \mathbb{R}$ be reward functions. Let $\pi_A^*$ and $\pi_B^*$ be policies optimal for rewards $R_A$ and $R_B$. Suppose the regret under the standardized reward $R_A^{SARD}$ from executing $\pi_B^*$ instead of $\pi_A^*$ is upper bounded by some $U \in \mathbb{R}$:*

$$G_{R_A^{SARD}}(\pi_A^*) - G_{R_A^{SARD}}(\pi_B^*) \leq U. \tag{37}$$

*Then the regret under the original reward $R_A$ is bounded by:*

$$G_{R_A}(\pi_A^*) - G_{R_A}(\pi_B^*) \leq 8U\|R_A\|_2. \tag{38}$$

*Proof.* Let the standardized reward can be represented as:

$$R^{SARD} = \frac{C_{SARD}(R)}{\|C_{SARD}(R)\|_2}, \tag{39}$$

It follows that:

$$G_{R^{SARD}}(\pi) = \frac{1}{\|C_{SARD}(R)\|_2} G_{C_{SARD}(R)}(\pi) = \frac{1}{\|C_{SARD}(R)\|_2}(G_R(\pi) - \mathbb{E}_{s_0 \sim d_0}[\Phi(s_0)]), \tag{40}$$

where, $s_0$ depends only on the initial state distribution $d_0$, but not $\pi$. Applying Equation 40 to $\pi_A^*$ and $\pi_B^*$:

$$G_{R^{SARD}}(\pi_A^*) - G_{R^{SARD}}(\pi_B^*) = \frac{1}{\|C_{SARD}(R_A)\|_2}(G_{R_A}(\pi_A^*) - G_{R_A}(\pi_B^*)). \tag{41}$$

Combining Equation 41 and 37:

$$G_{R_A}(\pi_A^*) - G_{R_A}(\pi_B^*) \leq U\|C_{SARD}(R_A)\|_2. \tag{42}$$

We now bound $\|C_{SARD}(R_A)\|_2$ in terms of $\|R_A\|_2$. The SARD canonical reward is expressed as:

$$C_{SARD}(R)(s,a,s') = R(s,a,s') + \mathbb{E}[\gamma R(s', A, S_1) - R(s, A, S_2) - \gamma R(S_3, A, S_4)$$
$$+ \gamma^2 R(S_1, A, S_5) - \gamma R(S_2, A, S_6) + \gamma R(S_3, A, S_6) - \gamma^2 R(S_4, A, S_5)]$$

Now, using the triangular equality rule on the $L_2$ distance, and linearity of expectation:

$$\|C_{SARD}(R)(s,a,s')\|_2 \leq \|R(s,a,s')\|_2 + \mathbb{E}[\gamma\|R(s', A, S_1)\|_2 + \| - R(s, A, S_2)\|_2 + \gamma\| - R(S_3, A, S_4)\|_2$$
$$+ \gamma^2\|R(S_1, A, S_5)\|_2 + \gamma\| - R(S_2, A, S_6)\|_2 + \gamma\|R(S_3, A, S_6)\|_2 + \gamma^2\| - R(S_4, A, S_5)\|_2]$$

Using Lemma 2, the $L_2$ norm of each reward subspace is such that:

$$\|R(S_i, A_j, S_k)\|_2 \leq \|R(S, A, S')\|_2 = \|R\|_2. \tag{43}$$

therefore,

$$\|C_{SARD}(R)(s,a,s')\|_2 \leq 8\|R\|_2 \tag{44}$$

Combining Equation 44 and 42 we get:

$$G_{R_A}(\pi_A^*) - G_{R_A}(\pi_B^*) \leq 8U\|R\|_2.$$

$\square$

**Theorem 2.** *Let $R_A, R_B : S \times A \times S \to \mathbb{R}$ be reward functions with respective optimal policies, $\pi_A^*, \pi_B^*$. Let $D_\pi(t, s_t, a_t, s_{t+1})$ be the distribution over transitions $S \times A \times S$ induced by policy $\pi$ at time $t$, and $D(s, a, s')$ be the coverage distribution. Suppose there exists $K > 0$ such that $KD(s_t, a_t, s_{t+1}) \geq D_\pi(t, s_t, a_t, s_{t+1})$ for all times $t \in \mathbb{N}$, triples $(s_t, a_t, s_{t+1}) \in S \times A \times S$ and policies $\pi \in \{\pi_A^*, \pi_B^*\}$. Then the regret under $R_A$ from executing $\pi_B^*$ instead of $\pi_A^*$ is at most:*

$$G_{R_A}(\pi_A^*) - G_{R_A}(\pi_B^*) \leq 32K\|R_A\|_2(1-\gamma)^{-1}D_{SARD}(R_A, R_B),$$

*where $G_R(\pi)$ is the return of policy $\pi$ under reward $R$.*

*Proof.* Adapting Gleave et al. (2020) [A.4], we can write:

$$D_{\text{SARD}}(R_A, R_B) = \frac{1}{2} \left\| R_A^{SARD}(S, A, S') - R_B^{SARD}(S, A, S') \right\|_2^2. \tag{45}$$

such that, when considering the $L_1$ distance:

$$D_{L_1,D}(R_A^{SARD}, R_B^{SARD}) = \left\| R_A^{SARD}(S, A, S') - R_B^{SARD}(S, A, S') \right\|_1 \leq 2D_{\text{SARD}}(R_A, R_B). \tag{46}$$

Combining Lemma 3 and Equation 46:

$$G_{R_A^{SARD}}(\pi_A^*) - G_{R_A^{SARD}}(\pi_B^*) \leq \frac{2K}{1-\gamma} D_{L_1,D}(R_A^{SARD}, R_B^{SARD}) \leq \frac{4K}{1-\gamma} D_{\text{SARD}}(R_A, R_B). \tag{47}$$

Applying Lemma 4, we get:

$$G_{R_A}(\pi_A^*) - G_{R_A}(\pi_B^*) \leq \frac{32K\|R_A\|_2}{1-\gamma} D_{\text{SARD}}(R_A, R_B). \tag{48}$$

$\square$

As shown, when $D_{SARD} \to 0$, the regret goes towards 0.

## A.9 Computational Considerations

Given a batch $B_M$ of $N_M$ samples from the joint state and action distribution. the computational complexity of all the presented psuedometrics are approximately $O(N_M^2)$.

**EPIC Complexity:**

$$C_{EPIC}(R)(s, a, s') = R(s, a, s') + \mathbb{E}[\gamma R(s', A, S') - R(s, A, S') - \gamma R(S, A, S')].$$

The most expensive computation for EPIC is calculating $\mathbb{E}[R(S, A, S')]$, which takes approximately $O(N_M^2)$ complexity, since we iterate $B_M$ in two loops to perform the computation. Other computations, $\mathbb{E}[R(s, A, S')]$ and $\mathbb{E}[R(s', A, S')]$ take $O(N_M)$ complexity, since we iterate $B_M$ in a single loop. Hence, the overall complexity is approximately $O(N_M^2)$.

**DARD Complexity:**

$$C_{DARD}(R)(s, a, s') = R(s, a, s') + \mathbb{E}[\gamma R(s', A, S'') - R(s, A, S') - \gamma R(S', A, S'')].$$

The most expensive computation for DARD is calculating $\mathbb{E}[R(S', A, S'')]$, which takes approximately $O(N_M^2)$ complexity, since we iterate $B_M$ in two loops to perform the computation. Other computations, $\mathbb{E}[R(s', A, S'')]$ and $\mathbb{E}[R(s, A, S')]$ take $O(N_M)$ complexity, since we iterate $B_M$ in a single loop. Hence, the overall complexity is approximately $O(N_M^2)$.

**SARD Complexity:**

$$\begin{aligned} C_{SARD}(R)(s, a, s') = R(s, a, s') + \mathbb{E}[&\gamma R(s', A, S_1) - R(s, A, S_2) - \gamma R(S_3, A, S_4) \\ &+ \gamma^2 R(S_1, A, S_5) - \gamma R(S_2, A, S_6) + \gamma R(S_3, A, S_6) - \gamma^2 R(S_4, A, S_5)]. \end{aligned}$$

For the terms $\mathbb{E}[R(s', A, S_1)]$ and $\mathbb{E}[R(s', A, S_1)]$, the computational complexity is $O(N_M)$ since we iterate $B_M$ in a single loop. For each of the terms, $\mathbb{E}[R(S_3, A, S_4)]$, $\mathbb{E}[R(S_1, A, S_5)]$, $\mathbb{E}[R(S_2, A, S_6)]$, $\mathbb{E}[R(S_3, A, S_6)]$, $\mathbb{E}[R(S_4, A, S_5)]$, the computational complexity is $O(N_M^2)$, since we iterate $B_M$ in two loops. Hence, the overall complexity is approximately $5 * O(N_M^2)$, which is asymptotically $O(N_M^2)$.

### A.10 SARD State Definitions

Figure 4 is a graph showing transitions in a reward sample with 10 states $S^{\mathcal{D}} = \{x_0, ..., x_9\}$, and a single action $A^{\mathcal{D}} = \{a_1\}$ between state transitions. The goal here is to illustrate an example showing how states $\{S_1, ..., S_6\}$ are defined in SARD, as well as the state relationships: $(S_1 \subseteq S_4)$ and $(\hat{S}_2 \subseteq S_3)$, which make SARD robust to missing transitions.

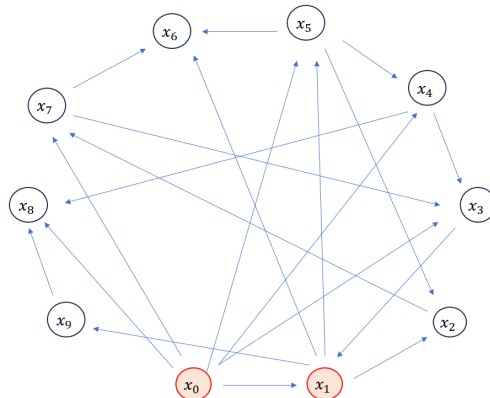

Figure 4: A transition graph with 10 states $\{x_0, ...x_9\}$, and a single action $\{a_1\}$. State subsets are defined based on the transition: $(x_0, a_1, x_1)$.

The Sparsity Agnostic Canonical Reward is given by:

$$C_{SARD}(R)(s,a,s') = R(s,a,s') + \mathbb{E}[\gamma R(s', A, S_1) - R(s, A, S_2) - \gamma R(S_3, A, S_4) + \gamma^2 R(S_1, A, S_5)$$
$$- \gamma R(S_2, A, S_6) + \gamma R(S_3, A, S_6) - \gamma^2 R(S_4, A, S_5)],$$

where: $S_1$ and $S_2$ are subsequent states to $s'$ and $s$, respectively; $S_3$ encompasses all initial states from all transitions; $S_4$, $S_5$, and $S_6$ are subsequent states to $S_3$, $S_1$ and $S_2$, respectively.

**Following the SARD definition, the states in Figure 4, are defined as follows:**

$s$: state $x_0$.

$s'$: state $x_1$.

$S_1$ (subsequent to $s'$): $\{x_2, x_5, x_6, x_9\}$.

$S_2$ (subsequent to $s$): $\{x_1, x_3, x_4, x_5, x_7, x_8\}$. $x_8$ is a terminal state

$S_3$ (initial states from all transitions): $\{x_0, x_1, x_2, x_3, x_4, x_5, x_7, x_9\}$. terminal states $x_6$ and $x_8$ not included

$S_4$ (subsequent states to $S_3$): $\{x_1, x_2, x_3, x_4, x_5, x_6, x_7, x_8, x_9\}$. starting state $x_0$ not included

$S_5$ (subsequent states to $S_1$): $\{x_2, x_4, x_6, x_7, x_8\}$

$S_6$ (subsequent states to $S_2$): $\{x_1, x_2, x_3, x_4, x_5, x_6, x_8, x_9\}$

**Transition Relationships (See Section 4 for Reference)**

**1.** $S_1 \subseteq S_4$, therefore, $(S_1, A, S_5) \subseteq (S_4, A, S_5)$.

**2.** Let $\hat{S}_2 = S_2 \setminus$ terminal states. excludes terminal state $x_8$

  $\hat{S}_2 \subseteq S_3$, therefore, $(\hat{S}_2, A, S_6 \subseteq S_3, A, S_6)$.

# B  Experiment 1: Additional Analysis

## B.1  Reward Functions

Extrinsic reward values are manually defined using a combination of state and action features. For the Starcraft2 and Drone Combat domains, we use the default game score (also based on state and action features), as the reward values. For the Gridworld and Bouncing Balls domains, in each reward function, the reward value, $R(s, a, s')$; is derived from the decomposition of state and action features, where, $(s_{f1}, ..., s_{fn})$ is from the starting state, $s$; $(a_{f1}, ..., a_{fm})$ is from the action, $a$; and $(s'_{f1}, ..., s'_{fn})$ is from the subsequent state, $s'$. For the Gridworld domain, these features are the $(x, y)$ coordinates, and for the Bouncing Balls domain, these include $(x, y, d)$, where $d$ is the distance of the obstacle nearest to the ball. Using the following randomly generated constants: $\{u_1, ..., u_n\}$ for incoming state features; $\{w_1, ..., w_m\}$ for action features; $\{v_1, ...v_n\}$ for subsequent state features; we created the following reward models:

- Linear:

$$R(s, a, s') = u_1 s_{f1} + ... + u_n s_{fn} + w_1 a_{f1} + ... + w_m a_{fm} + v_1 s'_{f1} + ... + v_n s'_{fn},$$

- Polynomial:

$$R(s, a, s') = u_1 s_{f1}^{\alpha} + ... + u_n s_{fn}^{\alpha} + w_1 a_{f1}^{\alpha} + ... + w_m a_{fm}^{\alpha} + v_1 s'^{\alpha}_{f1} + ... + v_n s'^{\alpha}_{fn},$$

  where, $\alpha$ is randomly generated from $1 - 10$, denoting the degree of the polynomial.

- Sinusoidal:

$$R(s, a, s') = u_1 sin(s_{f1}) + \cdots + u_n sin(s_{fn}) + w_1 sin(a_{f1}) + \cdots + w_m sin(a_{fm}) \\ + v_1 sin(s'_{f1}) + \cdots + v_n sin(s'_{fn})$$

- Random

$$R(s, a, s') = \beta,$$

  where, $\beta$ is a randomly generated reward for each given transition.

The same relationships are used to model potential functions, where: $\phi(s) = f(s_{f1}, .., s_{fn})$, and $f$ is the relationship drawn from the set: {polynomial, sinusoidal, linear, random}. For the Starcraft 2 and Drone Combat domains, we used the default game score provided the game engine, as the reward function. For the Starcraft2 domain, this score[5] focuses on the composition of unit and resource features as well as actions within the domain. Since the Drone Combat environment is originally designed for a predator-prey domain, we adapt the score[6] to essentially work the same for the Drone Combat scene (i.e instead of a predator being rewarded for eating some prey, the reward is now an ally attacking an enemy).

---

[5]https://steemit.com/steemstem/@cpufronz/building-a-bot-for-starcraft-ii-2-the-starcraft-ii-environment
[6]https://github.com/koulanurag/ma-gym/blob/master/ma$_g$ym/envs/

## B.2 Transition Sparsity Experiment

Algorithm 1 summarizes the pseudocode for Experiment 1 to examine transition sparsity. To test the effect of limited sampling, we run the algorithm with different number of rollouts (rollout count), dictated by the array $T$. To test the effect of feasibility constraints, we run Algorithm 1 but with imposed movement restrictions by setting $\epsilon = 0$.

---

**Algorithm 1** Analyzing the effect of limited sampling on reward distance

---

**Input**:

    $T$ - list of policy rollout counts,

    $E$ - number of experimental trials under same condition,

    $G$ - grid size,

    $RD$ - list to store reward distances at different coverages,

    $MC$ - maximum coverage $\approx S \times A \times S$.

**Output**: $RD$

 1: generate $GT$ - ground truth reward, $SH$ - shaped reward from all possible transitions.

 2: **for** *rollout_count* in $T$ **do**

 3:    *trial_distance*, *trial_coverage* = list(), list()

 4:    **for** *trial* in $E$ **do**

 5:       $B_{gt}, B_{sh} = set(), set()$

 6:       generate trajectories $\tau_{gt}$ and $\tau_{sh}$ using uniform policy rollouts.

 7:       **for** $(s, a, s') \in \tau_{gt}$ **do**

 8:          $B_{gt}.\text{add}(s, a, s')$

 9:       **end for**

10:       **for** $(s, a, s') \in \tau_{sh}$ **do**

11:          $B_{sh}.\text{add}(s, a, s')$

12:       **end for**

13:       for $(s, a, s') \in B_{gt}$, and $(s, a, s') \in B_{sh}$ retrieve $R(s, a, s')$ using $GT$ and $R'(s, a, s')$ using $SH$, respectively.

14:       $coverage = |B_{gt} \cup B_{sh}|/MC$

15:       compute $dist(R, R')$ using EPIC, SARD, DARD, or other comparison metrics.

16:       *trial_distance*.append($dist(R, R')$)

17:       *trial_coverage*.append(*coverage*)

18:    **end for**

19:    $RD$.append([mean(*trial_coverage*), mean(*trial_distance*)])

20: **end for**

---

## B.3 Experimental Parameters

A uniform policy in the Gridworld domain, would randomly select one of the four actions, {north, east, south, west}, at each timestep. For the Bouncing Balls domain, it would select {north, north-east, east, east-south, south, south-west, west, west-north, north}, randomly. The parameter $\epsilon$ dictates the ratio of times in which random transitions (instead of uniform policy), are executed. Table 2 and Table 3 shows the experimental parameters used to run Experiment 1 (Algorithm 1).

Table 2: Low Coverage: Parameters used to test the variation of coverage for the Gridworld and the Bouncing Balls domain.

| Parameter | Values |
|---|---|
| Rollout Counts, $T$ | $[1, 2, 3, 4, 5, 6, 7, 8, 9, 10, 15, 20, 30, 40, 50, 75, 100, 200, 300, 400, 500, 1000, 2000]$ |
| Epochs, $E$ | 200 |
| Policy, $\pi$ | uniform, $\epsilon = 0.1$ |
| Discount, $\gamma$ | 0.7 |
| Dimensions | $20 \times 20$ |

Table 3: Feasibility Constraints: Parameters used to test the variation of coverage in the presence of movement restrictions, $\epsilon = 0$.

| Parameter | Values |
|---|---|
| Rollout Counts, $T$ | $[1, 2, 3, 4, 5, 6, 7, 8, 9, 10, 15, 20, 30, 40, 50, 75, 100, 200, 300, 400, 500, 1000, 2000]$ |
| Epochs, $E$ | 200 |
| Policy, $\pi$ | uniform, $\epsilon = 0$ |
| Discount, $\gamma$ | 0.7 |
| Dimensions | $20 \times 20$ |

### B.4 Transition Sparsity: Additional Results

**Parameter Variation**   In both the Gridworld and the Bouncing Balls domains, we did not we did not see much difference in the structure of results between the $10 \times 10$ domain and the $20 \times 20$ domains. Results were fairly consistent in that SARD tends to outperform DARD and EPIC, and feasibility constraints tend to limit coverage significantly.

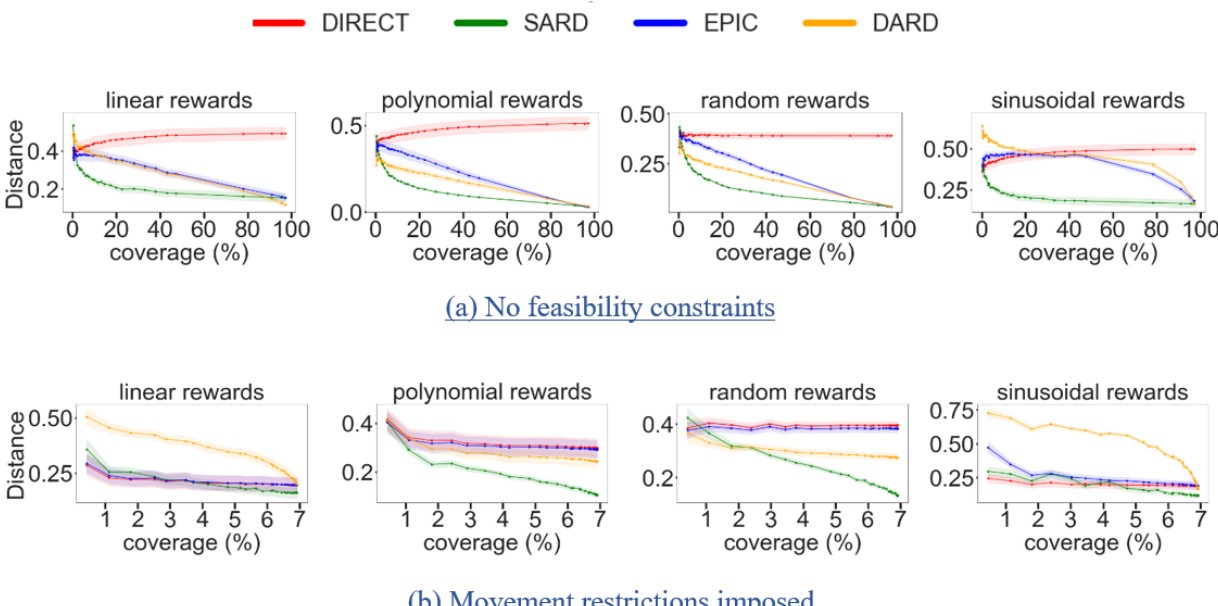

Figure 5: $10 \times 10$ Gridworld: Variation of reward relationships

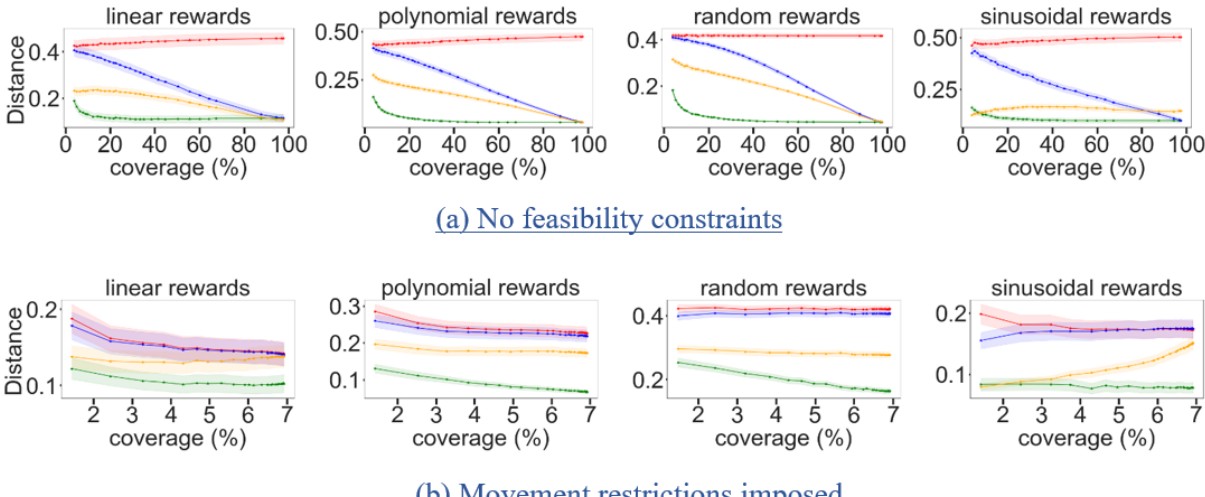

Figure 6: 20 ×20 Gridworld: Variation of reward relationships

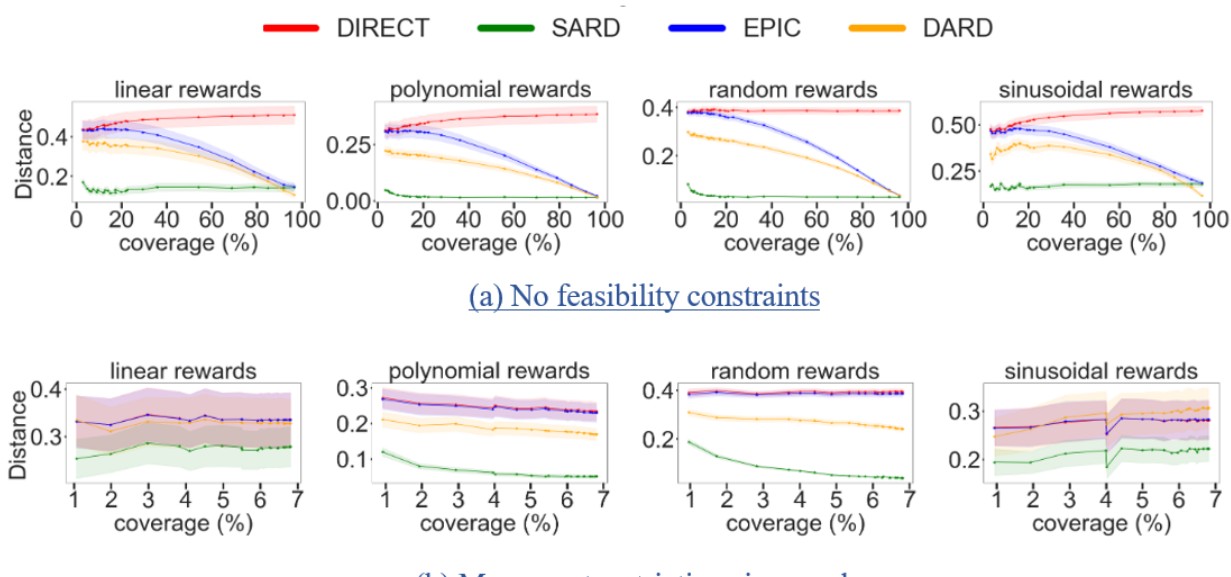

Figure 7: 10 × 10 Bouncing Balls: Variation of reward relationships

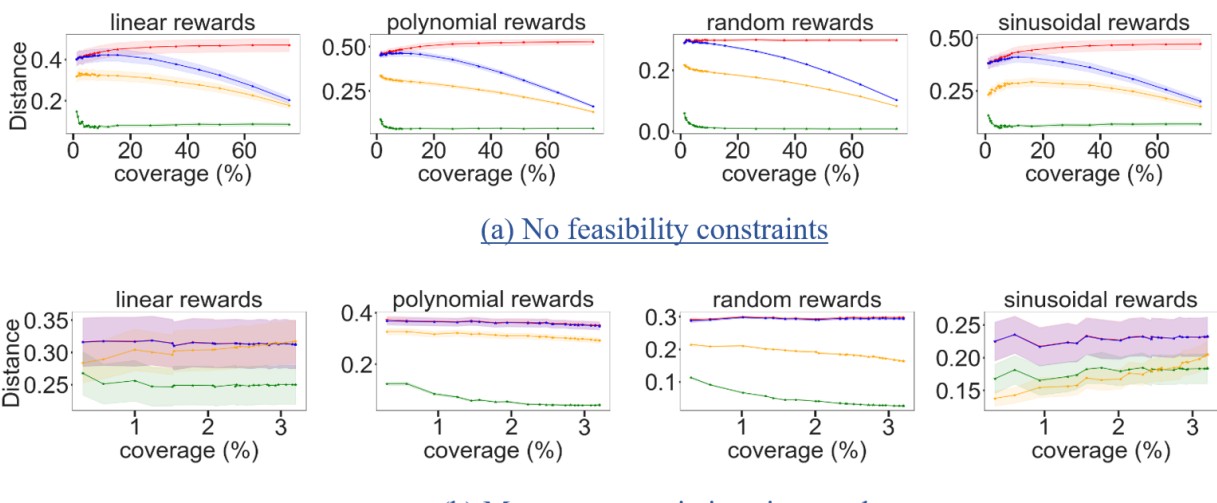

Figure 8: $20 \times 20$ Bouncing Balls: Variation of reward relationships

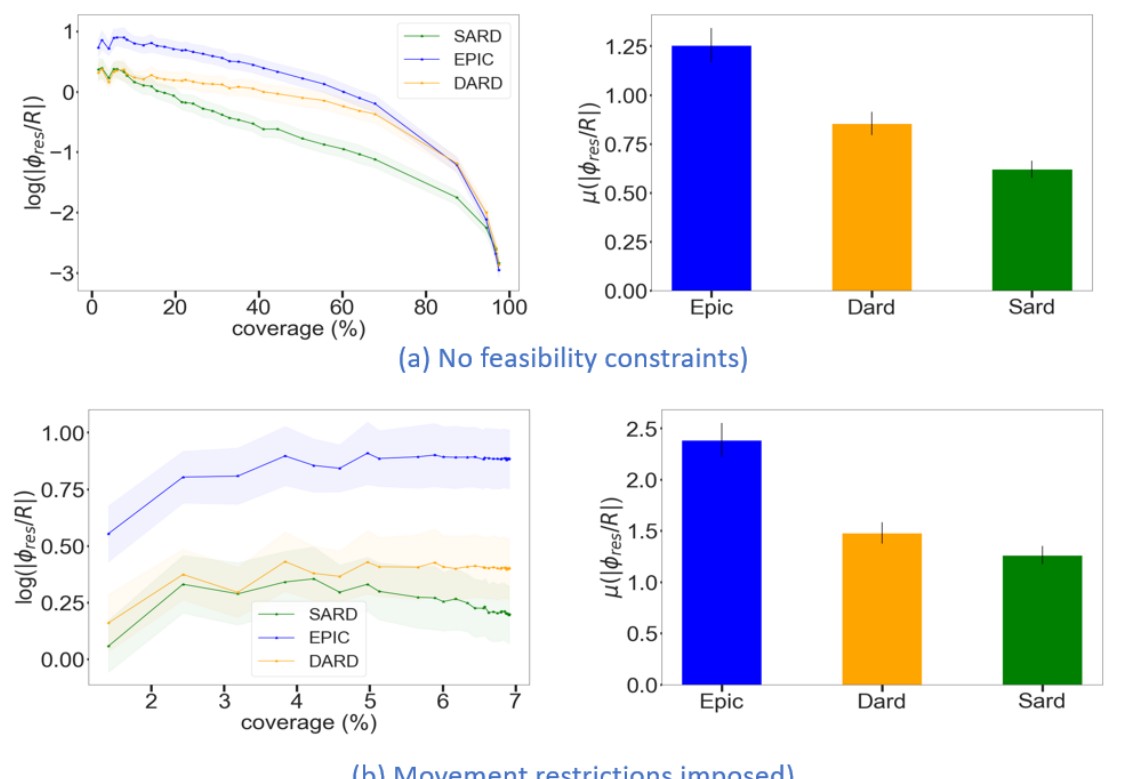

Figure 9: Here we compare the ratio of the residual potentials to the actual rewards for reward samples under comparison, on a $20 \times 20$ Bouncing Balls domain. As shown, SARD is less prone to residual shaping compared to DARD and EPIC.

## B.5 Deviations from Potential Shaping

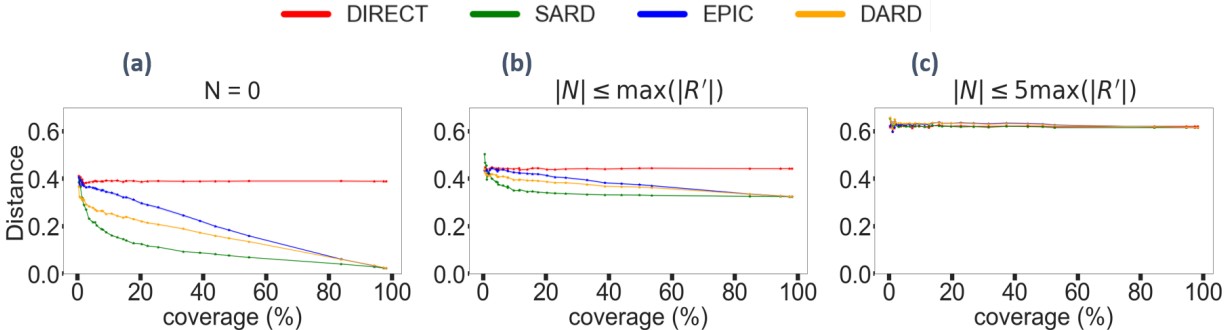

Figure 10: Non-Potential Shaping Effect: As the severity of randomly generated noise increases, rewards deviate more from potential shaping, hence all the pseudometrics degrade in performance. In the end, the pseudometrics perform to the level of DIRECT, showing that canonicalization does not yield any additional advantages at these levels.

Most reward comparison pseudometrics are designed with the goal of canonicalizing rewards that differ due to potential shaping. In this experiment, we examine how deviations from non-potential shaping can affect the performance of these pseudometrics. Within a $20 \times 20$ Gridworld domain, we generate a ground truth ($GT$) polynomial reward function and a corresponding shaped reward function ($SH$), both with full coverage (100% transitions). From $GT$ and $SH$, we sample rewards $R$ and $R'$ using uniform policy rollovers. For both samples, we add additional noise, $N$, with the following severity levels: **None:** $N = 0$, **Mild:** $|N| \leq \max(|R'|)$, and **High:** $|N| \leq 5\max(|R'|)$, where $N$ is randomly generated from a uniform distribution within bounds defined by these severity levels. Thus, the updated shaped reward is given by:

$$R'' = R' + N.$$

Figure 10 shows the performance variation of the reward comparison pseudometrics to the severity levels, dictated by $N$. As shown, when $N = 0$ (noise free), the difference between SARD, EPIC, DARD, and DIRECT is the greatest, with a performance order: SARD > DARD > EPIC > DIRECT, which demonstrates SARD's performance advantage over other pseudometrics under potential shaping. As the impact of $N$ increases, the shaped reward, $R''$, becomes almost entirely comprised of noise, and the shaping component significantly deviates from potential shaping. As shown, SARD's performance gap over other pseudometrics significantly diminishes. At high severity (Figure 10c), SARD's performance is nearly identical to all other pseudometrics, including DIRECT, which does not involve any canonicalization. In conclusion, these results still demonstrate SARD's superiority in canonicalizing rewards even with minor deviations from potential shaping (Figure 10 b). However, as the rewards become non-potentially shaped, all pseudometrics generally become ineffective, performing similarly to non-canonicalized techniques such as DIRECT.

## C  Experiment 2: Additional Analysis

### C.1  Reward Classification: Testbeds and IRL

**Gridworld:**  The Gridworld domain simulates agent movement from a given initial state to a specified terminal state under a static policy. Each state is defined by an $(x, y)$ coordinate where $0 \leq x < N$, and $0 \leq y < M$ implying $|\mathcal{S}| = NM$. For Experiment 2, the action space only consists of four cardinal directions {north, east, south, west}, and to define classes, we use static policies based on the action-selection distribution (out of 100) per state. Table 1 shows the Gridworld parameters used for Experiment 2.

**Bouncing Balls:**   The Bouncing Balls domain, adapted from (Wulfe et al., 2022), simulates a ball's motion from a starting state to a target state while avoiding randomly mobile obstacles. These obstacles

add complexity to the environment since the ball might need to change its strategy to avoid obstacles (at a distance, $d = 3$). Each state is defined by the tuple $(x, y, d)$, where $(x, y)$ indicates the ball's current location, and $d$ indicates the ball's Euclidean distance to the nearest obstacle, such that: $0 \leq x < N$, $0 \leq y < M$, and $d \leq max(M, N)$. The action space includes eight directions (cardinals and ordinals), with the stochastic parameter $\epsilon$ for choosing random transitions. Table 2 describes the parameters for Experiment 2.

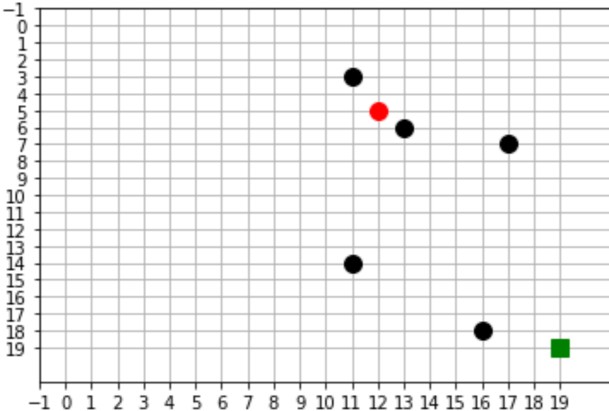

Figure 11: Bouncing Balls domain: The red ball starts at a randomly assigned state and aims to reach the green state while avoiding being close to the black obstacles. The presence of the obstacles makes the domain more complex than the simple Gridworld.

**Drone Combat:** The Drone Combat domain is derived from the multi-agent environment, which simulates a predator-prey interaction (Anurag, 2019). We adapt this testbed to simulate a battle between two drone swarms; a blue swarm denoting the ally team; and a red swarm denoting a default AI enemy. The goal is for the blue ally team to defeat the default AI team. This testbed offers discrete actions and states within a fully observable environment, while also offering flexibility for unit creation, and obstacle placement. However, the number of states and actions is still high such that we didn't use it in Experiment 1. Each unit (blue and red squares) possesses a distinct set of parameters and possible actions. Each team consists of drones and ships, and the team that wins either destroys the entire drones of the opponent or its ship. This ship adds complexity to the decision-making process of the teams which need to engage with the enemy, as well as safeguard their ships. Each drone is defined by the following attributes: visibility range (VR) - the range a unit can see from its current position (partial observability); health (H) - the number of firings a unit can sustain; movement range(MR) - the maximum distance that a unit can move to; and shot strength(SS) - the probability of a shot hitting its target. All these attributes are drawn from the set:

$$U = \{(VR, H, MR, SS) \mid VR \in \{1, 3, 5\}, H \in \{5, 10, 15\}, MR \in \{1, 2, 3\}, SS \in \{0.05, 0.1\}\}$$

Table 5 summarizes the parameters used for Experiment 2.

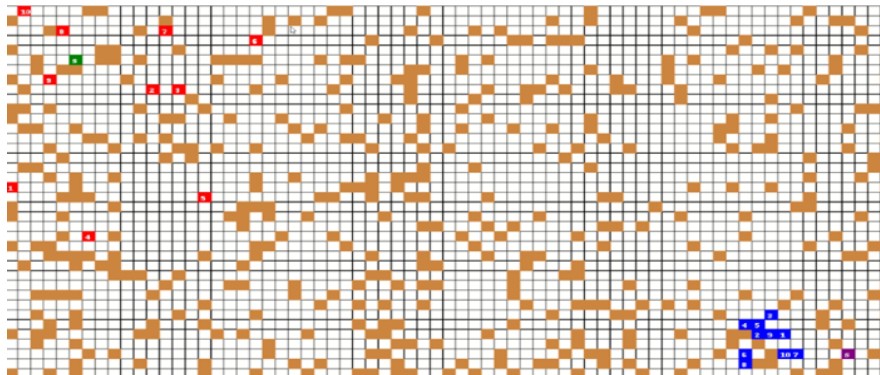

Figure 12: Drone Combat: The drone combat domain describes a battlefield scene where the blue team aims to attack the red AI team. The brown squares present movement obstacles; the green square represents the enemy's ship; and the purple square represents the team's ship.

**Starcraft 2 (SC2):** The SC2 domain is a strategy game created by Blizzard Entertainment that features real-time actions on a complex battlefield environment. The game involves planning, strategy, and quick decision-making to control a multi-agent ally team, aiming to defeat a default AI team in a competitive, challenging, and time-sensitive environment. SC2 serves not only as entertainment but also as a platform for professional player competitions. Due to its complexity and the availability of commonly used interactive Python libraries, the SC2 game is widely employed in Reinforcement Learning, serving as a testbed for multi-agent research. The goal of the ally team is to gather resources, and build attacking units that are used to execute a strategic mission to defeat the AI enemy; within an uncharted map, that gets revealed after extensive exploration (introduces partial observability). The sheer size of the map and the multitude of possible actions for each type of unit, as well as the number of units, contribute to the enormity of the action and state spaces. During combat, each ally unit, moves in a decentralized manner and attacks an enemy unit using an assigned cooperative strategy from the set: $C = \{c_1, c_2, c_3, c_4\}$; where $c_1$ - move towards ally closest to enemy's start base; $c_2$ - move towards a random enemy unit; $c_3$ - move towards ally closest to an enemy unit; and $c_4$ - move towards the map's center. We focus on attack-oriented ally units to reduce the state space. Non-attacking units such as pylons are treated as part of the environment. The game state records the number of ally units ($num_{ally}$), and the total number of opponent units ($num_{enemy}$); as well as the central coordinates of the ally and the enemy. The action records the number of ally units attacking the enemy at an instance. Table 6 describes the Starcraft 2 parameters used for Experiment 2.

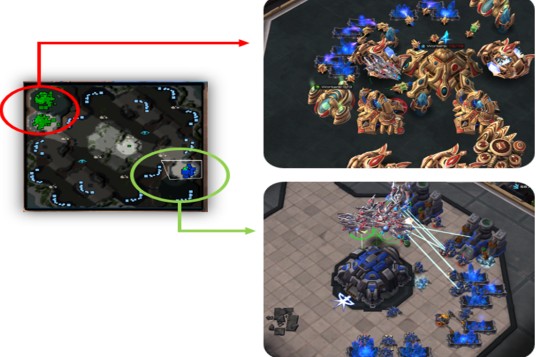

Figure 13: (Starcraft 2): The domain describes a multiagent team that aims to defeat a default AI enemy. In this figure, the the red section denotes the team's base where it builds resources and attacking units. In the green section, it shows the enemy's base, where the team is attacking the enemy.

## C.2 Testbed Parameters:

In all these domains, the optimal values for $\gamma$ (discount factor) and $k$ (the neighborhood size) are not fixed for each independent trial. Therefore, for hyperparameter selection, we employ a grid search over the set defined by:

$$\{(\gamma, k) : \gamma \in \{0, 0.1, \ldots, 1\}, k \in \{5, 10, 15, \ldots, 100\}\}$$

The agent classes shown describe the policy that an agent takes in each given state. For example, in the Gridworld domain, an agent with a policy [25, 25, 25, 25], randomly selects the cardinal direction to take from a uniform distribution. For the Drone Combat and Starcraft 2 domains, the agent behaves based on the combination of attributes defined (refer to Appendix C.1).

Table 4: Bouncing Balls parameters for Experiment 2.

| Parameter | Values |
|---|---|
| Agent fixed policies (10 classes) | [[12, 12, 12, 12, 13, 13, 13, 13], [5, 5, 25, 25, 25, 5, 5, 5], [25, 25, 25, 5, 5, 5, 5, 5], [5, 5, 5, 5, 5, 25, 25, 25], [5, 5, 65, 5, 5, 5, 5, 5], [5, 5, 5, 65, 5, 5, 5, 5], [5, 5, 5, 5, 65, 5, 5, 5], [5, 25, 5, 25, 5, 25, 5, 5], [20, 5, 20, 5, 20, 5, 20, 5], [5, 20, 5, 20, 5, 20, 5, 20]] |
| Trajectory sets per policy | 100 |
| Number of obstacles | 5 |
| Distance to deviate from obstacle (Manhattan) | 3 |
| Number of comparison trials | 200 |
| Actions | move: {north, north-east, east, east-south, south, south-west, west, west-north} |
| Dimensions | $20 \times 20$ |

Table 5: Gridworld parameters for Experiment 2.

| Parameter | Values |
|---|---|
| Agent fixed policies (10 classes), | [[25, 25, 25, 25], [5, 5, 5, 85], [85, 5, 5, 5], [5, 85, 5, 5], [5, 5, 85, 5], [5, 15, 30, 55], [55, 30, 15, 5], [15, 5, 55, 30], [5, 55, 30, 15], [15, 30, 5, 55]] |
| Trajectory sets per policy, | 100 |
| Number of comparison trials, | 200 |
| Actions, | move: {north, west, south, east} |
| Dimensions | $20 \times 20$  ( 400 states) |

Table 6: Drone Combat parameters for Experiment 2.

| Parameter | Values |
| --- | --- |
| Agent policies | 10 classes, each consisting of 5 agents. Each agent $x$ has attributes randomly drawn from the set: $U = \{(\text{VR, H, MR, SS}) \mid VR \in \{1, 3, 5\}, H \in \{5, 10, 15\}, MR \in \{1, 2, 3\}, SS \in \{0.05, 0.1\}\}$ |
| Trajectory sets per policy | 100 |
| Number of agents per team | 11 (1 ship, 10 drones) |
| Number of comparison trials | 200 |
| Actions, $\alpha$ denotes movement range, $1 \leq \alpha \leq 3$ | $\{\{\text{left}^\alpha, \text{up}^\alpha, \text{right}^\alpha, \text{down}^\alpha\}, \text{attack}\}$ |
| Dimensions | $40 \times 25$, with obstacles occupying $\approx 30\%$ of the area |

Table 7: Starcraft 2 parameters for Experiment 2.

| Parameter | Values |
| --- | --- |
| Agent policies (description in Starcraft 2 domain), generated based on resources and strategy | 10 classes, agents attributes randomly chosen from: $U = \{(c, \text{u}) \mid c \in \{c_1, c_2, c_3, c_4\}, \text{u} \in \{\text{adept, voidray, phoenix, stalker}\}\}.$ |
| Trajectory sets per policy | 100 |
| Comparison trials | 200 |
| Actions | Number of attacking units per unit time |
| State representation | $(num_{ally}, num_{enemy}, (x_{ally}, y_{ally}), (x_{enemy}, y_{enemy}))$ |

### C.3 Inverse Reinforcement Learning (IRL)

In our experiments, we utilize Inverse Reinforcement Learning (IRL) to compute agent rewards based on demonstrated behavior. Specifically, we employ three IRL algorithms: Maximum Entropy IRL (Maxent-IRL) (Ziebart et al., 2008); Adversarial IRL; and Preferential Trajectory IRL (PT-IRL) (Santos et al., 2021). In addition, we compute manual rewards that differ due to potential shaping.

A trajectory $\tau_j = \{(s_0, a_0), (s_1, a_1), ..., (s_d)\}$ is a sequence of states and actions. A set of trajectories can be written as:

$$\varphi = \{\tau_1, \tau_2, ... \tau_h\}, h \in \mathbb{Z}^+.$$

**Maxent IRL** The objective of the Maxent IRL[7] algorithm is to compute a reward function that will generate a policy (learner) $\pi_L$ that matches the feature expectations of the trajectories generated by the expert's policy (demonstrations, assumed to be optimal) $\pi_E$. Formally, this objective can be expressed as:

$$\mathbb{E}_{\pi_L}[\phi(\tau)] = \mathbb{E}_{\pi_E}[\phi(\tau)], \tag{49}$$

where $\mathbb{E}_{\pi_k} = \sum_{\tau \in \varphi_k} p_{\pi_k}(\tau) \cdot \phi(\tau)$, and $p_{\pi_k}(\tau)$ is the probability distribution of selecting trajectory $\tau$ from $\pi_k$ generated by the set of trajectories $\varphi_k$. Maxent IRL assumes that the relationship between state features and agent rewards is linear. To resolve the ambiguity of having multiple optimal policies which can explain an agent's behavior, this algorithm applies the principle of maximum entropy to select rewards yielding a policy with the largest entropy.

---

[7]Maxent and AIRL implementations adapted from: https://github.com/HumanCompatibleAI/imitation (Gleave et al., 2022)

**AIRL:** The AIRL algorithm uses generative adversarial networks to train a policy that can mimic the expert's behavior. The IRL problem can be seen as training a generative model over trajectories as:

$$\max_w J(w) = \max_w \mathbb{E}_{\tau \sim \mathcal{D}}[\log p_w(\tau)], \tag{50}$$

where $p_{\mathbf{w}}(\tau) \propto p(s_0) \prod_{t=0}^{T-1} P(s_{t+1}|s_t, a_t) e^{\gamma^t \cdot R_w(s,a_t)}$ and the parameterized reward is $R_w(s,a)$. Using the gradient of $J(w)$, the **entropy-regularized policy objective** can be shown to reduce to:

$$\max_\pi \mathbb{E}_\pi \left[ \sum_{t=0}^{T} (R_w(s_t, a_t) - log\pi(a_t|s_t)) \right] \tag{51}$$

The discriminator is designed to take the form:

$$D_w(s,a) = \frac{\exp\{f_w(s,a)\}}{\exp\{f_w(s,a)\} + \pi(a|s)}, \tag{52}$$

and the **training objective** aims to minimize the cross-entropy loss between expert demonstrations and generated samples:

$$L(w) = \sum_{t=0}^{T} \left( -\mathbb{E}_{\mathcal{D}}[\log D_w(s_t, a_t)] - \mathbb{E}_{\pi_t}[\log(1 - D_w(s_t, a_t))] \right) \tag{53}$$

The policy optimization objective then uses the reward:

$$R(s,a) = log(D_w(s,a)) - log(1 - D_w(s,a)) \tag{54}$$

The AIRL formulation can be seen as an extension of the Guided Cost Learning (GCL) by Finn et al. (2016), with scalability modifications of analyzing data from a state-action level, rather than a trajectory-centric formulation. For example, in Finn et al. (2016), Equation 3.10 uses parameters $D_w(\tau)$ instead of $D_w(s,a)$. The AIRL formulation can also be shown to be equivalent to the Maxent IRL formulation.

**PTIRL:** The PTIRL algorithm incorporates multiple agents, each with a set of demonstrated trajectories $\varphi_i$. In order to compute the rewards for each agent, PTIRL considers target and non-target trajectories. Target trajectories are demonstrated trajectories from a target agent, and non-target trajectories are demonstrated trajectories from other agents. Denoting $P_w$ as the probability transition function for all the agents, the linear expected reward for each trajectory $\tau$ is defined as

$$LER(\tau) = \sum_{k=1}^{m} P_w(s_k', a_k, s_k) \cdot r(s_k', a_k, s_k).$$

For each trajectory set, there is a lower bound value $lb(\varphi)$ and an upper bound value $ub(\varphi)$, respectively defined as

$$lb(\varphi) = min_{\tau \in \varphi}(LER(\tau))$$

and

$$ub(\varphi) = max_{\tau \in \varphi}(LER(\tau)).$$

With $lb(\varphi)$ and $ub(\varphi)$ definitions, the spread $\delta$ is defined as

$$\delta(\varphi_a, \varphi_b) = lb(\varphi_a) - ub(\varphi_b).$$

Preferential ordering for any two trajectories $\varphi_a$ and $\varphi_b$ is denoted by a poset, $\prec$, such that if $\varphi_b \prec \varphi_a$, then $\delta(\varphi_a, \varphi_b) > 0$.

Given the above definitions, let $\varphi_i$ be the set of target trajectories and $\varphi_{ni}$ the set of non-target trajectories. Assuming that the rewards of the target agent will ensure its behavior towards the target trajectories, the PT-IRL objective is to find rewards such that $\varphi_{nt} \prec \varphi_i$, $\delta(\varphi_i, \varphi_{nt}) \geq \alpha$, where $\alpha$ is the minimum threshold for the spread. PT-IRL is generally fast because it directly computes rewards via linear optimization.

### C.4 Reward Classification: Additional Results

Table 8: Experiment 2: Welch's t-tests

| Domain | Rewards | SARD_vs_DIRECT | | SARD_vs_EPIC | | SARD_vs_DARD | |
|---|---|---|---|---|---|---|---|
| | | t-statistic | p-value | t-statistic | p-value | t-statistic | p-value |
| Gridworld | Manual | 11.522 | 0 | 12.478 | 0 | 10.385 | 0 |
| | Maxent | 28.496 | 0 | 28.142 | 0 | 2.593 | 0.005 |
| | AIRL | 13.610 | 0 | 5.117 | 0 | 4.266 | 0 |
| | PTIRL | 11.209 | 0 | 5.719 | 0 | 7.725 | 0 |
| Bouncing Balls | Manual | 18.801 | 0 | 17.375 | 0 | 6.955 | 0 |
| | Maxent | 32.341 | 0 | 12.104 | 0 | -2.586 | 0.995 |
| | AIRL | 45.020 | 0 | 28.226 | 0 | 19.488 | 0 |
| | PTIRL | 5.089 | 0 | 3.101 | 0.001 | 7.096 | 0 |
| Drone Combat | Manual | 16.152 | 0 | 15.851 | 0 | 16.786 | 0 |
| | Maxent | 17.829 | 0 | -2.543 | 0.994 | 9.123 | 0 |
| | AIRL | 9.772 | 0 | 8.023 | 0 | 3.935 | 0 |
| | PTIRL | 61.534 | 0 | 34.679 | 0 | 30.384 | 0 |
| Starcraft 2 | Manual | 24.419 | 0 | 20.633 | 0 | 15.760 | 0 |
| | Maxent | 6.171 | 0 | 1.717 | 0.043 | 2.233 | 0.013 |
| | AIRL | 4.992 | 0 | 4.300 | 0 | -2.913 | 0.998 |
| | PTIRL | 6.054 | 0 | 3.631 | 0 | 4.961 | 0 |

In Table 8, we show the comprehensive results for the Welch's t-tests for unequal variances, which are conducted across all domain and reward type combinations, to test the null hypotheses: (1) $\mu_{SARD} \leq \mu_{DIRECT}$, (2) $\mu_{SARD} \leq \mu_{EPIC}$, and (3) $\mu_{SARD} \leq \mu_{DARD}$; against the alternative: (1) $\mu_{SARD} > \mu_{DIRECT}$, (2) $\mu_{SARD} > \mu_{EPIC}$, and (3) $\mu_{SARD} > \mu_{DARD}$, where $\mu$ represents the sample mean. Generally, the tests indicate that (1) $\mu_{SARD} > \mu_{DIRECT}$ for all instances; (2) $\mu_{SARD} > \mu_{EPIC}$ for 11 out of 12 instances, and (3) $\mu_{SARD} > \mu_{DARD}$ for 10 out of 12 instances. These tests are performed at a significant level of $\alpha = 0.05$, assuming normality as per central limit theorem, since the number of trials is 200. In summary, we conclude that the SARD pseudometric is more effective at classifying reward samples compared to its baselines. Full details on accuracy scores collected are shown in Table 9 and Table 10.

Table 9: Experiment 2: Accuracy results

| Domain | Rewards | Statistic | DIRECT | EPIC | DARD | SARD |
|--------|---------|-----------|--------|------|------|------|
| Gridworld | Manual | mean | 69.8 | 69.3 | 70.0 | 75.8 |
| | | stdev | 4.6 | 4.6 | 5.0 | 4.6 |
| | Maxent | mean | 57.4 | 57.5 | 68.9 | 70.0 |
| | | stdev | 4.5 | 4.5 | 4.5 | 4.4 |
| | AIRL | mean | 82.3 | 84.9 | 85.0 | 86.2 |
| | | stdev | 3.0 | 1.8 | 2.6 | 2.7 |
| | PTIRL | mean | 82.2 | 84.2 | 83.4 | 86.0 |
| | | stdev | 3.5 | 3.3 | 3.5 | 3.3 |
| Bouncing Balls | Manual | mean | 46.5 | 47.3 | 52.0 | 55.2 |
| | | stdev | 4.8 | 4.6 | 4.8 | 4.5 |
| | Maxent | mean | 39.7 | 46.0 | 50.8 | 49.9 |
| | | stdev | 3.1 | 3.3 | 3.2 | 3.2 |
| | AIRL | mean | 41.2 | 46.1 | 49.8 | 56.3 |
| | | stdev | 3.4 | 3.9 | 3.3 | 3.3 |
| | PTIRL | mean | 70.3 | 71.1 | 69.5 | 72.4 |
| | | stdev | 4.2 | 4.3 | 4.1 | 4.0 |
| Drone Combat | Manual | mean | 67.1 | 67.2 | 66.2 | 73.9 |
| | | stdev | 4.1 | 4.2 | 4.9 | 4.2 |
| | Maxent | mean | 70.3 | 77.7 | 73.2 | 76.8 |
| | | stdev | 3.7 | 3.8 | 4.2 | 3.5 |
| | AIRL | mean | 90.1 | 90.7 | 92.3 | 93.8 |
| | | stdev | 3.9 | 3.9 | 3.7 | 3.7 |
| | PTIRL | mean | 52.5 | 63.7 | 65.1 | 78.3 |
| | | stdev | 4.3 | 4.3 | 4.6 | 4.1 |
| Starcraft 2 | Manual | mean | 65.5 | 67.4 | 69.5 | 76.5 |
| | | stdev | 4.6 | 4.5 | 4.5 | 4.4 |
| | Maxent | mean | 72.3 | 74.1 | 73.9 | 74.8 |
| | | stdev | 4.1 | 4.1 | 4.2 | 4.1 |
| | AIRL | mean | 75.1 | 75.3 | 78.1 | 77.0 |
| | | stdev | 4.0 | 4.0 | 3.8 | 3.8 |
| | PTIRL | mean | 77.2 | 78.1 | 77.6 | 79.6 |
| | | stdev | 4.1 | 4.2 | 4.2 | 4.0 |

Table 10: Experiment 2: Precision, Recall, F1-scores

| Domain | Rewards | Statistic | DIRECT | | | EPIC | | | DARD | | | SARD | | |
|---|---|---|---|---|---|---|---|---|---|---|---|---|---|---|
| | | | precision | recall | f1-score | precision | recall | f1-score | precision | recall | f1-score | precision | recall | f1-score |
| Gridworld | Maxent | mean | 64.7 | 57.6 | 56.8 | 64.8 | 57.7 | 56.9 | 72.9 | 68.8 | 67.8 | 76.1 | 70.1 | 69.6 |
| | | stdev | 3.5 | 4.1 | 4.1 | 3.4 | 4.0 | 4.0 | 3.7 | 4.9 | 4.8 | 3.6 | 4.3 | 4.2 |
| | Manual | mean | 74.4 | 70.0 | 68.8 | 73.6 | 69.4 | 68.1 | 74.0 | 70.1 | 68.4 | 78.7 | 76.1 | 73.2 |
| | | stdev | 4.1 | 4.4 | 4.7 | 4.5 | 4.4 | 4.8 | 4.5 | 4.5 | 4.9 | 4.2 | 3.3 | 4.2 |
| | AIRL | mean | 77.5 | 82.5 | 78.3 | 85.9 | 85.2 | 84.0 | 78.3 | 85.5 | 81.0 | 81.1 | 86.4 | 82.5 |
| | | stdev | 2.3 | 1.9 | 2.3 | 1.5 | 1.1 | 1.7 | 2.1 | 1.5 | 1.8 | 1.9 | 1.7 | 1.9 |
| | PTIRL | mean | 77.9 | 82.3 | 77.9 | 80.4 | 84.2 | 79.8 | 78.0 | 83.6 | 78.8 | 82.5 | 86.4 | 81.8 |
| | | stdev | 5.5 | 2.8 | 3.0 | 5.1 | 2.2 | 2.7 | 4.8 | 2.4 | 2.9 | 5.7 | 2.2 | 2.8 |
| Bouncing Balls | Manual | mean | 44.6 | 46.3 | 39.7 | 45.2 | 47.3 | 40.5 | 46.8 | 52.4 | 44.0 | 52.7 | 55.7 | 45.8 |
| | | stdev | 7.1 | 3.9 | 4.4 | 7.0 | 3.6 | 4.1 | 6.4 | 3.3 | 4.0 | 7.0 | 2.7 | 3.9 |
| | Maxent | mean | 33.5 | 39.7 | 34.1 | 41.1 | 45.9 | 40.9 | 51.1 | 50.6 | 47.3 | 42.9 | 49.8 | 42.4 |
| | | stdev | 2.6 | 2.3 | 2.0 | 3.1 | 2.6 | 2.4 | 5.6 | 3.0 | 3.1 | 3.4 | 2.8 | 2.8 |
| | AIRL | mean | 35.0 | 41.2 | 35.7 | 45.0 | 46.1 | 43.1 | 49.6 | 50.1 | 46.6 | 58.4 | 56.7 | 50.9 |
| | | stdev | 3.0 | 2.7 | 2.7 | 3.8 | 3.0 | 3.0 | 3.4 | 2.8 | 2.9 | 5.6 | 2.3 | 3.1 |
| | PTIRL | mean | 67.4 | 69.7 | 64.3 | 69.2 | 70.5 | 65.1 | 64.2 | 69.0 | 61.8 | 65.4 | 71.9 | 66.2 |
| | | stdev | 6.4 | 2.7 | 3.8 | 6.4 | 2.7 | 3.8 | 6.4 | 2.7 | 3.8 | 5.1 | 2.8 | 3.5 |
| Drone Combat | Manual | mean | 64.1 | 67.1 | 60.5 | 67.1 | 67.0 | 64.0 | 65.1 | 66.3 | 62.7 | 67.4 | 73.1 | 70.2 |
| | | stdev | 4.0 | 4.1 | 4.0 | 3.9 | 4.1 | 4.0 | 5.0 | 3.9 | 4.4 | 4.5 | 4.2 | 4.3 |
| | Maxent | mean | 65.1 | 70.3 | 67.6 | 74.1 | 77.6 | 72.8 | 74.6 | 73.5 | 71.0 | 74.0 | 76.6 | 71.3 |
| | | stdev | 3.8 | 3.6 | 3.7 | 3.6 | 3.8 | 3.7 | 4.5 | 3.2 | 3.8 | 4.0 | 3.5 | 3.7 |
| | AIRL | mean | 90.0 | 90.1 | 89.0 | 88.2 | 90.4 | 89.3 | 90.2 | 92.5 | 86.3 | 92.2 | 93.6 | 91.9 |
| | | stdev | 4.0 | 3.9 | 3.9 | 3.8 | 4.0 | 3.9 | 4.2 | 3.8 | 4.0 | 3.6 | 3.6 | 3.6 |
| | PTIRL | mean | 50.2 | 52.5 | 48.3 | 60.4 | 64.3 | 60.7 | 63.1 | 65.0 | 64.6 | 79.3 | 77.7 | 78.5 |
| | | stdev | 4.8 | 4.3 | 4.5 | 4.6 | 4.4 | 4.8 | 4.7 | 4.4 | 4.4 | 4.5 | 4.0 | 4.2 |
| Starcraft 2 | Manual | mean | 60.2 | 65.4 | 62.7 | 64.1 | 67.3 | 64.7 | 70.1 | 69.2 | 67.9 | 76.1 | 76.9 | 74.5 |
| | | stdev | 4.5 | 4.6 | 4.5 | 4.8 | 4.4 | 4.6 | 4.5 | 3.5 | 3.9 | 4.7 | 4.3 | 4.5 |
| | Maxent | mean | 73.1 | 72.2 | 69.6 | 70.4 | 74.1 | 70.2 | 69.1 | 73.2 | 70.1 | 72.7 | 74.7 | 73.7 |
| | | stdev | 4.0 | 4.1 | 4.0 | 4.5 | 4.5 | 4.3 | 4.4 | 4.2 | 4.1 | 4.0 | 4.2 | 4.1 |
| | AIRL | mean | 73.8 | 75.0 | 70.4 | 73.9 | 73.8 | 69.8 | 76.7 | 77.9 | 73.7 | 76.8 | 78.6 | 76.7 |
| | | stdev | 3.7 | 4.0 | 3.8 | 4.3 | 3.9 | 4.1 | 3.9 | 3.9 | 3.9 | 3.9 | 4.0 | 3.9 |
| | PTIRL | mean | 72.1 | 77.1 | 72.5 | 77.6 | 78.6 | 77.1 | 78.1 | 77.0 | 75.7 | 80.0 | 79.5 | 76.7 |
| | | stdev | 3.9 | 4.1 | 4.0 | 4.2 | 4.3 | 4.4 | 4.5 | 4.3 | 4.2 | 4.0 | 3.9 | 4.0 |

