# OpenReview forum: "Reward Distance Comparisons Under Transition Sparsity"
_TMLR — Rejected by TMLR_

### Review · Reviewer_tq11 · 2024-08-06

**Summary Of Contributions:**

This paper extends the existing EPIC metric and proposes a new metric named SARD to compare reward functions facing with high transition sparsity. A suite of experiments under different setting and reward functions are provided to validate the good performance of the proposed algorithm under transition sparsity.

**Audience:**

Yes

**Broader Impact Concerns:**

No ethical concerns.

**Claims And Evidence:**

Yes

**Requested Changes:**

Please explain the existence of transition from S4 to S5 and S3 to S6 (page 8) in more detail.

**Strengths And Weaknesses:**

Strength:
1. This work is mostly clear and easy to understand. The authors used several tables and graphs to illustrate their ideas.
2. The conclusion part includes several interesting future directions.
3. The experimental results are comprehensive and seem solid.

Weaknesses:
I am not an expert in this area so I will look into other reviewers' feedback carefully. And please consider my rating sparingly.
1. I am a little bit concerned about the theoretical results of the current work, which seems weak. Can you deduce some results in terms of regret bounds? e.g. if two reward functions are equivalent then the difference of rewards of optimal policies can be bounded by your proposed pseudometric?
2. I am still concerned about the existence of transition from S4 to S5 and S3 to S6 (page 8). Although you give some reasons, it doesn't necessarily mean that there will be direct transition between S4 to S5 (S3 to S6). More explanation on this topic will make this work better.
3. As you mention in your conclusion, since your method is directly extended from EPIC, the DARD also inherits the drawbacks of EPIC, e.g. only work for potential shaping.

---

> ### Author Response · Authors · 2024-08-26
> **Response to reviewer tq11**
>
> Thank you very much for the invaluable review. We have incorporated the feedback and addressed your comments below:
>
> > "I am a little bit concerned about the theoretical results of the current work, which seems weak. Can you deduce some results in terms of regret bounds? e.g. if two reward functions are equivalent then the difference of rewards of optimal policies can be bounded by your proposed pseudometric?"
>
> In the previous paper version, we had presented the regret bound in Appendix A.8. We have added it to the main paper for more visibility and clarity in the current revision (see Section 4, page 10).
>
> > "As you mention in your conclusion, since your method is directly extended from EPIC, the DARD also inherits the drawbacks of EPIC, e.g. only work for potential shaping."
>
> Thank you. Indeed, our method still assumes that rewards are potentially shaped. The advantage of potential shaping is that it is an additive function that theoretically guarantees policy invariance. In Appendix B.5, as an attempt to explore violations or deviations from potential shaping, we explore how the canonicalization methods (EPIC, DARD and SARD) perform when random noise (normally distributed) is added to potentially-shaped rewards. Experimental results show that all these methods experience some level of degradation in terms of performance. However, SARD still outperforms the other methods even in the presence of this noise. A promising direction for future work is to explore the impact of non-potential shaping methods and how they might influence reward canonicalization.
>
> > "Please explain the existence of transition from S4 to S5 and S3 to S6 (page 8) in more detail."
>
> Thank you we have refined the explanations as follows:
>
> (Transitions $(S_1, A, S_5) \subseteq (S_4, A, S_5)$.)
>
> Since $S_1$ are subsequent states to $s'$, and $S_4$ are subsequent states for all sampled transitions. It follows that, $S_1 \subseteq S_4, $ hence, $(S_1, A, S_5) \subseteq (S_4, A, S_5)$.
>
> (Transitions $(\hat{S}_2, A, S_6) \subseteq (S_3, A, S_6).$)
>
> $S_2$ is the set of subsequent states from $s$, and it can contain non-terminal and terminal states (usually fewer). Lets denote all non-terminal states in $S_2$ (excludes all terminal) by $\hat{S}_2$, such that the states in $\hat{S}_2$ are initial states for some sampled transitions. Since $S_3$ encompasses all initial states for all sampled transitions, it follows that: $\hat{S}_2 \subseteq S_3.$, hence, $(\hat{S}_2, A, S_6) \subseteq (S_3, A, S_6).$
> \end{quote}
>
> In addition, we have added an example in Appendix A.10 to illustrate this phenomenon on a sparse reward sample.

---

### Review · Reviewer_yGe2 · 2024-08-08

**Summary Of Contributions:**

This paper introduces the sparsity agnostic reward canonicalization method to facilitate comparisons of reward functions in terms of the optimal policies they induce. The paper applies this canonicalization method with the Pearson distance to arrive at the sparsity agnostic reward distance (SARD) pseudometric. SARD canonicalization improves upon the previous EPIC and DARD canonicalization methods by adjusting the correction term used to remove the influence of potential-based shaping. The SARD correction term weights transitions according to their probability of actually occurring, or more practically, the empirical frequency of the transitions in a data sample. This more accurate weighting is in contrast to the uniform weighting that EPIC uses and SARD adds expected reward terms with extra discounting to DARD canonicalization formula. Both of these changes allow SARD to be more accurate when transitions are missing from the data sample. The authors show that the SARD canonicalization's residual potential is no larger than DARD's and that it goes to zero as the data sample approaches complete transition coverage. The authors show that SARD outperforms its predecessors empirically in a range of environments.

**Audience:**

Yes

**Broader Impact Concerns:**

No broader impact concerns.

**Claims And Evidence:**

Yes

**Requested Changes:**

### Critical Changes
1. Please address the questions and concerns in the "Strength and Weaknesses" section.
2. In Definition 1 the following distributions are unused and unnecessary.
- $\mathcal{D}_{\mathcal{S}}$
- $\mathcal{D}_{\mathcal{A}}$

    $C_{\text{EPIC}}(R) = C_{\text{EPIC}}(R')$ is simply a fact given the rest of the setup. (Sorry for the awkward writing, the Markdown formatter gave me a lot of trouble with this item.)
3. It looks like the double summation in Equation 5 requires $O(N_M^2)$ time to compute but I agree with the bullet in A.9 that states you can compute an estimate of $\mathbb{E}[ R(S, A, S') ]$ in $O(N_M)$ time. Please clarify the discrepancy in the paper.
4. At the start of Section 3.4, $\mathcal{T}^{\mathcal{D}}$ is stated to be the set of "possible transitions", but it is defined as the cross product of coverage distributions. Is it a set or a probability distribution? "Possible" here is also an ambiguous word that could mean it includes hypothetical transitions with zero probability or it could mean "physically possible"/realizable transitions. Please clarify in the paper. Related, what are $T^S$, $T^U$, and $T^D$ used without definition shortly after?
5. Multiple times in Section 3, the word "undersampling" is used when it is meant to refer to missing both realizable and non-realizable transitions. Not having an unrealizable transition in a batch of data is not undersampling because there is no way to sample those transitions. Please correct this language.
6. $S_2$ may not be a subset of $S_3$, as the text makes clear (it can contain terminal states). If you want to provide a formal subset relationship, ensure to exclude the terminal states in $S_2$ first.
7. Equations 3 and 13 contain $S$, $A$, and $S'$, which are undefined and unnecessary. Please remove them.
8. Some of the notation is confusing. In Definition 2, $B_M$ is a batch of data but in Appendix A.3, it is a coverage distribution while $B_V$ is the batch. Also, to what do $M$ and $V$ refer? Did I miss their definitions?
9. The implications in the proof of Theorem 1 (Equations 14-17) are in the wrong direction. The current version assumes what you intended to prove and proves facts. Also consider formulating the proof in terms of bounding the difference between the two terms to be greater or less than zero rather than as a sequence of inequalities, I think that would be easier to follow.
10. The caption for Figure 3 does not fully explain the figure and provides analysis when that would be better included in the main text. What "Distance" is being measured on the vertical axes? Is "coverage" measured in terms of the state--action--state cross product or the set of realizable transitions? Do all the distance metrics reach zero at 100% coverage in the top row of figures? Do all of the methods start at the same point at 0% coverage?

### Changes to Strengthen the Work
11. I had to look up the definition of the Pearson coefficient. Since it is a critical part of the definition of all of the distance functions discussed in this paper, I think it should be defined here in the main paper.
12. It is very difficult to follow the relationships between the states in Definition 3. Why not label $S_4$ as $S'_3$, $S_5$ as $S'_1$, and $S_6$ as $S'_2$?
13. The motivation for Theorem 1 is weak. Is not DARD already a version of EPIC with approximations that are more accurate the presence of undersampling? I think can be useful to compare to $C_1$ in some ways since it is a natural first step toward SARD from DARD and EPIC, but Equation makes it clear that $C_1$ is not a safe canonicalization method. Theorem 1 is good verification that SARD's canonicalization is always better, but that result is unsurprising.
14. For Theorem 1, the assumption is unnecessary. The proof shows that the residual potential is never larger and is smaller if the condition holds, so why not say that?

**Strengths And Weaknesses:**

The overall thrust of this paper is good. It does make sense to me that including additional reward expectations in the canonicalization procedure would improve the accuracy of sample-based approximations, and that better canonicalization would lead to better reward function comparisons. This paper is also mostly well written and the progression from introduction to experiments is largely clear. However, I have a number of questions and concerns about this paper.

As pointed out in Figure 1, this paper talks about two core problems for computing reward canonicalization, (1) transitions missing from a data sample and (2) sparsity in the transition distribution. I agree that (1) is a problem and the paper is strongest when it addresses this issue directly. However, I am unconvinced that (2) is a problem in the way the paper presents.

At the end of the discussion on difficulties caused by feasibility constraints, the paper states that

> In this scenario, the extent of sampled transitions is directly influenced by the frequency of policy rollouts. When the number of rollouts is low, fewer transitions will likely be sampled, and conversely, when the number of rollouts is high, a greater proportion of transitions is likely to be sampled (see Figure 1a and 1b).

What are these sentences trying to communicate? The frequency of policy rollouts always determine how many transitions are sampled where fewer transitions are sampled when the number of rollouts is small, regardless of the extent of sparsity in the transition distribution. In fact, relative to the number of realizable transitions, the proportion of the transitions covered per step can increase substantially along with transition distribution sparsity.

For the example of transition distribution sparsity provided in Figure 1 (c), there are only 100 transitions that could be sampled when movement is restricted to the cardinal directions. I do not understand how the agent could explore 400 transitions or why that would be a problem. Since other transitions are impossible, the unshaped reward for those transitions can be arbitrary, so we can choose them to exactly cancel the potential based shaping and the results is that $R(x, u, x') = 0$ for each $(x, u, x')$ where $T(x, u, x') = 0$. In fact, transition distribution sparsity can improve reward canonicalization because it allows the entire realizable transition space can be captured in fewer samples.

This argument makes me doubt some of the criticisms of DARD. It does not require transitions from $S'$ to $S''$ that are impossible. Why do (presumably sample-based approximations of) DARD require the transition model? The DARD expectations can be estimated from data in the same way as EPIC and SARD, and they presumably must have for DARD to be applied in the experiments. I do agree that SARD approximations are likely to be more accurate, but I do not understand why $C_1$ is used as the important baseline for Theorem 1 instead of $C_{\text{DARD}}$.

To me, the potential problem that transition distribution sparsity represents is that if the start state distribution is small, then a sparse transition distribution can cause the sampling procedure to take a long time to sample from particular states that require a long sequence of actions to reach. This effect is especially problematic because these hard to reach states are often goal or near goal states that are the most important states for comparing reward functions. It does not come across to me that the paper's concern for transition distribution sparsity reflects a concerns along these lines.

This paper states that EPIC requires or assumes that all states are evenly sampled. It seems to me that this assertion is false. EPIC requires that the reward from each transition is evenly weighted. One can sample transitions from the MDP arbitrarily because an unbiased estimate of $\mathbb{E}[R(S, A, S')]$ under uniform unconditional distributions of $S$, $A$, and $S'$ is $1/|\mathcal{S} \times \mathcal{A} \times \mathcal{S}| \sum_{x, u, x' \in U} R(x, u, x')$ where $U \subseteq B$ is the set of unique transitions from batch $B$. This estimate also accounts for transitions that have zero probability in the MDP's transition function, or if $|\mathcal{S} \times \mathcal{A} \times \mathcal{S}|$ is unknown, then $|U|$ is a reasonable alternative. Why does Equation 5 not contain this re-weighting. Equation (5) looks biased even when the batch contains all transitions. If $B_M$ contains duplicates, then transitions with a small number of entries are underweighted, but even if there are no duplicates, no transition distribution sparsity, and $B_M$ contains exactly the transitions in $\mathcal{S} \times \mathcal{A} \times \mathcal{S}$, the size of the batch can be very different from the number of unique action--state pairs, required to approximate the first two expectations, and the number of unique transitions for the third expectation is $1/N_M$, not $1/N_M^2$.

Following on along this line, this paper does not clearly explain why it has arrived at particular approximations. Since there is no difference between the EPIC, DARD, and SARD canonicalizations when exact expectations can be evaluated across the entire transition space, the lack of space and attention around the presentation of sample-based approximations is concerning.

A glaring omission is a presentation for a sample-based SARD. The presentation that we do get in Appendix A.3 is also confusing. Why does the coverage distribution $B_M$ look like a set of action--state pairs? My assumption is that a "coverage distribution" is a probability distribution over transitions that is used to generate a batch of transition samples. If not, a definition for "coverage distribution" would help. Why are $X_3$ to $X_6$ instead sets of state--action--next state triples when they are also defined to be subsets of $B_M$?

This section shows that Proposition 2 applies to Equation 18 but then a new supposedly faster approximation is presented as Equation 21. Are these equations equivalent somehow? Does Proposition 2 apply to Equation 18? How is Equation 21 faster? Which approximation is used in experiments?

The step splitting the double sum in Appendix A.3 is annotated by saying merely that "it can be shown". I expect a proof to then actually show how or to provide evidence. In this case, I agree that neither is required, but that means the actual operations being performed should be explained, *e.g.*, "splitting the independent terms and canceling the redundant factors splits the double summation into two single summations".

This paper does not always clearly delineate between exact canonical reward functions with exact expectations and approximations using samples. For example, Theorem 1 appears to be an obvious consequence of Proposition 2 because Proposition 2 implies that the residual potentials in $C_{\text{SARD}}(R)$ are zero. However, looking at the proof of Theorem 1, it becomes clear that Theorem 1 is not actually making as statement about $C_{\text{SARD}}(R)$, it is making a statement about sample-based approximations thereof.

---

> ### Author Response · Authors · 2024-08-26
> **Response to reviewer yGe2 part (a)**
>
> Thank you very much for your comprehensive and insightful review. We identified and improved many areas that needed clarity in the current paper version. We have addressed your comments below:
>
> > The paper states that:
> "In this scenario, the extent of sampled transitions is directly influenced by the frequency of policy rollouts. When the number of rollouts is low, fewer transitions will likely be sampled, and conversely, when the number of rollouts is high, a greater proportion of transitions is likely to be sampled (see Figure 1a and 1b)."
> What are these sentences trying to communicate? The frequency of policy rollouts always determine how many transitions are sampled where fewer transitions are sampled when the number of rollouts is small, regardless of the extent of sparsity in the transition distribution. In fact, relative to the number of realizable transitions, the proportion of the transitions covered per step can increase substantially along with transition distribution sparsity."
>
> In this section, we mention that transition sparsity is caused by feasibility constraints and limited sampling. To illustrate feasibility constraints, we provide a Gridworld example, where restricted actions (only cardinal directions), result in a significantly smaller space of realizable transitions than the unrestricted case.
>
> To illustrate limited sampling, we provide an example, where we consider transitions that are sampled through trajectories from policy rollouts. Assuming that the "feasibility constraints, transition model, and the policy rollout method (e.g uniform) are kept constant", we expect that fewer policy rollouts will result in few transitions being sampled, and when the number of policy rollouts increases, a higher number of transitions will likely be sampled.
>
> To improve clarity, we have added the quoted assumption in the revised paper version.
>
> > For the example of transition distribution sparsity provided in Figure 1 (c), there are only 100 transitions that could be sampled when movement is restricted to the cardinal directions. I do not understand how the agent could explore 400 transitions.
>
> For this example, we are looking at a $10 \times 10$ Gridworld problem where each (x, y) coordinate is a state (x-axis from 0 to 9, and y-axis from 0 to 9). Therefore, we have a total of $100$ states ($10 \times 10$). For each state, only 4 transitions are possible (move up, right, down or left), therefore, as stated in the paper, "fewer than 400 transitions" will be sampled (4 directions * 100 states). Note that the exact number of transitions is 360 due to boundaries that restrict out-of-grid actions, for example, can't move from $(0, 0) \rightarrow (-1, 0)$.
>
> We have rephrased "fewer than 400 transitions" to "fewer than 400 transitions (based on $100$ states $\times$ 4 directions; the exact number is 360 due to boundary limitations, such as the inability to transition from $(0, 0)$ to $(-1, 0)$)".

---

> ### Author Response · Authors · 2024-08-26
> **Response to reviewer yGe2 part (b)**
>
> > Why that would be a problem. Since other transitions are impossible, the unshaped reward for those transitions can be arbitrary, so we can choose them to exactly cancel the potential based shaping and the results is that $R(x, u, x') = 0$ where $T(x, u, x') = 0$. In fact, transition distribution sparsity can improve reward canonicalization because it allows the entire realizable transition space can be captured in fewer samples.
>
> We considered this approach at one point, but it doesn't work well since unfeasible transitions actually degrade the effectiveness of canonicalization. To illustrate this, consider an MDP with two states, \$s_1$ and $s_2$, and a stochastic action $a_1$. Suppose we observe the transitions $(s_1, a_1, s_2)$ , $(s_1, a_1, s_1)$, and $(s_2, a_1, s_2)$, but the transition $(s_2, a_1, s_1)$ is unfeasible due to some feasibility constraint.
>
> Let $R$, $R_1$, and $R_2$, be equivalent reward functions that differ due to potential shaping, where: $R$ is the original unshaped reward function; $R_1$ is the reward function shaped by $\phi_1$, and $R_2$ is the reward function shaped by $\phi_2$. For the shaped rewards, considering the unfeasible transition, we have:
>
> \begin{align*}
> R_1(s_2, a_1, s_1) &= R(s_2, a_1, s_1) + \gamma \phi_1(s_1) - \phi_1(s_2), \\\\
> R_2(s_2, a_1, s_1) &= R(s_2, a_1, s_1) + \gamma \phi_2(s_1) - \phi_2(s_2).
> \end{align*}
>
> The suggested solution assigns an arbitrary value to $R(s_2, a_1, s_1)$ and sets both $R_1(s_2, a_1, s_1) = R_2(s_2, a_1, s_1) = 0$, such that:
> $$
> R(s_2, a_1, s_1) + \gamma \phi_1(s_1) - \phi_1(s_2) = R(s_2, a_1, s_1) + \gamma \phi_2(s_1) - \phi_2(s_2).
> $$
> which simplifies to:
> $$
> \gamma \phi_1(s_1) - \phi_1(s_2) - \gamma \phi_2(s_1) + \phi_2(s_2) = 0.
> $$
>
> This equation should hold for any valid potentially shaped functions. However, we can find values for the potential shaping terms and discount factor that violate this equation. For example, if $\gamma = 0.5$, $\phi_1(s_1) = 10$, $\phi_1(s_2) = 0$, $ \phi_2(s_1) = 0$, and $\phi_1(s_2) = 10$, the left-hand side (LHS) equals 15, while the right-hand side (RHS) equals 0. This is a contradiction since $15 \neq 0 $, hence this solution does not work mathematically.
>
> Lets also consider applying $C_{EPIC}$ to a shaped reward $R'$, under this solution. For the transition $(s_1, a_1, s_2)$:
> \begin{aligned}
> C_{EPIC}(R')(s_1, a_1, s_2) =\ & R'(s_1, a_1, s_2) + \frac{\gamma}{2} \left[ R'(s_2, a_1, s_1) + R'(s_2, a_1, s_2) \right] - \frac{1}{2} \left[ R'(s_1, a_1, s_1) + R'(s_1, a_1, s_2) \right] \\\\
> & - \frac{\gamma}{4} \left[ R'(s_1, a_1, s_1) + R'(s_1, a_1, s_2) + R'(s_2, a_1, s_1) + R'(s_2, a_1, s_2) \right].
> \end{aligned}
> If all transitions had been observed, then: $C_{EPIC}(R')(s_2, a_1, s_1) = C_{EPIC}(R)(s_2, a_1, s_1)$, and $\phi_{res} = 0$, showing effective canonicalization. Now, considering the unfeasible transition, $(s_2, a_1, s_1)$, and always assigning it to $0$ as a solution. This leads to:
>
> \begin{aligned}
> C_{EPIC}(R')(s_1, a_1, s_2) =\ & R'(s_1, a_1, s_2) + \frac{\gamma}{2} \left[ 0 + R'(s_2, a_1, s_2) \right] - \frac{1}{2} \left[ R'(s_1, a_1, s_1) + R'(s_1, a_1, s_2) \right] \\\\
> & - \frac{\gamma}{4} \left[ R'(s_1, a_1, s_1) + R'(s_1, a_1, s_2) + 0 + R'(s_2, a_1, s_2) \right].
> \end{aligned}
>
> In this scenario, $C_{EPIC}(R')(s_1, a_1, s_2) = C_{EPIC}(R)(s_1, a_1, s_2) + \phi_{res}$, where, $\phi_{res} = (\gamma/4)\phi(s_2) - (\gamma^2/4)\phi(s_1)$. Therefore, treating the unfeasible transition as $0$ is prone to residual shaping, which degrades the effectiveness of canonicalization. Hence, we cannot just assign any unfeasible transition as a zero reward.

---

> > ### Comment · Reviewer_yGe2 · 2024-08-28
> > **Re: Response to reviewer yGe2 part (b)**
> >
> > I see, thanks, that is helpful. However, since the goal is to compare reward functions according to the optimal policies they induce, I do not see a reason to force that $R_1(s_2, a_1, s_1) - \gamma \phi_1(s_1) + \phi_1(s_2) = R_2(s_2, a_1, s_1) - \gamma \phi_2(s_1) + \phi_2(s_2)$ for the distance between $R_1$ and $R_2$ to be zero. Reward functions that differ only on reward signals that are unreachable will induce identical sets of optimal policies and thus the distance between them should be zero.

---

> > > ### Author Response · Authors · 2024-08-29
> > > **Re: Response to reviewer yGe2 part(b), response 1**
> > >
> > > Thank you very much for the feedback!
> > >
> > > Indeed we do not need to set the shaped reward functions $R_1$ and $R_2$ to be the same for the distance between their optimal policies to be zero.
> > >
> > > In the provided example, we were illustrating the negative implications of a solution to handle infeasible transitions by treating $R(x, u, x') = 0$ whenever $T(x, u, x') = 0$, based on the question:
> > >
> > > > Why that would be a problem. Since other transitions are impossible, the unshaped reward for those transitions can be arbitrary, so we can choose them to exactly cancel the potential based shaping and the results is that $R(x, u, x') = 0$ where $T(x, u, x') = 0$. In fact, transition distribution sparsity can improve reward canonicalization because it allows the entire realizable transition space can be captured in fewer samples.
> > >
> > > Suppose we are comparing shaped reward functions $R_1$ (shaped by $\phi_1$) and $R_2$ (shaped by $\phi_2$) which both have an unfeasible transition, $(s_2, a_1, s_1)$. By implementing the solution (treat $R(x, u, x') = 0$ when $T(x, u, x') = 0$), it means that:
> > >
> > > $0 = R_1(s_2, a_1, s_1)$ and $0 = R_2(s_2, a_1, s_1)$, since $T(s_2, a_1, s_1) = 0$ for both reward functions (unfeasible transition).
> > >
> > > Therefore, $0 = R_1(s_2, a_1, s_1) = R_2(s_2, a_1, s_1)$,
> > >
> > > such that:
> > >
> > > \begin{aligned}
> > >     R(s_2, a_1, s_1) + \gamma \phi_1(s_1) - \phi_1(s_2) = R(s_2, a_1, s_1) + \gamma \phi_2(s_1) - \phi_2(s_2)
> > > \end{aligned}
> > >
> > > resulting in the form:
> > >
> > > \begin{aligned}
> > >     \gamma \phi_1(s_1) - \phi_1(s_2) - \gamma \phi_2(s_1) + \phi_2(s_2),
> > > \end{aligned}
> > >
> > > which can be a contradiction depending on the values of the potential functions. For example, if $\gamma = 0.5$, $\phi_1(s_1) = 10$, $\phi_1(s_2) = 0$, $\phi_2(s_1) = 0$, and $\phi_1(s_2) = 10$, the left-hand side (LHS) equals $15$, while the right-hand side (RHS) equals $0$.
> > >
> > > Therefore, this contradiction shows the challenge of a solution that assigns $R(x, u, x') = 0$ (or any constant $\alpha \in \\mathbb{R^+}$) when $T(x, u, x') = 0$; which is one of the reasons why developing canonicalization methods that can handle transition sparsity is important.

---

> > > > ### Author Response · Authors · 2024-08-29
> > > > **Re: Response to reviewer yGe2 part(b), response 2**
> > > >
> > > > > Why that would be a problem. Since other transitions are impossible, the unshaped reward for those transitions can be arbitrary, so we can choose them to exactly cancel the potential based shaping and the results is that $R(x, u, x') = 0$ where $T(x, u, x') = 0$. In fact, transition distribution sparsity can improve reward canonicalization because it allows the entire realizable transition space can be captured in fewer samples.
> > > >
> > > > It seems like the question might be asking about a solution to set the unshaped reward (not the shaped reward) to an arbitrary value i.e set $R(x, u, x') = 0$ whenever $T(x, u, x') = 0$
> > > >
> > > > The challenge with this proposed approach is that even if we set the unshaped reward for infeasible transitions to an arbitrary value (for example, $R(x, u, x') = 0$), we still don't know the structure of the shaping functions, as well as the discount factor for potential shaping. Therefore, this solution might not be that useful in canonicalizing rewards.
> > > >
> > > > For example, if we are canonicalizing the infeasible transition $(s_2, a_1, s_1)$ on a shaped reward, $R'$. Then setting $R(s_2, a_1, s_1) = 0$ implies that:
> > > >
> > > > $$R'(s_2, a_1, s_1) = 0 + \gamma \phi(s_1) - \phi(s_2),$$
> > > >
> > > > which reduces to:
> > > >
> > > > $$R'(s_2, a_1, s_1) = \gamma \phi(s_1) - \phi(s_2).$$
> > > >
> > > > The challenge is that during canonicalization, we actually do not know the exact form of $\phi(s_1)$ or $\phi(s_2)$ and $\gamma$, hence, we cannot determine the value of $R'(s_2, a_1, s_1)$. Therefore, this solution might not be that useful in canonicalizing reward functions, since the structure of the shaping function and $R'(s_2, a_1, s_1) $, are still open-ended.

---

> ### Author Response · Authors · 2024-08-26
> **Response to reviewer yGe2 part (c)**
>
> > This argument makes me doubt some of the criticisms of DARD. It does not require transitions from $S'$ to $S''$ that are impossible. Why do (presumably sample-based approximations of) DARD require the transition model? The DARD expectations can be estimated from data in the same way as EPIC and SARD, and they presumably must have for DARD to be applied in the experiments.
>
> From the original paper (Wulfe et al., 2022), the DARD psuedometric is defined as follows:
>
> "Let $R : S \times A \times S \rightarrow \mathbb{R}$ be a reward function. Given distributions $\\mathcal{D}_S \in \Delta(S)$ and $\\mathcal{D}_A \in \Delta(A)$ over states and actions, let $S$ be a random variable distributed as $\\mathcal{D}_S$, and $A$ be a random variable distributed as $\\mathcal{D}_A$. Furthermore, given a probabilistic transition model defining a conditional distribution over next states $T(S' \mid s, A)$, let $S'$ and $S''$ be random variables distributed as $T(S' \mid s, A)$ and $T(S'' \mid s', A)$ respectively. Define the dynamics-aware transformation of $R$ as follows:"
> $$
> C_T(R)(s, a, s') = R(s, a, s') + \mathbb{E}\left[\gamma R(s', A, S'') - R(s, A, S') - \gamma R(S', A, S'')\right].
> $$
>
> (Effect of transitions from $S'$ to $S''$.) As shown in the equation, the states $S'$ are subsequent to the state $s$, the states $S''$ are subsequent to $s'$. Hence, the expectations $\mathbb{E}[R(s, A, S')]$ and $\mathbb{E}[R(s', A, S'')]$, can be reliably estimated for any reward sample. However, for the transitions $(S', A, S'')$, we cannot guarantee the existence of these transitions since the states in $S''$, are not by definition, guaranteed to be subsequent to $S'$  (they are subsequent to $s'$). Under transition sparsity, computing $E[(S', A, S'')]$, can be highly sensitive to variations in transition distributions, due to unsampled transitions.
>
> (Why DARD uses transition models) Indeed the requirement for transition models in DARD can be relaxed, and in our experiments, we did not use transition models since in a lot of instances, we were comparing reward samples under high transition sparsity and it would not make sense to estimate the transition models with inadequate data. However, as shown in the DARD equation, the transition model is incorporated into the definition. The justification is that DARD can be highly sensitive to variations in transition sample distributions between rewards under comparison. One of the causes for this, is that: depending on the sampled transitions, DARD might not necessarily utilize all sampled batch transitions in reward canonicalization. For example, consider a reward sample, $R'$, with the following transitions: $\{(s_1, a_1, s_2), (s_1, a_1, s_3), (s_2, a_1, s_4), (s_4, a_1, s_5), (s_5, a_1, s_6)\}$. To canonicalize $R'(s_1, a_1, s_2)$, DARD only uses transitions for:
>
> $(s_1, A, S')$ - which are: ${(s_1, a_1, s_2), (s_1, a_1, s_3)}$,
>
> $(s_2, A, S'')$ - which are: ${(s_2, a_1, s_4)}$,
>
> $(S', A, S'')$ - which are: ${(s_2, a_1, s_4)}$
>
> Transitions $(s_4, a_1, s_5)$, and $(s_5, a_1, s_6)$ are not used in canonicalizing $(s_1, a, s_2)$, even though they are sampled. DARD is more reliant on what we will refer as "local proximity transitions" that are very close to $s$ and $s'$ including ($S'$ and $S''$), but not distant batch transitions that are further from $s$ and $s'$. This can make DARD to be highly unstable when comparing reward samples with significant differences in transition compositions. This challenge is one of the reasons why DARD incorporates transition models to maintain some level of consistency and uniformity, when comparing reward functions under different transition distributions.

---

> ### Author Response · Authors · 2024-08-26
> **Response to reviewer yGe2 part(d)**
>
> > I do agree that SARD approximations are likely to be more accurate, but I do not understand why $C_1$ is used as the important baseline for Theorem 1 instead of $C_{DARD}$.
>
>
> (Why build SARD from $C_1$ instead of $C_{DARD}$).  Lets first consider the equations from $EPIC$, $DARD$ and $C_1$.
>
> \begin{aligned}
> C_{EPIC}(R)(s, a, s') & = R(s, a, s') + \mathbb{E}[\gamma R(s', A, S') - R(s, A, S')  - \gamma R(S, A, S')].
> \end{aligned}
>
> \begin{aligned}
> C_{DARD}(R)(s, a, s') & = R(s, a, s') + \mathbb{E}[\gamma R(s', A, S'') - R(s, A, S')  - \gamma R(S', A, S'')].
> \end{aligned}
>
> \begin{aligned}
> C_1(R)(s, a, s') & = R(s, a, s') + \mathbb{E}[\gamma R(s', A, S_1) - R(s, A, S_2)  - \gamma R(S_3, A, S_4)].
> \end{aligned}
>
> The reason we built $C_{SARD}$ from $C_1$, is because the structure of $C_1$ is a much more general canonical form without assumptions on the reward transition distribution or any sensitivity to unsampled transitions, even though its theoretically prone to residual shaping.
>
> From EPIC, to derive $C_1$, we removed the assumptions that: $S'$ and $S$ are identically distributed; and also, the requirement than the subsequent states from $s$ and $s'$ are equal. If we build SARD from DARD, our first step is to relax DARD's assumptions and limitations as well. Basically, for DARD, computing $\mathbb{E}[R(S', A, S'')]$, can be highly sensitive to unsampled (feasible or unfeasible) transitions since $S''$ is defined as subsequent states to $s'$, not $S'$, hence, might not align well with the sampled transition distribution under transition sparsity. Additionally, as we previously mentioned, DARD can be reliant on local proximity transitions since ($S'$ and $S''$ are closer to $s$ and $s'$), but sampled transitions that are distant from $s$ and $s'$, might not be utilized in canonicalization. Therefore, to build SARD from DARD, the first step is to alleviate these challenges, by replacing $\mathbb{E}[R(S', A, S'')]$, with an expectation term such as $\mathbb{E}[R(S_3, A, S_4)]$, which considers transitions from the entire sampled batch (both local and distant). By doing so, DARD is also transformed into the form $C_1$.
>
> Therefore, we build SARD from $C_1$ since $C_1$ is a more general canonical form with minimal assumptions. If we relax assumptions in both EPIC or DARD, we will reach $C_1$ either way. Empirically, we have also found $C_1$ to outperform both EPIC and DARD (but not SARD), even though its not theoretically robust due to remaining residual potentials.
>
> > To me, the potential problem that transition distribution sparsity represents is that if the start state distribution is small, then a sparse transition distribution can cause the sampling procedure to take a long time to sample from particular states that require a long sequence of actions to reach. This effect is especially problematic because these hard to reach states are often goal or near goal states that are the most important states for comparing reward functions. It does not come across to me that the paper's concern for transition distribution sparsity reflects a concerns along these lines.}
>
> This is indeed a legitimate concern for all the studied reward psuedometrics. For this paper, the main focus is to mitigate the problem of transition sparsity, and we assumed that the sampled transitions are already available (an offline dataset), hence, we do not worry too much about challenges in transition sampling.
>
> Nonetheless, with regards to transition sampling, the sample-based SARD (as well as EPIC) alleviate this challenge by sampling transitions from the batch $B_M$, which contains $N_M$ samples from the joint state and action distributions, $D_S \times D_A$. In practice, state-action pairs in $B_M$ are sampled uniformly such that the reward expectations are likely less biased due to the transition model. For example, to estimate $\mathbb{E}[R(s, A, S_1)]$, the approximation: $\frac{\gamma}{N_1} \sum_{(x_1, u) \in X_1} R(s, u, x)$ is used, hence, the final state, $x$, equally considers all subsequent states to $s$, in $X_1 \subseteq B_M$. Therefore, states that would be difficult to reach due to the structure of the transition model, will likely have a high chance of being sampled as states that would be easier to reach.
>
> For the reward sample comparison tasks, we adopt the structure of $B_M$, but ensure to check whether each considered transition is defined within the reward sample transitions so that we do not consider unfeasible or unsampled transitions.
>
> We have added a comment in the paper to address this concern (in Section 4).
>
> "The structure of the batch, $B_M$ (often sampled uniformly), ensures that we likely consider both underrepresented and overrepresented transitions, to reduce bias in computations due to the transition model. Undefined reward triples (not present in reward sample due to feasibility constraints or limited sampling), $R(x_i, u, x_j)$, are ignored in the reward summation terms."

---

> ### Author Response · Authors · 2024-08-26
> **Response to reviewer yGe2 part (e)**
>
> > This paper states that EPIC requires or assumes that all states are evenly sampled. It seems to me that this assertion is false. EPIC requires that the reward from each transition is evenly weighted. One can sample transitions from the MDP arbitrarily because an unbiased estimate of $\mathbb{E}[R(S, A, S')]$ under uniform unconditional distributions of $S$, $A$, and $S'$ is $\frac{1}{|\mathcal{S} \times \mathcal{A} \times \mathcal{S}|} \sum_{x, u, x' \in U} R(x, u, x')$ where $U \subseteq B$ is the set of unique transitions from batch $B$. This estimate also accounts for transitions that have zero probability in the MDP's transition function, or if $|\mathcal{S} \times \mathcal{A} \times \mathcal{S}|$ is unknown, then $|U|$ is a reasonable alternative. Why does \textbf{Equation 5} not contain this re-weighting. \textbf{Equation (5)} looks biased even when the batch contains all transitions. If $B_M$ contains duplicates, then transitions with a small number of entries are underweighted, but even if there are no duplicates, no transition distribution sparsity, and $B_M$ contains exactly the transitions in $\mathcal{S} \times \mathcal{A} \times \mathcal{S}$, the size of the batch can be very different from the number of unique action-state pairs, required to approximate the first two expectations, and the number of unique transitions for the third expectation is $\frac{1}{N_M}$, not $\frac{1}{N_M^2}$.
> }
>
> (Evenly sampled vs evenly weighted) Our statement follows from the structure of Equation 5, directly taken from the original EPIC paper (Gleave et al., 2020):
>
> \begin{aligned}
>     C_{\\mathcal{D}_S, \\mathcal{D}_A}(R)(s, a, s') &= R(s, a, s') + \\mathbb{E}[\\gamma R(s', A, S') - R(s, A, S') - \\gamma R(S, A, S')] \\\\
>     &\\approx R(s, a, s') + \\frac{\\gamma}{N_M} \\sum\_{(x, u) \\in B_M} R(s', u, x)
>     - \\frac{1}{N_M} \\sum\_{(x, u) \\in B_M} R(s, u, x) - c,
> \end{aligned}
>
> where,
> $$
> c = \frac{\gamma}{N_M^2} \sum_{(x, u) \in B_M} \sum_{(x', u) \in B_M} R(x, u, x').
> $$
>
> Here, $B_M$ is a batch of $N_M$ samples from the joint state action distribution, $\mathcal{D}_S \times \mathcal{D}_A$.
>
> As you mention, under transition sparsity, this approximation has several weaknesses. First, having the same denominators $N_M$  for reward sums for $R(s', u, x)$ and $R(s, u, x)$, might be prone to bias, when either one of these reward term transitions have few transitions. Second, the denominator for all transitions, $N_M^2$ might be too large compared to the actual number of sampled transition in the batch, distorting the estimation of $\mathbb{E}[R(S, A, S')]$; and last, the existence of duplicates could introduce bias in the estimation of all reward expectations. The denominator $1/{N_M^2}$ is necessary for the above equation to satisfy proposition 1 (under full coverage). This also makes sense, since for the term $c$, the double summation iterates $B_M$ (or size $N_M$) twice for the $x$ and $x'$ states in $R(x, u, x')$. Hence, using $1/N_M^2$ (instead of $1/{N_M})$ is appropriate due to the state-action tuple structure of $B_M$.
>
> The advantage of this Equation is that it makes EPIC resilient to differences in transition distributions for reward samples under comparison. By sampling state-action pair combinations transitions from $B_M$, the EPIC canonicalization equally prioritizes transitions that are hard to sample and those that are easy to sample, under the specified reward transition models. If the sample-based EPIC just took a weighted average of the reward samples, then canonicalization would be highly sensitive to the bias in the distribution of transitions for the rewards under comparison. Another advantage of the above approximation is that it theoretically satisfies proposition 1, $C\_{EPIC}(R) = C\_{EPIC}(R')$, when all required transitions are available. However, using unbiased estimates to evenly weigh transitions, is equivalent to using $C_1$ (Equation 7), which theoretically yields some residual shaping (Equation 9); though in practice, we found the evenly weighted approximations to perform better than the given EPIC approximation.
>
> > Following on along this line, this paper does not clearly explain why it has arrived at particular approximations. Since there is no difference between the EPIC, DARD, and SARD canonicalizations when exact expectations can be evaluated across the entire transition space, the lack of space and attention around the presentation of sample-based approximations is concerning.}
>
> Thank you, we have added the approximation in Appendix A.3 to the main paper and explained the intuition behind its derivation.

---

> ### Author Response · Authors · 2024-08-26
> **Response to reviewer yGe2 part (f)**
>
> > A glaring omission is a presentation for a sample-based SARD. The presentation that we do get in Appendix A.3 is also confusing. Why does the coverage distribution $B_M$ look like a set of action--state pairs? My assumption is that a "coverage distribution" is a probability distribution over transitions that is used to generate a batch of transition samples. If not, a definition for "coverage distribution" would be helpful.
>
> As defined in the original EPIC (as well as DARD) papers, $B_M$ is a batch of $N_M$ samples from the joint state and action distributions. We have added the sample-based SARD (from Appendix A.3) approximation to the main paper, and discuss the intuition behind its derivation. The coverage distribution $\mathcal{D}$, is indeed the probability distribution over transitions used to generate the batch of transition samples. We have added this definition in Section 3.1 for clarity.
>
> > This section shows that Proposition 2 applies to Equation 18 but then a new supposedly faster approximation is presented as Equation 21. Are these equations equivalent somehow? Does Proposition 2 apply to Equation 18? How is Equation 21 faster? Which approximation is used in experiments?
>
> For the experiments, the first approximation was used since its form is in line with previous works, EPIC and DARD, and is more theoretically robust since it satisfies proposition 2.
>
> The first approximation can be slower due to the need to perform double summations over the batch of state-action pairs. The second approximation can be faster since we just sum all reward triples,  $(x_i, u, x_j)$, and normalize them (without a double loop as the first).
> The second approximation can also be highly sensitive to biases in transition distribution since it takes a weighted average of the sampled transitions sampled. By taking a weighted average over transitions sampled, the second approximation doesn't satisfy proposition 2, hence its theoretically less robust. Empirically the difference in performance between the two approximations is not that significant. To avoid confusion, we have removed the second approximation due to the limitations we mentioned.
>
> > The step splitting the double sum in Appendix A.3 is annotated by saying merely that "it can be shown". I expect a proof to then actually show how or to provide evidence. In this case, I agree that neither is required, but that means the actual operations being performed should be explained, e.g., "splitting the independent terms and canceling the redundant factors splits the double summation into two single summations".
>
> Thank you, we have added Lemma 1 for this derivation.
>
> > This paper does not always clearly delineate between exact canonical reward functions with exact expectations and approximations using samples. For example, Theorem 1 appears to be an obvious consequence of Proposition 2 because Proposition 2 implies that the residual potentials in $C_{\text{SARD}}(R)$ are zero. However, looking at the proof of Theorem 1, it becomes clear that Theorem 1 is not actually making a statement about $C_{\text{SARD}}(R)$, it is making a statement about sample-based approximations thereof.
>
> Yes, Theorem 1 is centered on addressing missing transitions in the reward samples. We have revised Section 4 to clarify this point.
>
> > Notation: $\mathcal{D}_S$ and $\mathcal{D}_A$
>
> Definition 1 (and its notation) is directly taken from the original work in EPIC, however, we agree that $\mathcal{D}_S$ and $\mathcal{D}_A$ seem unnecessary hence we have removed them in the revised version.
>
>
> > It looks like the double summation in Equation 5 requires $\mathcal{O}(N_M^2)$ time to compute, but I agree with the bullet in A.9 that states you can compute an estimate of $\mathbb{E}[R(S, A, S')]$ in $\mathcal{O}(N_M)$ time. Please clarify the discrepancy in the paper.
>
> Indeed to compute the double summation (iterating $B_M$ twice) takes $O(N_M^2)$ complexity, and an estimate, which involves summing all observed reward triples at once takes $O(N_M)$ complexity. Our implementation performed the double loop hence we have updated the EPIC complexity to  $O(N_M^2)$. EPIC is generally still easier to compute that SARD however, although asymptotically they have the same order of complexity.

---

> ### Author Response · Authors · 2024-08-26
> **Response to reviewer yGe2 part (g)**
>
> > At the start of Section 3.4, $\mathcal{T}^D$ is stated to be the set of "possible transitions", but it is defined as the cross product of coverage distributions. Is it a set or a probability distribution? "Possible" here is also an ambiguous word that could mean it includes hypothetical transitions with zero probability or it could mean "physically possible"/realizable transitions. Please clarify in the paper. Related, what are $\mathcal{T}^S$, $\mathcal{T}^U$, and $\mathcal{T}^D$ used without definition shortly after?
>
> We have refined these variables for clarity as follows:
>
> Consider a reward function $R: \mathcal{S \times A \times S}$, where $\mathcal{S}$ is the state space and $\mathcal{A}$ is the action space. Rewards are sampled from a coverage distribution, $\mathcal{D}$, and they span a state space $S^\mathcal{D} \subseteq S$ and an action space $A^\mathcal{D} \subseteq A$. We define the following sets of transitions:
>
> Full Coverage Transitions ($\mathcal{T^D}$) - The set of all transitions (could be realizable or not) that span the explored state and action space from the reward sample, $\mathcal{T^D} = S^\mathcal{D} \times A^\mathcal{D} \times S^\mathcal{D} \subseteq \mathcal{S \times A \times S}$.
>
> Sampled Transitions ($\mathcal{T}^S$) - The set of transitions that are actually present in the reward sample. Due to feasibility constraints and limited sampling, this set is a subset of full coverage transitions: $\mathcal{T}^S \subseteq \mathcal{T^D}$.
>
> Unsampled Transitions ($\mathcal{T}^U$) - The set of full coverage transitions that are not explored in the reward sample. These transitions can be both realizable and unrealizable, and they form a subset of the full coverage transitions: $\mathcal{T}^U \subseteq \mathcal{T^D} \setminus \mathcal{T}^S$, where $\mathcal{T}^U \subseteq \mathcal{T}^D$.
>
> > Multiple times in Section 3, the word "undersampling" is used when it is meant to refer to missing both realizable and non-realizable transitions. Not having an unrealizable transition in a batch of data is not undersampling because there is no way to sample those transitions. Please correct this language.
>
> Thank you, we have revised the paper to use the term "unsampled transitions", which include missing transitions due to feasibility constraints or limited sampling.
>
> > $S_2$ may not be a subset of $S_3$, as the text makes clear (it can contain terminal states). If you want to provide a formal subset relationship, ensure to exclude the terminal states in $S_2$ first.
>
> We have revised the sentence by defining $\hat{S}_2$ as a subset of non-terminal states from $S_2$ (excludes terminal states). It then follows that: $\hat{S}_2 \subseteq S_3$.
>
> > Equations 3 and 13 contain \( S \), \( A \), and \( S' \), which are undefined and unnecessary. Please remove them.
>
> Thank you, addressed this.
>
> > Some of the notation is confusing. In Definition 2, \( B_M \) is a batch of data but in Appendix A.3, it is a coverage distribution while \( B_V \) is the batch. Also, to what do \( M \) and \( V \) refer? Did I miss their definitions?
>
> We have removed $B_V$. Basically, in the previous EPIC and DARD works, $B_V$ referred to the sample of transitions from reward models, while $B_M$ is the sample of state-action pairs sampled in the from the joint state and action distributions.
>
> For our work, $B_M$ is sufficient since we already assume we have the transitions from the reward sample. We have thus removed $B_V$ from the paper's definitions for consistency and clarity.
>
> >The implications in the proof of Theorem 1 (Equations 14--17) are in the wrong direction. The current version assumes what you intended to prove and proves facts. Also, consider formulating the proof in terms of bounding the difference between the two terms to be greater or less than zero rather than as a sequence of inequalities; I think that would be easier to follow.
>
> Thank you, we have revised the proof to remove implications and it should be clearer now.
>
> > The caption for Figure 3 does not fully explain the figure and provides analysis when that would be better included in the main text. What "Distance" is being measured on the vertical axes? Is "coverage" measured in terms of the state--action--state cross product or the set of realizable transitions? Do all the distance metrics reach zero at 100\% coverage in the top row of figures? Do all of the methods start at the same point at 0\% coverage?
>
> We have changed the y-axis to $D_\rho$, to show the Pearson distance. We have also revised the caption in more detail.
>
> > I had to look up the definition of the Pearson coefficient. Since it is a critical part of the definition of all the distance functions discussed in this paper, I think it should be defined here in the main text.
>
> Thank you, we have added the definition.

---

> ### Author Response · Authors · 2024-08-26
> **Response to reviewer yGe2 part (h)**
>
> > It is very difficult to follow the relationships between the states in Definition 3. Why not label $S_4$ as $S_3'$, $S_5$ as $S_1'$, and $S_6$ as $S_2'$)?
>
> Thank you for the suggestion. We indeed considered this notation change to improve readability. However, we found that this might not flow well with some parts of the paper. For example, in one section we write:
>
> "To cancel $\mathbb{E}[\phi(S_i)], \, $ $\forall_i \in \\{1,...,4\\}$, we can add rewards $R(S_i, A, k_i)$ to induce potentials $\gamma \phi(k_i) - \phi(S_i)$
> "
>
> Having a mixture of the primed and unprimed notation, e.g {$S_3$, $S_3'$, $S_1$, $S_1'$}, might make these descriptions less succinct and potentially harder to follow. There are several other instances, where introducing the primed notation, might complicate the notation. However, if this is a significant issue, we are open to making the necessary changes.
>
> > For Theorem 1, the assumption is unnecessary. The proof shows that the residual potential is never larger and is smaller if the condition holds, so why not say that?
>
> Thank you, have addressed this.
>
>
> References
>
> Wulfe, Blake, Logan Michael Ellis, Jean Mercat, Rowan Thomas McAllister, and Adrien Gaidon. "Dynamics-Aware Comparison of Learned Reward Functions." In International Conference on Learning Representations, 2022.
>
> Gleave, Adam, Michael D. Dennis, Shane Legg, Stuart Russell, and Jan Leike. "Quantifying Differences in Reward Functions." In International Conference on Learning Representations, 2020

---

> ### Comment · Reviewer_yGe2 · 2024-09-04
> **Re: Re: Response to reviewer yGe2 part(b), response 2**
>
> > It seems like the question might be asking about a solution to set the unshaped reward (not the shaped reward) to an arbitrary value i.e set $R(x, u, x') = 0$ whenever $T(x, u, x') = 0$.
>
> Yes, I am asking about that idea, thanks. However, there's no reason I see to set $R(x, u, x') = 0$. Instead, set it to $R(x, u, x') = \phi(s_2) - \gamma \phi(s_1)$. That way, the actual reward signal (including shaping) that you would observe from that reward function (if that transition were possible) would be zero. Since $R(x, u, x')$ can be set independently and arbitrarily for every reward function, both the underlying reward and the shaping can be eliminated on each unrealizable transition for every reward function. I think that eliminates all possible errors related to unrealizable transitions.
>
> Since the only reward signals that can influence the optimal policy sets are the ones on realizable transitions, I see no reason why the distance between reward functions should depend on signals assigned to unrealizable transitions.

---

> ### Author Response · Authors · 2024-09-05
> **Response to reviewer yGe2 part (b), response 3**
>
> > Yes, I am asking about that idea, thanks. However, there's no reason I see to set $R(x, u, x') = 0$. Instead, set it to $R(x, u, x') = \phi(s_2) - \gamma\phi(s_1)$. That way, the actual reward signal (including shaping) that you would observe from that reward function (if that transition were possible) would be zero. Since $R(x, u, x')$ can be set independently and arbitrarily for every unrealizable transition, both the underlying reward and the shaping can be eliminated on each unrealizable transition for every reward function. I think that eliminates all possible errors related to unrealizable transitions. Since the only reward signals that can influence the optimal policy sets are the ones on realizable transitions, I see no reason why the distance between reward functions should depend on signals assigned to unrealizable transitions.
>
>
> Thank you for the detailed response.
>
> The suggested approach independently and arbitrarily sets: $R(x, u, x') = \phi(s_2) - \gamma \phi(s_1)$ such that $R'(x, u, x') = 0$, for unrealizable transitions.
>
> The challenge with this approach is that whatever values of $\phi(s_2)$ and $\phi(s_1)$ we arbitrarily set, must be compatible with the shaping terms of the realizable transitions as well. However, during canonicalization, we usually don't know the structure of the potential shaping function beforehand. Thus, if we arbitrarily set incorrect values for the shaping components within the unrealizable transitions, we can introduce significant errors in the computation.
>
> To illustrate with an example, consider a reward function with a state space $S = (s_1, s_2)$, and an action space $A = {a_1}$. Assume we want to canonicalize a shaped reward sample $R'$ with the following observed realizable transitions: $\\{ (s_1, a_1, s_1), (s_1, a_1, s_2), (s_2, a_1, s_2) \\}$ and the unrealizable transition:
> $\\{(s_2, a_1, s_1) \\}$.  Suppose we observed $R'$ with the following shaped reward values for the realizable transitions: $R'(s_1, a_1, s_1) = -5$, $R'(s_1, a_1, s_2) = 5$, $R'(s_2, a_1, s_2) = 15$. Note that we don't have values for $R'(s_2, a_1, s_1)$ since its unrealizable.
>
> From the reward sample information provided during canonicalization, $R'$, we don't know beforehand, the values for the potential shaping function, $\\{ \phi(s_1), \phi(s_2) \\}$. Therefore, we don't really have an easy way to arbitrary set: $R(s_2, a_1, s_1) = \phi(s_2) - \gamma \phi(s_1)$. Note that the values we set for $\phi(s_1)$ and $\phi(s_2)$ on the unrealizable transition, should be the same as the **unknown actual shaping function values** for the realizable transitions, inorder for the suggested approach to work well.
>
> Provided all necessary transitions are available, canonicalization methods provide great utility since they can transform $R'$ into canonical forms: $C_{SARD}(R)$, $C_{DARD}(R)$ and $C_{EPIC}(R)$ that are free of potential shaping; **without prior knowledge of the potential shaping function beforehand**.

---

> ### Comment · Reviewer_yGe2 · 2024-09-05
> **Re: Response to reviewer yGe2 part (b), response 3**
>
> Why does one need to know the values of $\phi$? The value of the reward function, including shaping, can be assumed to be zero at unrealizable transitions. This assumption is valid, even when the reward function is constructed with a shaping potential because we can say that the unshaped reward at unrealizable transitions exactly cancels the shaping. The expectations used for canonicalization can then be split in two, one over realizable transitions and another over unrealizable transitions, which is zero.

---

> ### Author Response · Authors · 2024-09-05
> **Response to reviewer yGe2 part(b), response 4**
>
> > Why does one need to know the values of $\phi$? The value of the reward function, including shaping, can be assumed to be zero at unrealizable transitions. This assumption is valid, even when the reward function is constructed with a shaping potential because we can say that the unshaped reward at unrealizable transitions exactly cancels the shaping. The expectations used for canonicalization can then be split in two, one over realizable transitions and another over unrealizable transitions, which is zero.
>
> Thank you for the prompt response and insightful comment.
>
> Indeed the suggested approach of treating the unrealizable rewards, including the shaping as $0$ could be a possible solution, but its not theoretically robust since its prone to shaping. Using the previous example, lets assume we have a shaped reward function $R'$, with realizable transitions: $\\{(s_1, a_1, s_1)$, $(s_1, a_1, s_2)$, $(s_2, a_1, s_2)\\}$ and the unrealizable transition $(s_2, a_1, s_1)$.
>
> Applying $C_{EPIC}$ for the transition $(s_1, a_1, s_2)$, we get:
> \begin{aligned}
> C_{EPIC}(R')(s_1, a_1, s_2) =\ & R'(s_1, a_1, s_2) + \frac{\gamma}{2} \left[ R'(s_2, a_1, s_1) + R'(s_2, a_1, s_2) \right] - \frac{1}{2} \left[ R'(s_1, a_1, s_1) + R'(s_1, a_1, s_2) \right] \\\\
> & - \frac{\gamma}{4} \left[ R'(s_1, a_1, s_1) + R'(s_1, a_1, s_2) + R'(s_2, a_1, s_1) + R'(s_2, a_1, s_2) \right].
> \end{aligned}
>
> If all transitions had been observed, $C_{EPIC}(R')(s_2, a_1, s_1) = C_{EPIC}(R)(s_2, a_1, s_1)$, and $\phi_{res} = 0$, showing effective canonicalization. **Now, lets consider a solultion where we assign the reward for the unfeasible transition, $(s_2, a_1, s_1)$ as $0$, and consider all realizable and unrealizable transitions in canonicalization**. This leads to:
>
> \begin{aligned}
> C_{EPIC}(R')(s_1, a_1, s_2) =\ & R'(s_1, a_1, s_2) + \frac{\gamma}{2} \left[ 0 + R'(s_2, a_1, s_2) \right] - \frac{1}{2} \left[ R'(s_1, a_1, s_1) + R'(s_1, a_1, s_2) \right] \\\\
> & - \frac{\gamma}{4} \left[ R'(s_1, a_1, s_1) + R'(s_1, a_1, s_2) + 0 + R'(s_2, a_1, s_2) \right].
> \end{aligned}
>
>
> In this scenario, $C_{EPIC}(R')(s_1, a_1, s_2) = C_{EPIC}(R)(s_1, a_1, s_2) + \phi_{res}$, where, $\phi_{res} = (\gamma/4)\phi(s_2) - (\gamma^2/4)\phi(s_1)$. Therefore, this solution is prone to residual shaping.
>
> **Lets now consider the recently proposed solution, where we only apply $C_{EPIC}$ to the realizable transitions, after separating the realizable and unrealizable transitions**. The modified equation for $C_{EPIC}(s_1, a_1, s_2)$ using the realizable transitions only becomes:
>
> \begin{aligned}
> C_{EPIC}(R')(s_1, a_1, s_2) =\ R'(s_1, a_1, s_2) &+ \frac{\gamma}{1} \left[R'(s_2, a_1, s_2) \right] - \frac{1}{2} \left[ R'(s_1, a_1, s_1) + R'(s_1, a_1, s_2) \right] \\\\
> & - \frac{\gamma}{3} \left[ R'(s_1, a_1, s_1) + R'(s_1, a_1, s_2) + R'(s_2, a_1, s_2) \right].
> \end{aligned}
>
> In this scenario, $C_{EPIC}(R') = C_{EPIC}(R) + \phi_{res}$, where $\phi_{res} = (\gamma^2/3 - \gamma/6)\phi(s_2) + (-\gamma^2/3 + \gamma/6)\phi(s_1)$. As shown, this solution is also prone to shaping.
>
> Another potential downside of such an approach is that we need to know whether an unsampled transition is unrealizable or realizable so that unrealizable transitions can be treated as $0$. This would require labeling all unrealizable transitions beforehand, which sometimes can be inconvenient or impractical, especially when comparing reward samples without information on whether the missing transitions are unfeasible or a consequence of limited sampling.

---

> > ### Comment · Reviewer_yGe2 · 2024-09-13
> > **Responses to various comments**
> >
> > Thanks for your responses. I have a few outstanding comments and questions.
> >
> > **Re: Response to reviewer yGe2 part(d)**
> >
> > My question was not "Why build SARD from $C_1$ instead of $C_{\text{DARD}}$?", it was "why is $C_1$ the important baseline for Theorem 1?" Theorem 1 is exactly a statement about errors from shaping in expectation so $C_1$ would seem to be a straw-man comparison. It would be surprising for an unsound formula like $C_1$ to have smaller errors than SARD. A better comparison would be SARD against a sound method like DARD.
> >
> >
> > **Re: Response to reviewer yGe2 part (e)**
> >
> > I am confused why there is any sampling of the batch at all. If you walk over the batch once to identify all the unique transitions, you can compute the estimate from these unique transitions without any duplicates. Are you assuming that all the transitions are actually unique, e.g., if the state or action spaces are actually continuous?
> >
> > Regardless, this response did not answer my question about the language around how EPIC requires state transitions to be "identically distributed". The expectations in EPIC do not have to be computed under any distribution related to the actual sampling of transitions or the proportion of transitions in the batch. An expectation under a uniform distribution is merely a uniform average and that can be computed regardless of how the batch is constructed.
> >
> > > Another advantage of the above approximation is that it theoretically satisfies proposition 1, $C_{\text{EPIC}}(R) = C_{\text{EPIC}}(R')$, when all required transitions are available.
> >
> > I do not understand. The estimate method cannot ensure that the exact canonicalization function, $C_{\text{EPIC}}$, removes potential shaping. Are you saying that even though the given estimate of EPIC is biased even when all transitions are available, that it nonetheless is guaranteed to remove all potential shaping? An unbiased estimate for EPIC could do that as well so I do not see how this is an advantage.
> >
> > The new content around sample-based estimates is still unsatisfying because they do not include proofs of unbiasedness or that a statement like Proposition 2 holds. I see that it is stated that Proposition 2 holds for the SARD given estimate (which technically does not make sense because Proposition 2 is specific to $C_{\text{SARD}}$) but is the proof so obvious that it need not be stated? Especially when we are talking about a batch of data that may contain duplicates, the proof that something like Proposition 2 holds for the SARD estimate is not obvious to me.
> >
> >
> > **Re: Response to reviewer yGe2 part(b), response 4**
> >
> > Thanks, I understand now. All of the canonicalization methods in this line of work require the cross product transitions to ensure that all of the potential terms cancel. This assumption should be included in the statement of Proposition 2.
> >
> > I think a better assumption would be that the MDP is communicating, because that is all you should need to ensure that all potential terms cancel. But I admit that this conjecture is beyond the scope of this work.
> >
> > This understanding also leads me to question the definition of SARD (Definition 3). $\mathcal{D}\_{\mathcal{S}}$ and $\mathcal{D}\_{\mathcal{A}}$, but what role do they play in the definition? $S\_1, \ldots, S\_6$ are defined as random initial or subsequent states, so it is unclear how $\mathcal{D}\_{\mathcal{S}}$ and $\mathcal{D}\_{\mathcal{A}}$ could be used to define the distributions for these states. Maybe the definition simply requires the expectations to be uniform over the set of possible states (under the initial state or subsequent state constraints described in Definition 3)? But if we need to assume that all transitions are realizable, does that not ultimately make all the $S\_i$ variables identical? I think they may also make $C\_{\text{SARD}} = C\_{\text{EPIC}}$.
> >
> > In the proof of Proposition 2, there is an $\mathbb{E}[-\gamma^2( \phi(S\_4) + \phi(S\_4))]$ term that is non-zero. Is this a typo?

---

> ### Author Response · Authors · 2024-09-17
> **Re: Responses to reviewer yGe2 part (d), response 1**
>
> >My question was not ``Why build SARD from $\\mathcal{C}_1$ instead of  $\\mathcal{C}_{DARD}$?", it was ``why is $\\mathcal{C}_1$ the important baseline for Theorem 1?".  A better comparison would be SARD against a sound method like DARD.
>
> Thank you for the note, initially we thought $C_1$ would be an appropriate comparison since it relaxes many assumptions for EPIC. But as you mentioned, it might not be the most "sound" justification for SARD.
>
> We have updated Theorem 1, and we now compare SARD to both EPIC and DARD. The proof is now based on the comparisons of relative shaping errors, which we define to quantify the impact of shaping normalized by the the base or unshaped canonical reward function.
>
> Comparing the shaping terms without reward normalization might not be robust, since, if the base canonical reward is large enough, the impact of shaping is less and vice-versa, hence, the RSE normalizes the shaping term.
>
> We have revised the paper, to include the new Theorem 1, which compares all the three algorithms.

---

> ### Author Response · Authors · 2024-09-18
> **Re: Response to Reviewer yGe2, part (e), part 1**
>
> > I am confused why there is any sampling of the batch at all. If you walk over the batch once to identify all the unique transitions, you can compute the estimate from these unique transitions without any duplicates. Are you assuming that all the transitions are actually unique, e.g., if the state or action spaces are actually continuous?
>
> In the EPIC work, the goal was not to tackle challenges such as transition sparsity, hence, in their experiments, they assumed access to all the necessary transitions during reward comparisons. In developing the sample-based methods, they used batches to compare rewards that might be represented as neural networks. The reward networks would provide the reward, even when states and actions where continuous. The batch notation was important because the rewards under comparison could be infinite, which is the case with continuous domains.
>
> In DARD, their main goal was to address the fact that some transitions might be unfeasible. To mitigate this, DARD incorporated transition models to reduce the effect of feasibility concerns. However, they still assumed access to the ground truth reward models (nueral networks), hence, they needed the sample-based methods.
>
> In our work, our primary goal is to address the challenge of reward comparisons under transition sparsity, which can occur due to feasibility constraints or limited sampling in the reward samples under comparison. We do not assume access to the ground truth reward model, but only to the samples of rewards we need to compare. Therefore, while we incorporated the batch notation (for consistency with other approximation methods such as DARD and EPIC), it might not be necessary for the use-case where we are comparing reward samples under transition sparsity. However, the batch notation can still be useful when the samples are too large (such as Starcraft 2), and canonicalizing all transitions is nearly impossible.
>
> In relation to duplicate state-action pairs in $B_M$, the previous works do not discuss how they handled any of these issues. In our work, for the SARD approximation, we derive sets $X_i$, for $i \\in \\{1,.., 6\\}$ from $B_M$, hence, we don't have challenges with duplicates. For our implementations of EPIC and DARD, we also use sets to avoid duplicates.
>
> > Regardless, this response did not answer my question about the language around how EPIC requires state transitions to be "identically distributed". The expectations in EPIC do not have to be computed under any distribution related to the actual sampling of transitions or the proportion of transitions in the batch. An expectation under a uniform distribution is merely a uniform average and that can be computed regardless of how the batch is constructed.
>
> The original EPIC equation (from Gleave et al., 2020) is described as:
>
> \begin{equation}
> C_{\mathcal{D}_S, \mathcal{D}_A}(\mathcal{R})(s, a, s') = \mathcal{R}(s, a, s') + \mathbb{E}\left[\gamma \mathcal{R}(s', A, S') - \mathcal{R}(s, A, S') - \gamma \mathcal{R}(S, A, S')\right].
> \end{equation}
>
> Applying the equation to a potentially shaped reward, $R'(s, a, s') = R(s, a, s') + \gamma \phi(s') - \phi(s)$, you get the result:
>
> $$C_{EPIC}(R') = C_{EPIC}(R) + \phi_{res}$$,
>
> where, $$\phi_{\text{res}} = \gamma \mathbb{E}[\phi(S)] - \gamma \mathbb{E}[\phi(S')]$$
>
> To ensure that $\phi_{res} = 0$, EPIC assumes that $S$ and $S'$ are identically distributed, such that: $\mathbb{E}[\phi(S)] = \mathbb{E}[\phi(S')]$. This is also mentioned in the original paper as part of the proof for the proposition to show that: $C_{EPIC}(R') = C_{EPIC}(R)$:
>
> ``.... where the penultimate step uses $\mathbb{E}[\Phi(S')] = \mathbb{E}[\Phi(S)]$ since $S$ and $S'$ are identically distributed." (Gleave et al., 2020)
>
> If $S$ and $S'$ are not identically distributed, then the impact of shaping is likely higher under the approximation presented in EPIC.
>
> Note that EPIC does not necessarily require "transitions" to be identically distributed, but requires states $S'$ and $S$ to be identically distributed. However, this implies that each state should have transitions connected to identical states to all other explored states, since from the equation, we need transitions from $(s, A, S')$, $(s', A, S')$ and $(S, A, S')$; as well as the requirement that $S$ and $S'$ are identically distributed.

---

> ### Author Response · Authors · 2024-09-18
> **Re: Response to Reviewer yGe2, part (e), part 2**
>
> >**Another advantage of the above approximation is that it theoretically satisfies proposition 1, $C_{EPIC}(R) = C_{EPIC}(R')$, when all required transitions are available**
>
> >I do not understand. The estimate method cannot ensure that the exact canonicalization function, $C_{\text{EPIC}}$, removes potential shaping. Are you saying that even though the given estimate of EPIC is biased even when all transitions are available, that it nonetheless is guaranteed to remove all potential shaping? An unbiased estimate for EPIC could do that as well so I do not see how this is an advantage.
>
> Thank you, indeed, this statement is not necessary.  When all transitions are available, the EPIC equation as well as the unbiased estimate will satisfy proposition 1. However, the presented structure of EPIC, is necessary, due to how the distance estimate is computed, as described below:
>
> (Both the EPIC and DARD papers, did not really provide any justification for the structure of their approximation methods, hence, we are hypothesizing what could be the likely motivation.)
>
> A possible reason why EPIC takes the current sample-based method structure, instead of taking the unbiased estimate is that their estimate **could be more efficient and fairly accurate to compute under the objective of satisfying the requirement for identical distribution of $S$ and $S'$, and no transition sparsity**
>
> The EPIC approximation first samples a batch $B_V$ of transition samples, and then another batch $B_M$ from the state-action distribution.
>
> Consider a task to canonicalize a reward function with $1000$ states, and $100$ actions, under computational constraints. To perform EPIC, it is necessary to ensure that we have transitions: $(s, A, S')$, $(s, A, S')$ and $(S, A, S')$. Basically every state in the transition batch, should have transitions to $S'$, under the same actions, $A$.
>
> With the EPIC approximation, you could sample $100$ $(s, a, s')$,transitions for $B_V$, and sample $10 states$ and $10 actions$, to have a cross-product of $N_M = 100$ state-action pairs for $B_M$.
>
> For each $(s, a, s') \in B_V$:
>
> $\mathbb{E}[R(s, A, S')]$ is computed from $100$ ($N_M$) transitions.
>
> $\mathbb{E}[R(s', A, S')]$ is computed from $100$ ($N_M$) transitions
>
> $\mathbb{E}[R(S, A, S')]$ is computed from $100 *100 = 10 000$ ($N_M^2$) transitions (double sum needed).
>
> This satisfies the structure of the approximation Equation 6, even though all the possible transitions where not explored. Note that for each $(s, a, s')$ canonicalization, this estimation uses significantly fewer transitions than using the entire reward function: ($10^4$ vs $10^{10}$)
> For the sampled batches, the approximation still has all the necessary $N_M$ transitions needed to approximate EPIC such that:
>
> ($s$ goes to $S'$ through $A$ actions)
>
> ($s'$ goes to $S'$ through $A$ actions)
>
> ($S$ goes to $S'$ through $A$ actions)
>
> Note that $S = S'$ by virtue of the double summation implementation. Therefore, this approximation still satisfies the identical distribution assumption that EPIC needs, even though we do not have the entire full set of transitions from the reward function.
>
> **However, once transition sparsity is introduced, this approximation, is prone to error due to missing transitions. Some of the transitions in $N_M$ cannot be explored, because of feasibility constraints or limited data from the input reward samples.
> This is where the EPIC approximation presented significantly struggles.

---

> ### Author Response · Authors · 2024-09-18
> **Re: Response to yGe2 part (b)**
>
> > The new content around sample-based estimates is still unsatisfying because they do not include proofs of unbiasedness or that a statement like Proposition 2 holds. I see that it is stated that Proposition 2 holds for the SARD given estimate (which technically does not make sense because Proposition 2 is specific to $C_{\text{SARD}}$) but is the proof so obvious that it need not be stated? Especially when we are talking about a batch of data that may contain duplicates, the proof that something like Proposition 2 holds for the SARD estimate is not obvious to me.
>
> We appreciate the feedback. The proof holds only if the reward sample has enough transitions to cover all terms in the approximation. We have added proposition 3 for the sample-based SARD, with this assumption, as well as the proof in the main paper.
>
> > Thanks, I understand now. All of the canonicalization methods in this line of work require the cross product transitions to ensure that all of the potential terms cancel. This assumption should be included in the statement of Proposition 2.
>
> Thank you, addressed this.
>
> > This understanding also leads me to question the definition of SARD (Definition 3). $\\mathcal{D}_S$ and $\\mathcal{D}_A$, but what role do they play in the definition? $\\{S_1, \\ldots, S_6\\}$ are defined as random initial or subsequent states, so it is unclear how $\\mathcal{D}_S$ and $\\mathcal{D}_A$ could be used to define the distributions for these states. Maybe the definition simply requires the expectations to be uniform over the set of possible states (under the initial state or subsequent state constraints described in Definition 3)  ? But if we need to assume that all transitions are realizable, does that not ultimately make all the $S_i$ variables identical  ?
>
> >I think they may also make $C_{SARD} = C_{EPIC}$.
>
> Thank you for the observation. Indeed our SARD definition does not clearly show how $\{S_1, .. S_6\}$ can be drawn from $\mathcal{D}_S$ and $\mathcal{D}_A$.
>
> Inorder for the SARD definition to be standalone and different from EPIC, we need a rule to generate transitions sets $\{S_1, ... S_6\}$ given states $s$ and $s'$. The most reasonable solution is to define a transition model governing how transitions are defined. We have updated the paper to include this.
>
> > In the proof of Proposition 2, there is an $\mathbb{E}[-\gamma^2 (\phi(S_4) + \phi(S_4))]$ term that is non-zero. Is this a typo?
>
> Thank you, this was a typo and we have fixed it.

---

### Review · Reviewer_TBym · 2024-08-16

**Summary Of Contributions:**

In sequential decision problems, characterizing the similarity of reward functions in terms of different aspects allows one to compare and contrast different reward functions to identify better reward functions among similar reward functions, and also to evaluate the agent behavior that maximizes a particular reward function.  While theoretically well grounded, existing reward similarity measuring pseudometrics might be unsuitable for practical sequential decision problems such as reinforcement learning (RL) where there might exist high transition sparsity (low coverage of states). In this paper, the authors propose a novel reward similarity measuring pseudometric called Sparsity Agnostic Reward Distance (SARD) that can improve reward comparisons in environments with high transition sparsity. Theoretical results for the validity of the proposed pseudometric is provided. Empirical evaluation of SARD and comparison with existing reward comparison baselines shows the applicability and efficacy of SARD in practice.

**Audience:**

Yes

**Broader Impact Concerns:**

N/A (Theoretical work)

**Claims And Evidence:**

Yes

**Requested Changes:**

* Could the authors add an explanation as to why the accuracy of reward comparison for DIRECT and EPIC methods seems comparable to that of SARD with reward functions learned using AIRL, for example, in the Drone Combat domain?
* Please refer to the Minor Weaknesses above for minor changes needed.

**Strengths And Weaknesses:**

Strengths

* The problem considered in the paper is clearly defined and well-motivated.
* The solution method to the problem is explained in an intuitive way, which is fairly easy to understand.
* Theoretical results for the robustness and validity of the proposed pseudo metric are provided, and the proposed pseudo metric is * empirically validated by experiments with varying complexity.

Weaknesses

* In Experiment 2 (Table 1), why can simple methods like EPIC (and in some domains, DIRECT) perform comparably to SARD (except for the Bouncing Ball domain) when considering the reward function learned from AIRL? Does this suggest the effect of policy invariance and transition sparsity on reward comparison can be avoided by the reward function learning method? For example, this similarity in performance is very much evident in the Drone Combat domain.
* While SARD is more robust to under-sampling compared to EPIC, it is not clear why SARD can be ‘agnostic’ to transition sparsity, since SARD seems to be affected by transition sparsity to some extent.


Minor Weaknesses

* Page 5 last paragraph, should $\mathcal{D}_ \mathcal{S} \in \mathcal{S}$ and $\mathcal{D}_ \mathcal{A} \in \mathcal{A}$ be $\mathcal{D}_ \mathcal{S} \subseteq \mathcal{S}$ and $\mathcal{D}_ \mathcal{A} \subseteq \mathcal{A}$, respectively?
* Using $\mathcal{D}_ \mathcal{S}$, $\mathcal{D}_ \mathcal{A}$ for distributions over $\mathcal{S}$,$\mathcal{A}$ (e.g., in  Proposition 1) and also subspaces of $\mathcal{S}$,$\mathcal{A}$ (Section 3.4 first paragraph) is bit confusing.
* Usage of $T^S$, $T^D$ and $\mathcal{T}^\mathcal{S}$, $\mathcal{T}^\mathcal{D}$ seemingly interchangeably (Section 3.4 first paragraph) is a bit confusing.

---

> ### Author Response · Authors · 2024-08-26
> **Respose to reviewer TBym**
>
> Thank you very much for the insightful review. We have addressed the questions and incorporated your feedback into our revised paper version. We address the comments below:
>
> > "Could the authors add an explanation as to why the accuracy of reward comparison for DIRECT and EPIC methods seems comparable to that of SARD with reward functions learned using AIRL, for example, in the Drone Combat domain?"
>
> Indeed, all algorithms generally achieve higher accuracy on rewards computed via AIRL, in comparison to other IRL algorithms. This is likely due to the robustness of AIRL to reward shaping, as AIRL has the ability to learn rewards "disentangled from the dynamics of the environment they are trained on" (Fu et al., 2018). In fact, the discriminator for the AIRL algorithm is designed to handle reward shaping, by taking the form:
>
> $$D_{\theta, \phi}(s, a, s') = \frac{\exp\{f_{\theta,\phi}(s, a, s')\}}{\exp\{f_{\theta,\phi}(s, a, s')\} + \pi(a \mid s)},$$
>
> where $f_{\theta, \phi}$ is restricted to a reward approximator $g_{\theta}$ and a shaping term $h_{\phi}$ as:
>
> $$f_{\theta,\phi}(s, a, s') = g_{\theta}(s, a) + \gamma h_{\phi}(s') - h_{\phi}(s)$$
>
> As mentioned in Fu et al., (2018), the additional shaping term helps mitigate the effects of unwanted shaping on our reward approximator $g_{\theta}$. Therefore, the results from AIRL show that the ability to handle shaping can definitely be incorporated in the design of IRL algorithms. We have added a note in the paper, describing this result, and exploring the robustness of different IRL to shaping is a interesting direction for future work.
>
> > "While SARD is more robust to under-sampling compared to EPIC, it is not clear why SARD can be 'agnostic' to transition sparsity, since SARD seems to be affected by transition sparsity to some extent."
>
> Thank you for your insightful comment. The term 'agnostic' is part of our proposed algorithm's name "Sparsity Agnostic Reward Distance". The naming of the algorithm is intended to underscore SARD's high robustness to transition sparsity, which is a significant improvement over existing methods. We acknowledge, however, that the term might be a bit optimistic, as SARD still exhibits sensitivity to extreme sparsity conditions. However, we still believe this name effectively communicates SARD's overarching goal of addressing and mitigating reward comparisons under transition sparsity; and it highlights the significance of this problem as well.
>
> > "Page 5 last paragraph, should $\mathcal{D}_S \subseteq \mathcal{S}$ and $\mathcal{D}_A \subseteq \mathcal{A}$ be $\mathcal{D}_S \subseteq \mathcal{S}$ and $\mathcal{D}_A \subseteq \mathcal{A}$, respectively?"
> > "Using $\mathcal{D}_S$, $\mathcal{D}_A$ for distributions over $\mathcal{S}$, $\mathcal{A}$ (e.g., in Proposition 1) and also subspaces of $\mathcal{S}$, $\mathcal{A}$ (Section 3.4 first paragraph) is a bit confusing."
>
> We have updated the notation. We now use $S^\mathcal{D} \subseteq \mathcal{S}$, to denote the states covered in the sample, and  $A^\mathcal{D} \subseteq \mathcal{A}$ to denote the actions covered in the sample.
>
> > "Usage of $\mathcal{T}^S$, $\mathcal{T}^D$ and $\mathcal{T}^S$, $\mathcal{T}^D$ seemingly interchangeably (Section 3.4 first paragraph) is a bit confusing."
>
> We have refined the section and added descriptions to describe these transitions. The notation should be easier to follow now.
>
>
>
> Reference for AIRL:
>
> Fu, Justin, Katie Luo, and Sergey Levine. "Learning Robust Rewards with Adversarial Inverse Reinforcement Learning." In International Conference on Learning Representations. 2018.

---

> > ### Comment · Reviewer_TBym · 2024-09-01
> >
> > Thank you for the response and the clarification. I think for the sake of clarity and correctness, it's better to explain the usage of the word 'agnostic' in SARD in the paper.

---

> ### Author Response · Authors · 2024-09-02
> **Re: Response to reviewer TBym: response 1**
>
> > Thank you for the response and the clarification. I think for the sake of clarity and correctness, it's better to explain the usage of the word 'agnostic' in SARD in the paper.
>
> Thank you for the feedback and request to update the paper for improved clarity.
>
> We have revised the paper by explaining the rationale for using the term 'agnostic', in Section 4, just before the presentation of the sample-based SARD approximation method.
>
> "The term \`agnostic' in $C_{SARD}$ underscores its robustness and flexibility in accommodating variations in transition distributions when canonicalizing reward samples that may lack full coverage but have enough transitions to compute each reward expectation term in Equation 13. However, in many practical scenarios with high transition sparsity, some transitions required for at least one of the reward expectation terms in $C_{SARD}$ might still remain unsampled. In these cases (with missing transitions), although the term `agnostic' might seem somewhat optimistic, it still reflects SARD's high robustness relative to other methods. This robustness is facilitated by the strategic choices of $k_i$ in $C_{SARD}$ (Equation 13), $S_5$ and $S_6$, ensuring that ... "

---

### Decision · Action_Editor_iD8x · 2024-09-17

**Recommendation:** Reject

**Comment:**

According to the reviews and the recommendations, the paper is rated as borderline. Thus, I have carefully considered the reviews as well as the expertise and experience of the reviewers (with one of the positive reviewers not being confident in the assessment) and came to the conclusion that there is too much uncertainty about assumptions and conditions that are crucial for the theoretical soundness as well as the wording, which can lead to misunderstandings.

Addressing these concerns will require a major revision and, thus, another round of reviews.

**Audience:**

All reviewers agree that the direction of the paper is interesting for TMLR.

**Claims And Evidence:**

The paper supports its claims with theoretical and experimental results. However, there is some unclarity about the assumptions and conditions needed for the theoretical soundness.

**Resubmission Of Major Revision:**

The authors may consider submitting a major revision at a later time.